# Gaze-centered gating, reactivation, and reevaluation of economic value in orbitofrontal cortex

Demetrio Ferro [1,2] ✉, Tyler Cash-Padgett[3], Maya Zhe Wang [3],
Benjamin Y. Hayden [4] & Rubén Moreno-Bote [1,2,5]

During economic choice, options are often considered in alternation, until commitment. Nonetheless, neuroeconomics typically ignores the dynamic aspects of deliberation. We trained two male macaques to perform a value-based decision-making task in which two risky offers were presented in sequence at the opposite sides of the visual field, each followed by a delay epoch where offers were invisible. Surprisingly, during the two delays, subjects tend to look at empty locations where the offers had previously appeared, with longer fixations increasing the probability of choosing the associated offer. Spiking activity in orbitofrontal cortex reflects the value of the gazed offer, or of the offer associated with the gazed empty spatial location, even if it is not the most recent. This reactivation reflects a reevaluation process, as fluctuations in neural spiking correlate with upcoming choice. Our results suggest that look-at-nothing gazing triggers the reactivation of a previously seen offer for further evaluation.

Introspection and a myriad of studies imply that eye fixations play important roles in decision-making. For instance, when confronted with multiple visual options, we tend to look at them in sequence until we commit to a choice[1–4]. In addition, we tend to look longer, and most imminently at the offers that we finally choose[4–6], a process that possibly starts with a coarser, covert evaluation stage[7–9]. These results raise the possibility that there is a causal interplay between eye movements and choice deliberation[10]. At a first glance, the sequential nature of fixations can be thought of as an inevitable consequence of the design of the visual system, as visual objects need to be foveally sampled to be accurately processed[11]. However, the evaluation of options seems to follow a sequential structure also in non-visually guided, abstract decisions[12]. Therefore, a sequential evaluation of options may be the default computation in economic choices[13]. This view is supported by the computational benefits of a focused evaluation of fewer options as compared to an overall screening of all available options[14–16] as well as by anatomical, physiological, and evolutionary considerations[17–19].

Neurons in several brain areas, most notably the orbitofrontal cortex (OFC) and ventromedial prefrontal cortex (vmPFC), show modulations in firing rate activity that correlate with the economic value of choice targets in vision[18,20–32] or with the initiation of gaze transitions to choice-relevant cues[33]. One current view is that partially segregated neural populations are selective for different available offers, and each independently computes its value as they are compared[34,35]. A similar view holds in perceptual decision-making, where it has been shown that several competing motor outputs can be prepared in parallel by partially segregated neural populations in premotor and prefrontal areas[36,37]. However, some brain areas seem to exclusively represent the options that are attended[13,38–40], with some work also showing that neurons in OFC and value-coding brain regions in general represent offer values in alternation, not in parallel[26]. Therefore, it is still unclear how many alternative offers can be simultaneously encoded, to what extent offer values are simultaneously computed and compared, and how these processes depend on gaze

[1]Center for Brain and Cognition, Universitat Pompeu Fabra, 08002 Barcelona, Spain. [2]Department of Information and Communication Technologies, Universitat Pompeu Fabra, 08002 Barcelona, Spain. [3]Department of Neuroscience, Center for Magnetic Resonance Research, University of Minnesota, Minneapolis MN55455, USA. [4]Department of Neurosurgery, Baylor College of Medicine, Houston, TX 77030, USA. [5]Serra Húnter Fellow Programme, Universitat Pompeu Fabra, Barcelona, Spain. ✉e-mail: demetrio.ferro@upf.edu

and attention patterns. Answering these questions is important to understand the canonical computations underlying value-based choice.

To tackle these questions, the behavioral paradigm must consider that the encoding of visual offers (e.g., recognition) and their evaluation are not conflated, which automatically excludes tasks where offers are permanently visible. This is necessary since it is not otherwise possible to determine whether sequential processing of offers is the result of the need to move the eyes to scan the visual scene and direct attention in sequence to the alternative stimuli. An effective decoupling of encoding and evaluation can be achieved by interleaving the offer presentations with empty screen delay epochs, which is the approach adopted here. Further, to test whether evaluation is sequential by default, we let subjects free to move their eyes at their will, and expect to be able to track the internal deliberation process by using the *look-at-nothing* effect[41–43], a visuo-motor tendency to gaze at locations currently empty where relevant stimuli were formerly presented. Looking at empty locations is thought to aid memory recollection of the formerly presented visual stimuli[42,44]—like looking at the center of the yard to remember the maple tree that used to be there. We hypothesized that look-at-nothing may be associated with the retrieval of memories of choice options for their further evaluation.

We used a two-alternative economic choice task where sequentially presented visual offers were respectively followed by empty screen delays. We tracked the uninstructed gazing behavior of two macaque subjects both during the respective offer presentation (look-at-something) and during blank screen delays (look-at-nothing). As expected, we found that subjects directed their gaze to the offers when visible, with longer fixations on the offer most likely to be chosen later. The same pattern is observed during the blank screen delays: the more the empty side is looked at, the higher the probability that the offer formerly presented at that location will be chosen. We confirmed that this effect is independent of the expected value, magnitude, and risk of the offers by factoring them out in this analysis. In simultaneous recordings of OFC neurons, we confirm the well-known finding that neural activity represents the value of the gazed offer. More interestingly, looking at the empty location where a previous offer was presented elicits a neural reactivation of its value, regardless of whether another offer was seen and encoded in the interim. This representation reflects an active reevaluation process, as fluctuations of neural activity during look-at-nothing, while controlling for offer subjective values, correlate with choice. We find that the encoding of value during stimulus presentation and reactivation occurs in overlapping cell populations. Our results show an unexpected connection between gazing at empty stimuli locations, reactivation, and reevaluation, which reveals a sequential default mode underlying economic choice.

## Results

### Performance and look-at-nothing gazing in a gambling task
We combine the analysis of choices, oculomotor data and spiking neural recordings of two macaque monkeys performing a visually cued, two-alternative reward gambling task (Fig. 1A; Supplementary Tables ST1-ST2 for exact number of trials)[28,45]. The task consists of the sequential presentation of two vertical bar stimuli (*offer 1 / offer 2* for 400 ms) at the opposite sides of a visual screen display. Each stimulus presentation is followed by an empty screen delay time (*delay 1 / delay 2* for 600 ms). Subjects are left free to move their gaze at their will during both offer and delay epochs. At the end of *delay 2*, subjects are instructed to re-acquire fixation (*re-fixate*) for at least 100 ms at the center of the screen. Upon fixation reacquisition, the two offers appear side by side on their old locations (*choice-go* cue), which cues the animals to choose by performing a saccade to the target side. The reward magnitude *m* of an offer is indicated by the color of the bottom part of its associated vertical bar stimulus (gray: small; blue: medium; green: large; see Methods), which could include a red top part. The height of the bottom part indicates the (success) probability *p* of

obtaining the liquid reward of magnitude *m* if the offer is chosen. The height of the red part corresponds to the complementary probability $1 - p$ for an unsuccessful outcome. If the gamble is unsuccessful, no reward is provided. We define the expected value $EV = mp$ of an offer as the product of its magnitude *m* and its probability *p*, while its risk is defined as the value variance, $\sigma^2 = mp(1 - p)$. We define subjective value as $SV = w_1 EV + w_2 m + w_3 \sigma^2$, which we compute for each session via logistic regression of choice (Methods "Neural encoding of offer subjective value and choice"; Supplementary Table ST3).

Both subjects performed the task by successfully reporting the choice and following value-based contingencies (most often choosing the offer with the higher expected value; subject 1: 72.19%; subject 2: 75.72%). We find a significant relationship between choice and difference in expected value for the two offers (Fig. 1B, logistic regression, $p < 0.001$; Supplementary Fig. S1A, for each subject). As the task consists of the sequential presentation of visual offers on opposite sides of the screen, we expected subjects to perform overt visual search to collect sensory information, thus to estimate the value of the offers. Indeed, the eye position of the two subjects mainly follows the stimuli locations (Fig. 1C, top), when offers are presented. In most trials, subjects first fixate the first offer location (*offer 1*: 78.83% and 81.57% of the trials with first offer on the left and right side, respectively; Supplementary Table ST2), then the opposite location during the second offer epoch (*offer 2*: 75.78% and 81.82% when second offer is on the left and right side, respectively; Supplementary Table ST2). Analogous results are found by computing the fraction of trials where subjects look more on average at the right or at the left of the screen in 10 ms bins (Fig. 1C, bottom).

When we analyze eye data in two dimensions and map them on the screen, we find that gaze is concentrated on the offer locations also along vertical axis, i.e., subjects preferentially look at the physical rectangular shapes of the offers (Fig. 1D, e.g., *offer 1* and *2* epochs; Supplementary Fig. S2A for each subject). Interestingly, we observe that gaze locations during delay times show a look-at-nothing bias: subjects tend to look at the location of the formerly presented stimuli (*delay 1* and *2* epochs in Fig. 1D), indicating that in the absence of stimuli, subjects recall the spatial location where the stimulus had been presented. This effect is unsurprising during *delay 1*, as gaze most often remains on the same side where the first offer has been presented, but less so during *delay 2*: subjects asymmetrically split looking time between the two sides of the screen (see the two bumps in *delay 2*), approximately matching the locations of the two formerly presented offers. These results are observed regardless of whether the first offer is presented on the left or right (Supplementary Figs. S1B, S2B). For this reason, in all following analyses we mirrored data by arbitrarily referencing the first offer to be shown on the left, and the second one on the right, and kept this alignment in all subsequent results. Besides doubling the data size per condition, folding the visual screen to a single arbitrary reference also simplifies the presentation of results (Fig. 1D). Using this convention, further analyses show, as expected, that subjects tend to look more at the sides with the best offer values (Supplementary Fig. S2C, S2E) and to the locations that correspond to the offers that we will be finally chosen (Supplementary Fig. S2D, S2F), both during offer presentation and, interestingly, also during delay epochs.

### Gaze position modulates choice during look-at-something and look-at-nothing
To determine the role of gaze in evaluation and choice, we tested whether look-at-something and look-at-nothing gazing are predictive of choices. First, we analyzed the relationship between choice and expected value for three gaze conditions and for each task epoch: using all trials, using only trials where the subject spends a larger fraction of time looking at the right side of the screen in that epoch ($f_R > 0.75$), or using trials where the subject spends a larger fraction of

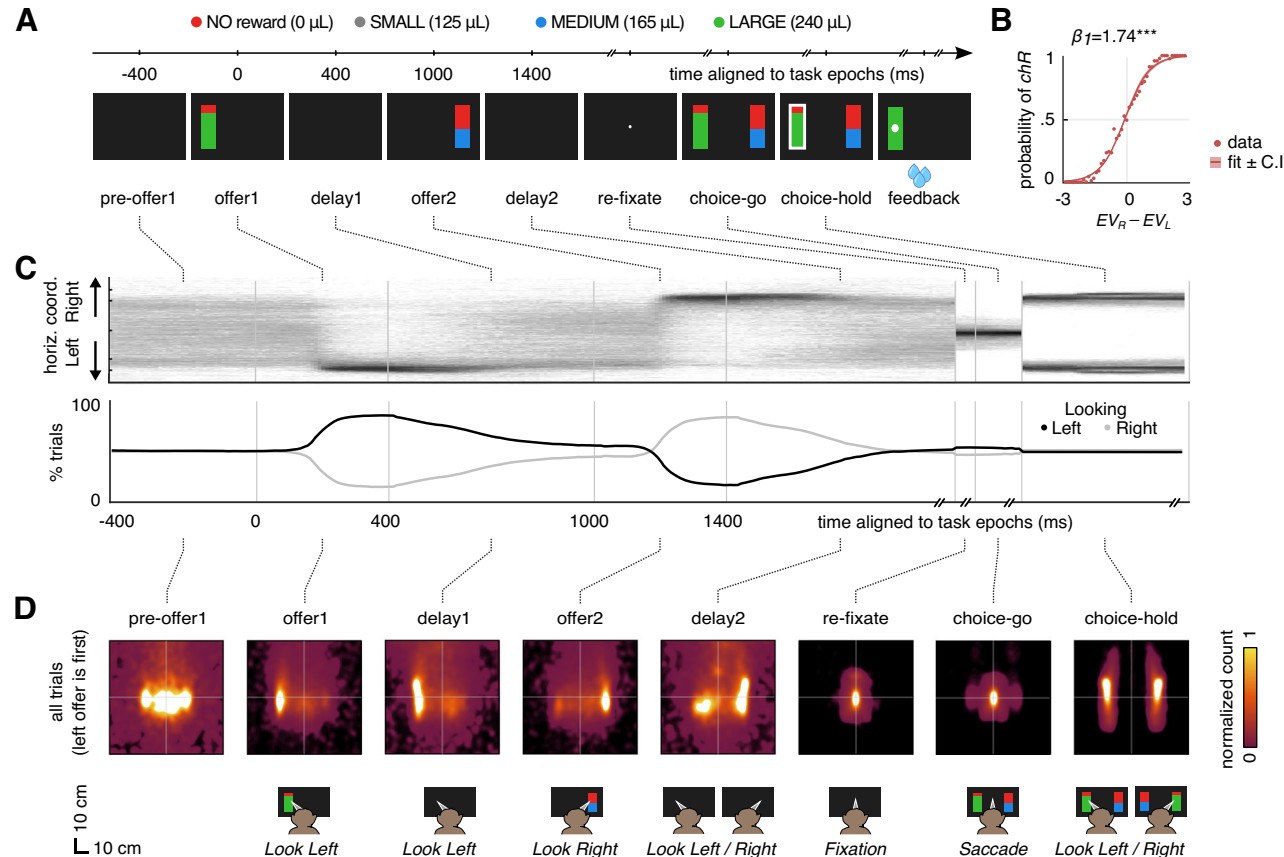

**Fig. 1 | Gambling task and gaze behavior. A** Timeline of the two-alternative sequential gambling task for a sample trial. Reward offers are cued by sequential presentation of vertical bar stimuli at opposite screen sides (*offer 1 / offer 2*) for 400 ms, interleaved with blank screen delay times (*delay 1 / delay 2*) for 600 ms. Stimuli colors either cue to safe (always achieved, reward probability = 1) but small reward (gray), or to a probabilistic reward with medium (blue) or large (green) magnitude. Following fixation to the center of the screen (*re-fixate*), choice is performed after *choice-go* cue by performing a saccade to the target side and holding the gaze for at least 200 ms (*choice-hold*). The trial concludes by providing visual feedback and reward or no reward (*feedback*). Reward offers order (first presented left or right) were pseudo-randomized. For analysis we mirrored choices and eye data along the midline in trials that started with the right offer, so that all trials 'started' with the first offer located on the 'left' screen side. **B** The probability of choosing the right offer ($p_{chR}$) increases with the difference between right and left expected values. Solid line shows logistic fit $logit(p_{chR}) = \beta_0 + \beta_1(EV_R - EV_L)$; shaded areas show ±95% Confidence Interval (C.I.), two-tailed *F*-Statistics test, $p = 3.3 \cdot 10^{-307}$. Data points for each trial are averaged in bins to facilitate visualization ($n = 5971$). **C** Top: distribution of horizontal eye position throughout the trial (spatial bin size is 0.14 cm, time bin size 1 ms); bottom: fraction (%) of trials separated by looking side (left: average horizontal eye position <0; right: average >0) in 10 ms bins during task execution. **D** Gaze distribution over the visual display throughout the trial. Importantly, during the second delay epoch, subjects split gazing to both sizes of the screen, even when the offers are invisible. Sketched icons at the bottom show the most common gaze patterns. **A**−**D** Data includes 5971 trials with valid behavioral choice report (2463 for subject 1; 3328 for subject 2, Supplementary Tables ST1 and ST2 for exact number of trials). Significance: ***$p$ < 0.001.

time looking at the left side of screen ($f_R$<0.25) (Methods "Probability of choice as a function of offer expected values and looking times"). As shown above (Fig. 1B), the probability of choosing the right offer increases with the expected value difference $EV_R - EV_L$, but it is also modulated by which side is mostly looked at (mostly right $f_R$>0.75, mostly left $f_R$<0.25) in all task epochs (Fig. 2A). A significant shift of the curves is observed in all task epochs, being most prominent during the *delay 2* epoch, apart from the choice epoch. These results show that, after controlling for expected value differences, looking time during both look-and-something and look-at-nothing is predictive of the choice that subjects will make at the end of the trial. In a second analysis, we find that the probability of choosing the right offer increases as a function of the difference of the time looking on the right minus the time looking on the left side, $t_R - t_L$, in all epochs (Fig. 2B, all trials). The same monotonic increase is observed when dividing trials by the expected offers: higher for left offer ($EV_L$>$EV_R$), and higher for right offer ($EV_R$>$EV_L$), but vertically shifted (Fig. 2B; Methods "Probability of choice as a function of offer expected values and looking times"; Supplementary Table ST5).

To predict choice on a trial-by-trial basis, we used a logistic regression model that included as regressors the values of the offers, their magnitude, their risk, the location of the first offer, and the fraction of time that the subject looks at the right side in each epoch (Fig. 2C; Supplementary Fig. S3A for each subject; Methods "Probability of choice as a function of offer expected values and looking times"). Adding the expected value of the offer, their magnitude, and their risk (variance) is important to control for factors known to determine choices and that could affect gaze position[46]. Further, a model including these regressors provided the best choice prediction accuracy (Supplementary Fig. S4). In the *choice-hold* epoch, the strongest regressor is the fraction of looking time, due to the causal role of visual fixation in reporting the choice by task construction. Confirming the previous analyses, in earlier epochs (*offer 1* to *re-fixate*), the fraction of looking at the right side has a small but significant effect on the final choice. This effect is not only observed during the offer epochs, but also in the delay epochs, when no stimulus is displayed. Thus, in the absence of stimuli, the subjects have an increased probability of choosing the right offer if they look at the right empty side of

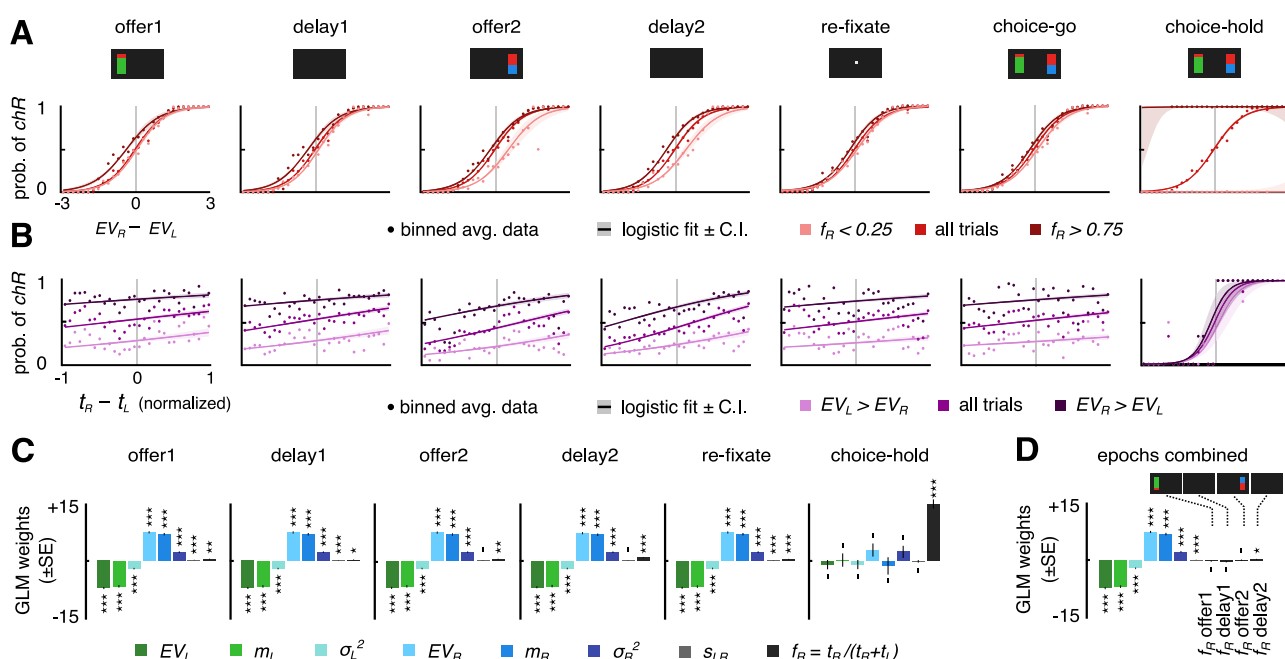

**Fig. 2 | The fraction of time spent looking at either screen side predicts choice, even after controlling for expected value, magnitude, and risk of the offers.**
**A** Probability of choosing the right offer as a function of difference in expected values of the two offers $EV_R - EV_L$. Solid lines are logistic regression fits
$\mathrm{logit}(p_{chR}) = \beta_0 + \beta_1(EV_R - EV_L)$ of choice, shaded areas are ±95% C.I. Data points for each trial are averaged in bins to facilitate visualization ($n = 5971$). The analysis is applied in each epoch for all trials where subjects mostly look to the left ($f_R = t_R/(t_R + t_L) < 0.25$) or to the right ($f_R > 0.75$) screen side. Significance assessment in Supplementary Table ST4. **B** Probability of choosing the right offer as a function of the difference in the right and left looking times $t_R - t_L$ normalized to the epoch duration $t_R + t_L$. Solid lines are logistic regression fits $\mathrm{logit}(p_{chR}) = \beta_0 + \beta_1(t_R - t_L)$, shaded areas are ±95% C.I. Data points for each trial are averaged in bins ($n = 5971$). The analysis is applied for trials with the higher $EV$ on the left and for trials with the higher $EV$ on the right. Significance assessment in Supplementary Table ST5.

**C** Logistic model of behavioral choice. Regression weights (least squares estimate ± SE, $n = 5971$) include $EV_L, EV_R$, offer magnitudes $m_L, m_R$, variance (risk) $\sigma_L^2, \sigma_R^2$, order of presentation ($s_{LR} = +1$ if first offer is on the left; $-1$ if it is on the right), and fraction of time looking at right screen side $f_R$. Significance of weights is assessed via $F$-Statistics. **D** Logistic model of behavioral choice, including all regression weights (least squares estimate ± SE, $n = 5971$) in **C** and the fractions of right-looking time $f_R$ for most relevant task epochs (*offer 1, delay 1, offer 2, delay 2*). Significance assessed as in **C**. **A–D** Data include $n = 5971$ trials with valid behavioral choice report ($n = 2463$ for subject 1; $n = 3328$ for subject 2, Supplementary Tables ST1 and ST2 for exact number of trials). Pooling across trials is made with reference to the first offer side: data in trials with first offer on the right side are horizontally mirrored. **C–D** Significance assessment in table ST6. All $p$-values are FDR corrected[47]. Significance levels: *$p < 0.05$, **$p < 0.01$, ***$p < 0.001$.

the screen for a longer duration, and the opposite is found when looking more to the left empty side, even after controlling for the expected values, reward magnitudes, risks of each offer and order of presentation. The results are reproduced when using for the analysis the fraction of time that subjects spend looking over the physical location of the offers (Supplementary Fig. S3B). We also asked whether the gaze patterns during each of the delay epochs can account by themselves for some variability in choice even after accounting for gaze during the stimulus epochs. We therefore used a new logistic regression model where the fractions of looking at the right side for all epochs prior to choice are included as regressors (Fig. 2D; *offer 1 / 2, delay 1 / 2*). This regression again shows that the fraction of time looking at the right during the *delay 2* epoch has a significant effect on choice (Fig. 2D; $p = 0.029$, corrected for False Discovery Rate (FDR) via Benjamini & Hochberg[47] method).

### Look-at-something and look-at-nothing value encoding
While look-at-nothing gazing suggests the existence of an evaluation process that persists after stimulus disappearance from the visual field and contributes to the choice, further support for this hypothesis would require showing that neural activity indeed reflects reevaluation of the offers. We recorded responses from $n = 248$ neurons ($n = 163$ for subject 1, $n = 85$ for subject 2, Supplementary Table ST1) in two core reward regions, areas 11 and 13 of orbitofrontal cortex OFC (Fig. 3A)[18,20,22,24,27,29,30]. We first studied the neural encoding of expected value of the offers by analyzing the evoked spiking activity at different task epochs. To do so, we compute the average number of spikes in

sliding time windows of 200 ms, shifted by 10 ms steps, for each trial ($n = 5791$ total trials, $n = 2643$ for Subject 1; $n = 3328$ for Subject 2; Supplementary Tables ST1-ST2 for details). To calculate the fraction of neurons that encode the value of each of the two offers, we used linear regression models where the activity of each neuron was exclusively predicted by the value of either offer (first or second, respectively: left $EV_L$, or right $EV_R$) and computed the fraction of neurons with significant regression weights ("Methods "Neural spiking and the encoding of reward expected value").

By using all the trials (without conditioning on gaze), we find that the fraction of neurons encoding either the first or second expected values is not significantly different from chance in the *pre-offer 1* epoch, as expected, but it is strongly modulated in the following epochs in a dynamic way (Fig. 3B). Specifically, the fraction of neurons that encode the value of the first offer (Fig. 3B, $EV_L$, green line) increases significantly above chance (shaded area) in the *offer 1* epoch after a latency of 150–200 ms, and it remains above chance for almost all *delay 1* time. In contrast, the fraction of neurons that encode the value of the second offer ($EV_R$, blue line) is not significantly above chance during both *offer 1* and *delay 1* epochs, as second offer has not been presented yet. During *offer 2* and *delay 2* epochs the reversed pattern is observed: there is a significant encoding of the value of the second offer (blue line), while the encoding of the first offer value lowers to n.s. level. The absence of encoding of the value of the first offer during the *offer 2* epoch is consistent the observation that in most trials the subjects look at the location of the second offer (Fig. 1C; Supplementary Table ST2), and therefore the encoding of the second offer grows

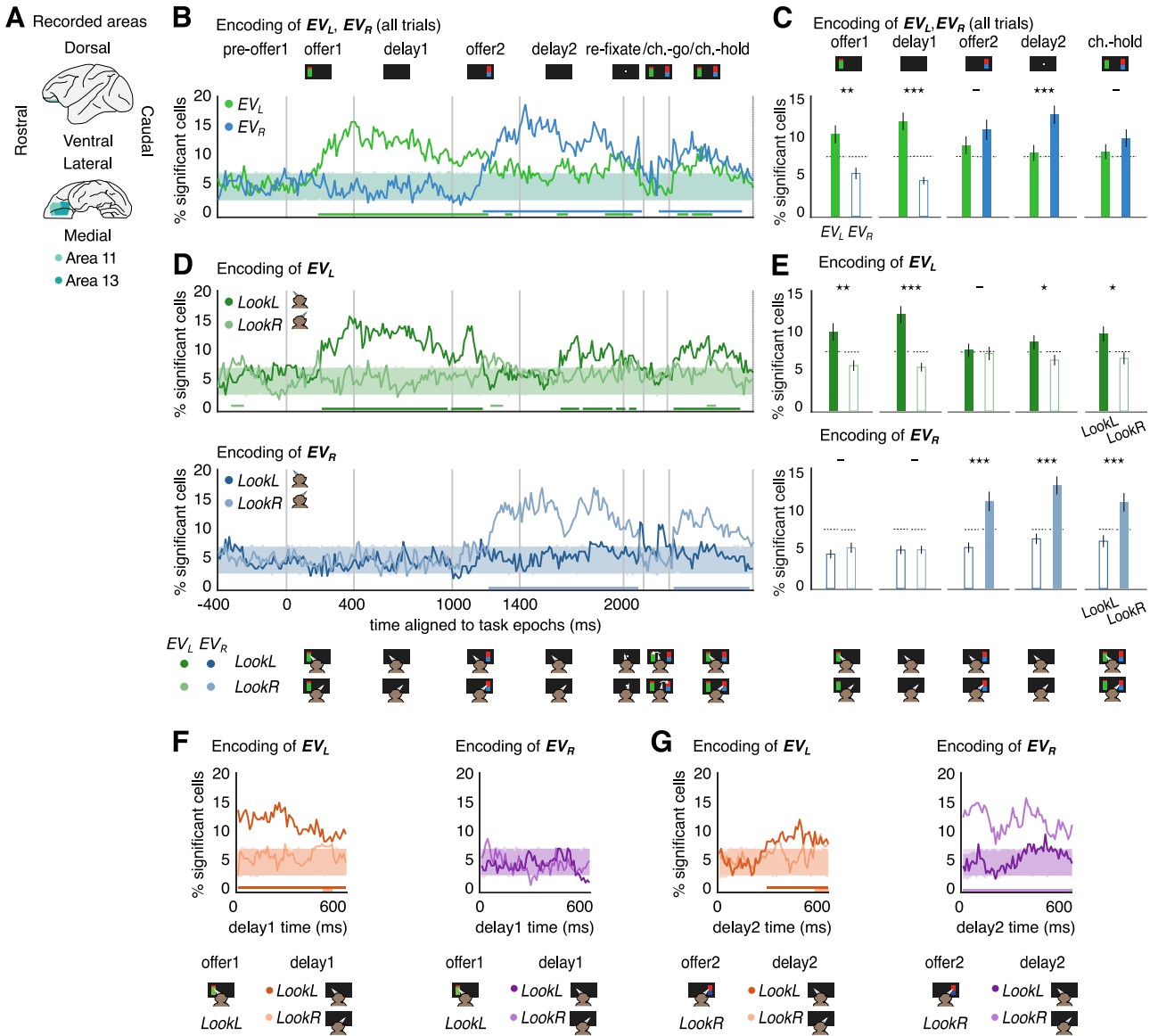

**Fig. 3 | Gaze-dependent encoding of the offer *EVs* and reactivation in OFC.**
**A** Macaque brain areas 11 and 13. **B** Fraction of cells showing significant encoding of $EV_L$ (green) or $EV_R$ (blue) using the linear model $\eta = \beta_0 + \beta_1 EV$ for the spike rate $\eta$ in each cell, time bin and trial. Solid lines show empirical fractions in 10 ms bins; shaded areas (computed independently and overlaid) cover the 5th to 95th percentile of significant fractions for $n = 1000$ bin-wise trial order permutations; bottom lines: consecutive runs of time bins with significant encoding (assessed via run length analysis). **C** Time-averaged fractions of significant cells (mean ± s.e.m.; $n = 248$). Significance assessed as exceeding 95th percentile (dotted lines) of the same values run over $n = 1000$ shuffles of the data (non-significant bars are in white, with colored frame). Results for $EV_L$ and $EV_R$ are compared via two-tailed Wilcoxon signed-rank test, FDR corrected[47]: - n.s., *$p < 0.05$, **$p < 0.01$, ***$p < 0.001$. **D** Top: fraction of significant cells encoding $EV_L$ in trials where subjects *LookL* (average eye position for current bin <0, green) or *LookR* (average eye position >0, light green). Bottom: same as top but for $EV_R$ in *LookL* (blue) and *LookR* trials (light blue). Shaded areas and bottom lines as in **A**. **E** Top: same as **C** but for $EV_L$, comparing *LookL* vs *LookR*. Bottom: same as top but for $EV_R$. Comparisons are run via one-tailed Wilcoxon signed-rank tests, FDR corrected[47], testing that looking left/right yields stronger encoding for ipsilateral *EV*. **F** Encoding of *EVs* at *delay 1* in trials when subjects gazed to left side of the screen during the last 200 ms of *offer 1* presentation. Results show the encoding of $EV_L$ (left) and $EV_R$ (right), comparing *LookL* (orange for $EV_L$, purple for $EV_R$) and *LookR* (light orange for $EV_L$, light purple for $EV_R$). Solid lines, shaded areas, and bottom lines as in **B**. **G** Same as **F**, but at *delay 2*, in trials when subjects inspected the right side of the screen during the last 200 ms of *offer 2*. **C**, **E**. Significance assessment in Supplementary Table ST7.

in alternation to the encoding of the first one. The difference in encoding of the expected values of the first and second offers is also evident when comparing the fractions of neurons encoding each value averaged within task epochs (Fig. 3C).

We next asked whether gaze direction modulates neural encoding, even in delay epochs where the screen is empty. For each 10 ms bin, and in each trial, we separate looking conditions depending on whether subjects mainly look at the right (*LookR*) or left (*LookL*) sides of the screen, based on whether the average eye position in that bin is positive or negative, respectively (Methods "Difference in encoding

$EV_L$ versus $EV_R$ in task epochs, and difference in encoding $EV_L$ or $EV_R$ when looking at different looking sides: *LookL* versus *LookR*"). Since the distribution of eye position is bimodal (Fig. S1D) and the fraction of time looking at the right, $f_R$, computed in 10 ms bins peaks at the extreme values 0 and 1 (Fig. S1E), splitting trials as a function of short-timed average gaze position is a good proxy for time-resolved gaze-centered screen side looking. Running this analysis, we find that there is a significant fraction of cells encoding the first offer during the *offer 1* epoch when subjects look at the left side of the screen (Fig. 3D top, *LookL*, dark green line), but not when they look at the right (light green

line). As expected, there is not significant encoding of the second offer during the *offer 1* epoch (Fig. 3D bottom) regardless of the average gaze position within the epoch (*LookL*, dark blue line; *LookR*, light blue), as the second offer has not been presented yet.

The same pattern of results described so far is reproduced during the *delay 1* epoch (Fig. 3D, *delay 1*), with the relevant novelty that during this epoch the first offer is no longer visible. We find that *delay 1* activity is significantly modulated by offer value (Fig. 3D top, dark green line). This encoding is not simply a passive, gaze-independent memory of the formerly presented first offer: by using only the trials where subjects look at the first offer during *offer 1* (*LookL* during the last 200 ms of *offer 1*), but then look at the opposite side during *delay 1* (*LookR* during the last 200 ms of *delay 1*), we indeed find lack of encoding of the expected value of the first offer (Fig. 3F, left, light orange; 32.26% of trials, Supplementary Table ST2). Thus, looking at the empty side where no previous offer has been presented so far coincides with the decrease of first offer value encoding, suggesting that its encoding persistence or lack thereof depends on gaze (see Supplementary Fig. S5A for other, infrequent, gazing conditions).

During the *offer 2* epoch, we observe the reversed pattern compared to the *offer 1* epoch: significant encoding of the expected value of the second offer is found when subjects look to the right, where the second offer is, (Fig. 3D bottom, *offer 2*, light blue line) but such encoding is absent when the left side is gazed (Fig. 3D bottom, dark blue). Again, during the *delay 2* epoch, where the screen is empty, we find that there is encoding of the second offer value only when the subjects looks at the right (Fig. 3D bottom, *delay 2*, light blue line). Crucially, if subjects look at the left side during the *delay 2* epoch, there is a significant reactivation of the encoding of the first offer value (Fig. 3D top, *delay 2*, dark green line), but encoding of the first offer value is not observed when they look at the right (light green line). Therefore, look-at-nothing gazing during the *delay 2* epoch shows related to the maintenance of neural representation of the most recently presented (second) offer value when looking at its former location (*LookR*; Fig. 3D bottom, light blue line), and to the reactivation of the first offer value if looking at the first offer presentation side (*LookL*; Fig. 3D top, dark green line).

Reactivation of the first offer value during the *delay 2* epoch cannot be accounted for by a pure memory trace of the last seen offer in trials where subjects simply have not looked at the second offer (which occurs in 21.2% of the trials, Supplementary Table ST2): when the second offer is gazed during the *offer 2* epoch, but then gaze is diverted to the opposite side during the *delay 2* epoch (36.48% of the trials, Supplementary Table ST2), we find a significant reactivation of the first offer value (Fig. 3G, left, dark orange line). As control, we observe the maintenance of the encoding of the value of the second offer when the subjects keep looking at the right side of the empty screen in the *delay 2* epoch after looking at the second offer in the *offer 2* epoch (Fig. 3G, right; 43.32% of the trials, Supplementary Table ST2; see Supplementary Fig. S5B for other gazing conditions). Note that we refer to this case as encoding maintenance in *delay 2*, not reactivation, as here subjects keep looking at the same location of the immediately presented offer. Moreover, there is no reactivation of the left offer value during the *delay 2* epoch if subjects do not previously look to the left during *offer 1* nor during *delay 1*, because they did not visually sample the first offer, implying that they did not encode the first offer value during earlier epochs (Supplementary Fig. S6A, C, E).

The result that during the *delay 2* epoch there is reactivation of the first offer value only when looking back to the empty left side of the screen and not when looking at the right cannot be explained by the difference number of trials available in each condition, as reactivation is observed precisely when there is less number of trials (that is, subjects typically tend to look at the right side during *delay 2*; Fig. 2C, Supplementary Table ST2). Further, equivalent results to all those described above (see Fig. 3) are obtained across most epochs,

including *delay 2*, after subsampling trials to match the number of trials per looking condition (*LookL* and *LookR*), even if doing so the number of trials, and thus the statistical power of the analysis, is reduced (Supplementary Fig. S7; Methods "Neural spiking and the encoding of reward expected value"). Consistent with the above (Fig. 3F–G), repeating the same subsampling analysis, we observe reactivation only when looking at the left, and not at the right, during the *delay 2* epoch after looking at the second offer in the *offer 2* epoch (Supplementary Fig. S8).

The results obtained so far using the time-resolved analysis were confirmed when considering average fractions of significant cells across bins in each epoch (Fig. 3E; Methods "Difference in encoding $EV_L$ versus $EV_R$ in task epochs, and difference in encoding $EV_L$ or $EV_R$ when looking at different looking sides: *LookL* versus *LookR*"; shown also after balancing number of trials in Supplementary Fig. S7F). We also confirmed that results in Fig. 3B, D hold for each individual subject (Supplementary Fig. S9), using window sizes of 150 ms and 250 ms instead of 200 ms (Supplementary Fig. S10), normalizing activity by subtraction of the *pre-offer 1* activity (Supplementary Fig. S11, Methods "Neural spiking and the encoding of reward expected value"), taking the coefficient of determination $R^2$ of the linear fits as statistics instead of the fraction of cells (Supplementary Fig. S12A, C; Methods "Neural spiking and the encoding of reward expected value"), and using model-free permutation tests instead of *F*-test statistics to test significance (Supplementary Fig. S12B, D; Methods "Neural spiking and the encoding of reward expected value").

The gaze analysis shown above is somehow coarse and does not address subtle differences such as looking at the right at around the center or far to the right, nor whether neural responses are time-locked to gaze shifts from one side to another of the screen. To address these questions, we first run an analysis by removing trials when gaze covered the central portion of the screen (±4 cm, ±9.5 cm from midline; Supplementary Fig. S13; Methods "Difference in encoding $EV_L$ versus $EV_R$ in task epochs, and difference in encoding $EV_L$ or $EV_R$ when looking at different looking sides: *LookL* versus *LookR*") and replicated the same results previously found (compare with Fig. 3C, E).

In addition, we performed an analysis of neural responses time-locked to the first saccade that crossed screen midline during the delay epochs (Fig. 4), from the right to the left screen side, or vice versa. This analysis is subject to the large variability in timing and patterns of gaze shifts due to the free gazing experimental setup (Supplementary Fig. S1C; Methods "Analyses of eye position and gaze shifts"), yet the results obtained corroborate our previous findings (fractions of trials used in Fig. 4B, C described below are reported in Supplementary Fig. S15A–D). Midline-crossing gaze shifts during delay epochs were time-locked to their initiation, and could be transient, i.e., could entail post-shift midline crossings (Fig. 4A top). We find that reactivation of the left offer value during the *delay 2* epoch occurs soon after the gaze is first shifted from right to left screen sides (Fig. 4C, dark orange), but not when gaze shifts from left to right (light orange). This result confirms our previous reactivation result based on time-resolved average gaze (Fig. 3G). Furthermore, encoding of both offer values lowers to n.s. level whenever gaze is shifted to the opposite screen sides imminently after the respective offer presentation (*delay 1* for $EV_L$, Fig. 4B, *shift R*, light orange; *delay 2* for $EV_R$, Fig. 4C, *shift L*, dark purple). As expected, left offer value encoding is weak before and after a gaze shift to the left side during *delay 1* epoch (Fig. 4B, dark orange) since shifting to left implies that gaze is first directed to the right screen side (at least) at the end of the *offer 1* epoch, and therefore the left offer is necessarily poorly encoded. Similarly, for the right offer value there is weak encoding before and after a gaze shift to the right screen side during *delay 2* epoch (Fig. 4C, light purple): shifting right implies having directed gaze to left side (at least) at the end of *offer 2*, hence no or weak encoding of the right offer is expected. Lastly, the right offer value is not encoded in any case during *delay 1* (Fig. 4B, light and dark

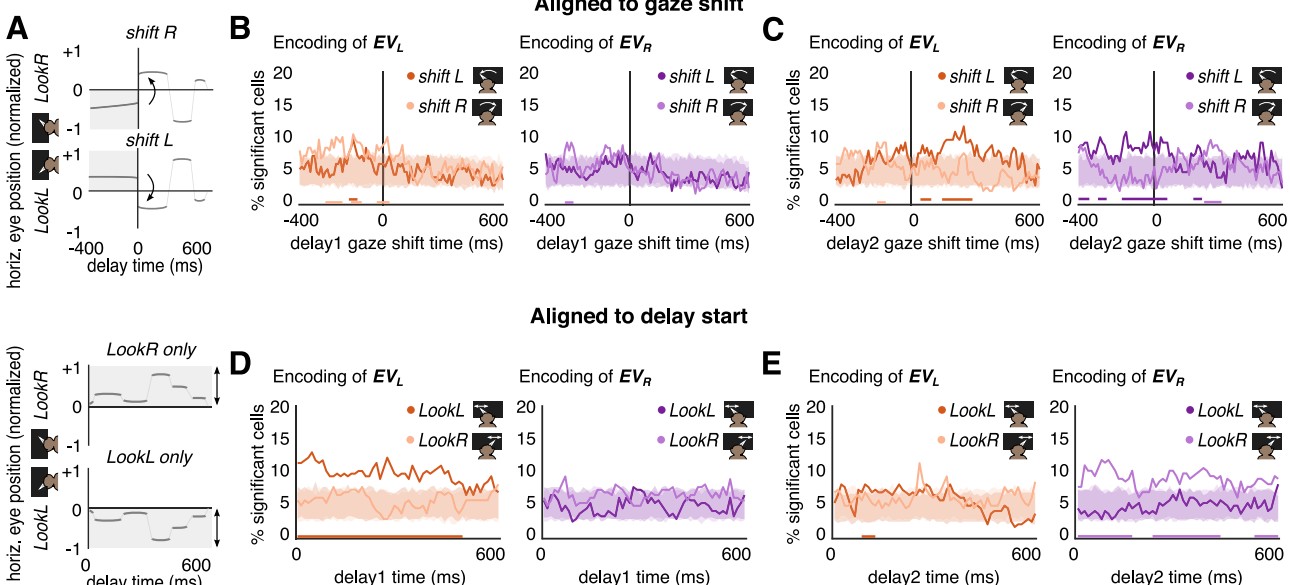

**Fig. 4 | Gaze centered reactivation of offer value encoding for gaze-shift trials and maintenance of offer value encoding for non-shift trials. A** Sketched illustration of timepoints selection and time-locking criteria used in **B–E**. Top: data are time-locked to the start of the earliest delay epoch gaze shift (involving a midline gaze crossing from right to left screen sides –*shift L*–, or vice versa –*shift R*) and truncated at the end of delay epoch, allowing return on previous screen side; bottom: data are time-locked to delay onset and do not include gaze shifts. **B** Encoding of offer values time-locked to a gaze shift during *delay 1*. We exclusively find significant, albeit weak, encoding of $EV_L$, only before the gaze is shifted to right (left panel, *shift R*, light orange). **C** Encoding of offer values time-locked to a gaze shift during *delay 2*. Notably, left shows that in *delay 2* there is encoding of $EV_L$ (reactivation) after gaze is shifted from right to left (*shift L*, dark orange), in

accordance with *LookL* in Fig. 3G (dark orange). In right, we find encoding of $EV_R$ before gaze is shifted to left (*shift L*, dark purple), in accordance with *LookR* in Fig. 3G (light purple). **D** Encoding of offer values in trials where gaze falls within either screen side throughout the *delay 1* epoch. In left, we report encoding of $EV_L$ only for *LookL* (dark orange) in agreement with *LookL* in Fig. 3F (dark orange). **E** Same as **D** but for *delay 2*. In right, we find encoding of $EV_R$ for the *LookR* condition, in agreement with *LookR* in Fig. 3G (light purple). **B–E** The fraction of trials available in the respective configurations are reported in Supplementary Fig. S15A–D. Note that the fraction of trials in **D** *LookR* and **E** *LookL* is much smaller compared to other configurations due to heavy constraining the gaze to stay on either screen side.

purple) since the right (second) offer has not been displayed yet, and the left offer value is not encoded before or after a gaze shift to the right side in the *delay 2* epoch (Fig. 4C, light orange) as the first offer was on the opposite side.

To make sure that our analysis includes only trials where subjects look at the offer during the offer epochs for the full stimulus duration before gaze shifts, we repeated the above analysis by fixing pre-shift gaze (200 ms prior to shift initiation) over the left offer side in *offer 1* epoch, and the right offer side in *offer 2*. Using the last 200 ms during offer epochs allows keeping a substantial number of trials for the analysis, and roughly corresponds to the time window of the offer epochs that excludes the first 200 ms needed for subjects to look at the offer (see Fig. 1C). Our results are corroborated under this more restrictive analysis (Supplementary Fig. S15E–H).

Finally, by analyzing trials where no midline-crossing gaze shifts occurred, we find encoding of the respective left / right offer value during the *delay 1* (Fig. 4D, dark orange) / *delay 2* epochs (Fig. 4E, light purple) whenever subjects kept their gaze on the screen side where most recent offer was presented. This indicates that maintenance of the respective offer value encoding is facilitated by holding gaze side fixation after the offer presentation. As expected, no significant encoding of offer values is observed in all other non-shift cases (Fig. 4D, light orange, purple; Fig. 4E, orange, dark purple).

By performing the same analysis as in Fig. 4B, C on gaze shifts occurring during offer epochs, we confirm that results are consistent with a gaze-centered encoding of offer values, with encoding of the gazed offer after a gaze shift to it, but not before (Supplementary Fig. S15I–L).

## Reactivation of value encoding during look-at-nothing correlates with choice and suggests reevaluation

A final question is whether activity in OFC during look-at-nothing simply reflects a reactivation of a spatial association of the formerly presented offer, or whether there is reevaluation during this reactivation, implying additional accumulation of information during this process. To answer this question, we performed various analyses. First, we used a linear model to predict the activity of OFC neurons using as regressors the subjective value (*SV*) of the offers and the choice of the subject at the end of the trial (Methods "Neural encoding of offer subjective value and choice"). In this section we always considered looking conditions based on average gaze position (*LookL*: average >0; *LookR*: average <0). We argue that if look-at-nothing gazing reflects reevaluation of the offers, we should see a significant effect of choice in the delay epochs that cannot be accounted for by the subjective values of the offers alone (which, in turn take into account expected value, magnitude, and risk). Indeed, by performing a gaze-independent analysis, we find that a significant fraction of cells carries choice signals through the latest task epochs and that this becomes prominent during the choice epoch (Fig. 5A, B, gray). Therefore, fluctuations of neural activity are accounted for by choices beyond what would be predicted by only using the subjective values of the offers in the linear model of firing rate (Methods "Neural encoding of offer subjective value and choice"), suggesting a role of the OFC activity in choice encoding, in accordance with previous literature[28,48,49]. Similar patterns are revealed when conditioning on gaze being directed towards presented offer sides or to their empty locations in the *delay 2* epoch (Fig. 5C, D), also suggesting that evaluation and reevaluation of the offers continues through latest task epochs.

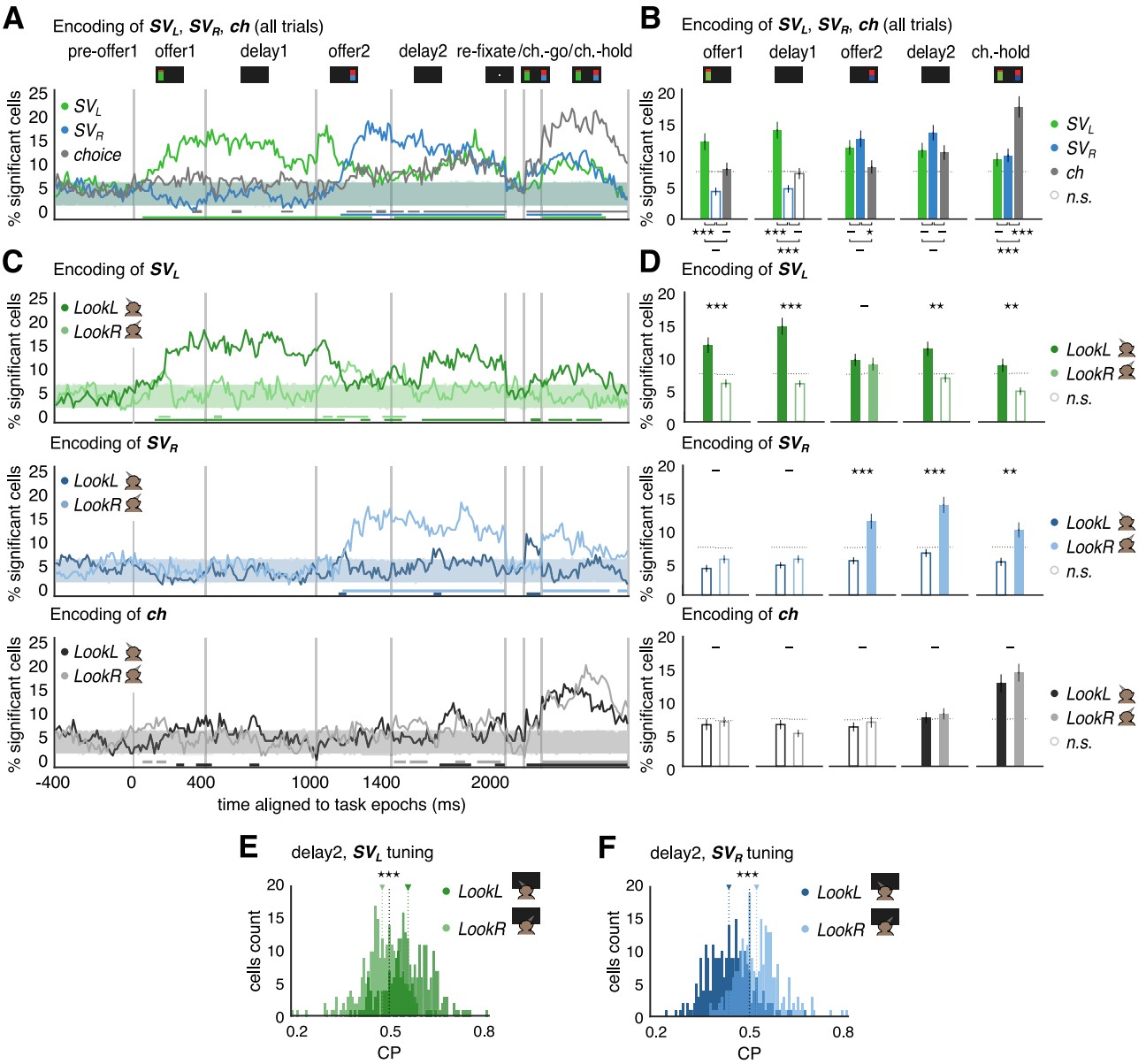

**Fig. 5 | Choice-related activity supports the reactivation of offer value encoding for reevaluation. A** Fraction of significant cells encoding subjective values $(SV_L, SV_R)$ and choice ($ch = 1$ right offer is chosen, $-1$ otherwise), fit to a linear model of the spike rate $\eta$, as $\eta = \beta_0 + \beta_1 SV_R + \beta_2 SV_L + \beta_3 ch$ using all trials. Subjective values ($SV_L = w_{1L}EV_L + w_{2L}m_L + w_{3L}\sigma_L^2$ and $SV_R = w_{1R}EV_R + w_{2R}m_R + w_{3R}\sigma_R^2$) are computed in each session via logistic regression of the choice: $\text{logit}(p_{chR}) = w_0 - w_{1L}EV_L - w_{2L}m_L - w_{3L}\sigma_L^2 + w_{1R}EV_R + w_{2R}m_R + w_{3R}\sigma_R^2$ (Supplementary Table ST3). Solid lines, shaded areas and bottom lines are computed as in Fig. 3B. **B** Time-averaged fraction of cells (mean ± s.e.m.; $n = 248$) showing significant encoding of offer $SV$s and *choice*, computed as in Fig. 3D. The time-averaged fraction of time bins with significant encoding for $SV_L$, $SV_R$ and $ch$ are compared via non-parametric tests (Wilcoxon signed rank, two-tailed) corrected at FDR = 0.05[47]: - n.s., *$p < 0.05$, **$p < 0.01$, ***$p < 0.001$. **C** Top: same as **A**, but for the encoding of $SV_L$, comparing trials where subjects mostly *LookL* vs *LookR* in each time bin; middle:

same as **A**, focusing on the encoding of $SV_R$, comparing *LookL* vs *LookR*; bottom: same as **A**, but for the encoding of *choice*, comparing *LookL* vs *LookR*. **D** Top: same as **B** but for $SV_L$ and comparing *LookL* vs *LookR*; middle: same as top, but for $SV_R$; bottom: same as top, but for *choice*. Time-averaged fraction of time bins with significant encoding for *LookL* and *LookR* are compared via non-parametric tests (Wilcoxon signed rank, one-tailed for $SV$s, testing that ipsilateral look yields larger fraction of encoding cells; two-tailed for *choice*, testing for non-zero mean in their difference) corrected at FDR = 0.05[47]: - n.s., *$p < 0.05$, **$p < 0.01$, ***$p < 0.001$. **E** Analysis of $SV$ regression residuals. Choice Probability (CP) tuned to $SV_L$ sign and comparison of *LookL* vs *LookR* (median CP: *LookL* = 0.56; *LookR* = 0.48; one-tailed Wilcoxon signed rank test for *LookL* > *LookR*, *** $p = 7.83 \cdot 10^{-37}$). **F** Same as **E**, but showing CP tuned to $SV_R$ sign in *LookR* and *LookL*. (Left: median CP: *LookL* = 0.44; *LookR* = 0.52; one-tailed Wilcoxon signed rank test for *LookR* > *LookL*, *** $p = 6.14 \cdot 10^{-40}$). **B, D** Significance assessment in Supplementary Table ST8.

Second, if reactivation of the first offer encoding reflects reevaluation, we expect to see that the same neurons participate in both the encoding of the first (left) offer value and in its reactivation. Thus, we considered the encoding of both subjective values (using the simplified model $\eta = \beta_0' + \beta_1' SV_L + \beta_2' SV_R$, Methods "Neural encoding of offer subjective value and choice") in combination with gaze. We observe that the epoch-wise encoding weights for the value of an offer when it is looked at correlate across neurons with the weights for the value of

the same offer when looking at its empty location during *delay 2* (*offer 1*: $\rho = 0.55$, *offer 2*: $\rho = 0.22$; $p < 0.001$ in both cases; weights are computed using the last 200 ms of each relevant epoch). This confirms that overlapping neural populations are responsible for the encoding of both values during stimulus presentation and value reactivation in empty screen epochs. However, the same weights are negatively, or not correlated when subjects look to the opposite location during *delay 2* (*offer 1*: $\rho = -0.21$, $p < 0.001$; *offer 2*: $\rho = -0.12$, n.s.). In accord

with previous work[28], we find significant Pearson's correlation between the encoding weights of the first offer during the *offer 1* epoch and the weights of the second offer during the *offer 2* epoch ($\rho = 0.84$, $p < 0.001$), suggesting that there are overlapping neural populations that asynchronously encode offers as they are presented. On top of this, we found that 1/3 of cells asynchronously encoding the two offers at their respective presentation times, also simultaneously encode of the two offers at *offer 2*. These results suggest that neurons tend to encode one offer at a time, and that overlapping populations are responsible for offer value encoding across all epochs, including reactivation.

To more directly test for reevaluation of the first offer during its reactivation, we show that fluctuations of activity during the reactivation in the *delay 2* epoch correlate with the choice in the expected direction[24,50,51]. For instance, we ask whether a positive activity fluctuation during reactivation of a neuron with positive tuning to $SV_L$ is correlated with an increased probability of choosing the left offer, as an increase of its activity can be interpreted as an improved estimate of the subjective value of the first offer. To take into account all possible cases, we first consider the relationship between choice and fluctuations of activity using the residuals of the linear model of the spike rate $\eta = \beta'_0 + \beta'_1 SV_L + \beta'_2 SV_R$ during *delay 2*. Based on the sign of linear regression weights, we identify *preferred* choice and *non-preferred* choice data pools for each neuron (Methods "Analysis of regression residuals: preferred vs non-preferred choice"; for analyses with tuning to $SV_L$, *preferred* choice includes data with positive $\beta'_1$ and choice = left, or negative $\beta'_1$ and choice = right; *non-preferred* choice includes data with positive $\beta'_1$ and choice = right, or negative $\beta'_1$ and choice = left; for analyses with tuning to $SV_R$ choice preference is set by $\beta'_2$ sign instead of $\beta'_1$ and by swapping choice = left/right with right/left). Next to this, for each trial and in each cell, we consider the time-averaged residuals $z = \eta - \beta'_0 - \beta'_1 SV_L - \beta'_2 SV_R$ (Methods "Analysis of regression residuals: preferred vs non-preferred choice"). To test our prediction that the sign of the residuals (fluctuations) during reactivation correlates with choice depending on the subjective value tuning on the cell, we use trial-based Choice Probability (CP) analyses (Methods "Analysis of regression residuals: Choice Probability"[50]). For each cell, we define CP as the fraction of trials where the time-averaged regression residuals are larger for *preferred* than for *non-preferred* choices (Methods "Analysis of regression residuals: Choice Probability"). We applied this analysis separately for tuning to $SV_L$ (Fig. 5E) or to $SV_R$ (Fig. 5F), and in each case also separately using *LookR* or *LookL* trials during the *delay 2* epoch. During reactivation (i.e., in *LookL* trials) we find CP significantly larger than 0.5 using neural tuning for $SV_L$ (Fig. 5E; CP > 0.5 in n = 205 cells, 83% of the total; CP median = 0.56; $p = 1.44 \cdot 10^{-41}$, one-tailed Wilcoxon signed-rank test) and significantly lower than 0.5 using neural tuning for $SV_R$ (Fig. 5F; CP < 0.5 in n = 212 cells, 85% of the total; CP median = 0.44; $p = 1.08 \cdot 10^{-48}$, one-tailed Wilcoxon signed-rank test). This result implies that positive activity fluctuations during reactivation (*LookL*) of cells with positive tuning to $SV_L$ coincide with larger probability of choosing the left offer; but at the same time, negative activity fluctuations during reactivation of cells with positive tuning to $SV_R$ also coincide with a higher probability of choosing the left offer. The effect of gazing at ipsilateral screen location is more prominent when looking at the left than at the right sides (Fig. E-F), though the difference in CP is significant when comparing residuals for firing tuning to $SV_L$ vs $SV_R$ both in *LookL* (one-tailed Wilcoxon signed-rank $SV_L > SV_R$; $p = 7.64 \cdot 10^{-56}$), and in *LookR* (one-tailed Wilcoxon signed-rank $SV_R > SV_L$; $p = 2.57 \cdot 10^{-15}$). Taken together, these results suggest that reactivation during the empty screen in *delay 2* serves to offer reevaluation.

We checked that running linear spike count fits and CP characterization analyses by only including trials where the two offers had low *EV* difference (in the range [−1, +1]) leads comparable values of CP (CP distributions not shown; for $SV_L$ *LookL*, median CP = 0.56; $SV_L$ *LookR*, median CP = 0.48; $SV_R$ *LookL*, median CP = 0.44; $SV_R$ *LookR*, median CP = 0.52). All results for CP analyses can be also shown by tracing *cell-wise* Receiver Operating Characteristics (ROC) curves (CPs tuned to $SV_L$, $SV_R$ for both *LookL*, *LookR*; Supplementary Fig. S14A; Methods "Analysis of regression residuals: cell-wise and cells-combined ROC"), instead of estimating the CPs using the fraction of times that fluctuations in *preferred* choices were larger than in *non-preferred* ones, leading to the same numerical results.

Lastly, we performed an independent analysis, based on time- and trial-averaged residuals. Although this analysis is less powered, we find that trial-average residuals are significantly larger for the *preferred* choice than for the *non-preferred* choice tuned to $SV_L$ during reactivation in *LookL* trials (Supplementary Fig. S14B, top left, $p = 0.0155$, one-tailed, paired Wilcoxon signed rank test), but not in *LookR* trials (Supplementary Fig. S14B, top right, $p = 0.405$, one-tailed, paired Wilcoxon signed rank test), consistent with previous results.

## Discussion

Using an economic choice task with sequential offer presentation and unconstrained gaze, we show that gaze reactivates neural responses that correlate with the value of previously shown offers. After accounting for offer value, reward magnitude, and risk, residual variability in choice is explained by two additional factors: how much time the subjects were looking at the offers when presented (a look-at-something bias), and how much subjects looked at the empty screen sides where offers were formerly presented (a look-at-nothing bias). Therefore, gaze biases choice beyond the subjective value of the visual stimuli. Based on single-neuron activity in OFC, we find that (1) fixating an offer triggers the neural encoding of its value, (2) gazing to an empty region where no stimulus has been presented coincides with a reduction in value encoding strength, and (3) gazing to an empty region where a stimulus was formerly presented reactivates encoding of its value. Furthermore, the size and sign of this reactivation predict choice, and overlapping neural populations are involved in both sensory encoding of value and its reactivation during look-at-nothing. These results argue for an active role of the gaze in decision-making by facilitating memory reactivation and reevaluation.

Our study builds on a large body of research showing that gaze is an integral part of decision-making[1-4,6,10]. Previous works that kept stimuli in vision[38] also observed a reduction in value encoding as soon as the gaze moved away from the stimulus, consistent with our results. Another recent study about the encoding of the values of simultaneously presented offers show decreasing, yet substantial, encoding of both offers after gaze switches between them[51]. Our work shows the perhaps more intriguing result that looking at empty regions where an offer previously appeared triggers the reactivation of its associated value. This reactivation is not a simple memory trace of the most recently gazed stimulus, as reactivation of the first offer occurs even when subjects look to a second offer in the meantime, eventually playing the role of a distractor. Further, reactivation does not purely include a retrospective (memory) component of a previously observed offer; it also has a prospective component since fluctuations of activity are predictive of the upcoming choice (Fig. 5E, F). Our results are also incompatible with the possibility that both gaze and reevaluation purely reflect choice commitment[52] or motor preparation for the choice report[53-55]: reactivation is found to correlate with the value of a previously seen offer (see Figs. 3D, 4C and 5C), and fluctuations of activity predict choice at fixed looking conditions (see Fig. 5E, F). Nevertheless, a motor component participating in addition to the reactivation and reevaluation components cannot be ruled out based on our data. Finally, although we cannot argue in favor of a causal interplay between gaze and reevaluation or choice, our results suggest that gaze-dependent reactivation might be an integral part of the reevaluation of offers that are either visible or invisible, stored in working memory. We have shown that shifting gaze from one side to the

other side of the screen, even when the target side is empty, coincides with a decrease in the strength of encoding in OFC for the previously seen offer value. This might imply that OFC does not hold a memory of all previously considered offers, but only of the most recently sampled side. These results are consistent with recent research showing that OFC mainly encodes the value of fixated offers, while other areas, such as the anterior cingulate cortex (ACC), might have a more detailed map of the relevant offers in the visual field and possibly perform offer comparison[38]. Ours and previous results suggest that OFC is unique in holding and evaluating the most recent offer, reactivating it when necessary for further evaluation. This view contrasts with the simultaneous encoding of offers previously reported in OFC in tasks where the gaze remained fixed on a central fixation point[34]. An important difference between this study and ours is the lack of eye movements in the former. As stimuli are passively processed, the task of maintaining previously encoded information might be simplified, thus making the encoding of multiple offer values possible. In contrast, in the more naturalistic scenario where gaze is required to shift around offers, it is possible that an association is made between the offer and its location, such that later reactivation of a formerly presented offer requires, or is facilitated by, look-at-nothing gazing. Our results are also more in accord with other work showing that fluctuations of covert attention are translated into alternations between encoding one of the two presented offers as a function of time[26], although this work surprisingly does not find any dependence between gaze and choice[26]. In another study, where cues to reward magnitude and probability of alternative offers are kept invisible and subjects are attentionally driven to perform visual sampling, OFC shows encoding of both stored and visually sampled cues[33], as they are both needed to extract value. Recent work where attention is decoupled from stimulus saliency shows that value encoding of well-known offers in single-neuron OFC is independent of attentional shifts[56]. It is possible that in this study, overtrained stimuli facilitate the encoding of value, and thus gaze and attention can be decoupled. In contrast, in our case, offers are more complex and varied, but also include the requirement to combine two features (color for reward magnitude and height for probability) to form an estimate of their value, and then it is possible that a more robust, gaze-centered working memory mechanism needs to be at place.

Our results are in line with previous theorizations of looking-at-nothing gazing, which has proposed that gaze is used as memory spatial indexing mechanism[57,58], thus. One possibility is that coupling between gaze and evaluation may leverage brain machinery that allows for an efficient spatial representation of visual offers to be used for storage and comparison. Indeed, behavioral studies have proposed that gazing allows for memory indexing in spatial coordinates[58–60], therefore facilitating memory retrieval and decision-making. For instance, Wynn et al.[61], show that reinstating gaze patterns during free viewing facilitates subsequent memory retrieval in cognitively demanding tasks. At the neural level, prefrontal cortex[62] has been associated to memory maintenance through sustained firing in post-presentation delay times, while hippocampus have been shown to participate in memory-based decision formation[63,64]. These circuits might participate in the orchestration of gaze, memory retrieval and decision-making. Among most related, Bone et al.[65], found that the fMRI encoding, and retrieval of sensory visual images had significantly similar gaze patterns and that neural reactivation in occipital cortex correlated with gaze reinstatement. Also, in other conditions, gaze has been shown to re-activate sensory memories. For instance, work in rodents has shown that during rapid eye movement (REM) sleep, eye movements trigger neural activity in the anterodorsal nucleus of the thalamus that reflects internal heading in the virtual world of the sleeping subject, showing a tight coupling between eye and internal variables[66]. Our work in addition shows that the coupling between eye position and internal variables extends to awake primates and might

subserve an important mechanism for memory reactivation during decision-making. Our results extend those of reactivation in the hippocampus for memory reconsolidation[67] and planning[68], but here we show that it can also be a mechanism for reevaluation of information in the cortex to aid visually guided economic decision-making.

All in all, our results generally support the hypothesis of sequential value encoding[13], whereby specific brain areas allocate their neural resources to a single offer being scrutinized at a time, using gaze as a facilitating mechanism for evaluation and reevaluation when offers have been spatially presented. In accord with the sequential hypothesis, recent theoretical work shows that focused evaluation of offers is generally a better strategy than dividing neural resources into several, or many, offers to be processed in parallel[14,16]. The one-at-a-time processing of offers during look-at-something and the specific reactivation of one offer at a time during look-at-nothing thus suggest a syntax of syllable-like neural operations that are executed sequentially in decision-making.

## Methods

### Statistics and reproducibility

Data include 5971 trials correctly performed (2643 from subject 1, 3328 from subject 2) and 248 cells (163 from subject 1, 85 from subject 2; the average number of simultaneously recorded cells per session is $40.75 \pm 16.9$ (mean $\pm$ s.d.) for subject 1, and $21.25 \pm 9.98$ for subject 2). Neuronal units with firing rate <0.6 spikes/s were excluded. Detailed number of trials and cells for each session and for each subject are reported in Supplementary Tables ST1 and ST2. All analyses of neural data were compared with equivalent results over randomized versions of the data.

**Experimental procedures.** All procedures were approved by the University Committee on Animal Resources at the University of Rochester and the University of Minnesota and were designed and conducted in compliance with the Guide for the Care and Use of Animals of the Public Health Service. The animals were handled according to approved institutional animal care and use committee of the University of Rochester and Minnesota. Two adult, male rhesus macaques (*Macaca mulatta*) were trained to perform a two-alternative risky choice task[28,45,69]. Animals were habituated to the experimental facility before training. Recordings covered brain Areas 11 and 13, in orbitofrontal cortex (Fig. 3A). The positioning was guided by magnetic resonance imaging with the aid of a Brainsight system (Rogue Research). Prior to surgery, monkeys received intramuscular tranquilizers (ketamine: 10 mg/kg; midazolam: 0.25 mg/kg; atropine: 0.04 mg/kg), were placed in a stereotaxic instrument (Kopf Instruments) and were maintained in anesthesia via inhalation of isofluorane (1–3%). The craniotomy was made ($\sim$ 2–3 cm$^2$) to place Cilux recording chambers (Crist Instruments) over the areas of interest (as detailed in previous work[69]). Animals received appropriate analgesics and antibiotics after all procedures. In each experimental session, recording chambers were washed and sealed with sterile caps.

**Behavioral task.** The behavioral task consisted in the sequential presentation of two visual stimuli at the two opposite sides of the screen. Each visual stimulus indicated the size of a reward that could be chosen at the end of the trial by saccade to stimuli location (Fig. 1A). The task starts with a first offer presentation (*offer 1*) displayed on the screen for 400 ms, followed by a first blank screen delay time with duration 600 ms (*delay 1*); the same timings are used for a subsequent, second offer presentation (*offer 2*), respectively followed by a second, blank screen delay time (*delay 2*). After the presentation of the two alternative offers, subjects were instructed to reacquire fixation (*re-fixate*), for at least 100 ms, at the center of the screen through the appearance of a central fixation dot. Following fixation, the choice could be indicated after the *choice-go* cue, which consisted of the simultaneous

presentation of both offer stimuli in the same locations where they were previously shown. Choice was reported by shifting gaze to the preferred offer location and holding fixation on it for 200 ms (*choice-hold*). Fixation breaks during choice-hold periods led to a return to the choice report stage, giving subjects the chance to change their mind, thus to re-start fixation to either offer location for a duration of 200 ms. In trials with a valid choice report, the trial follows with the gamble outcome resolution: reward delivery for a successful outcome and lack thereof for an unsuccessful outcome (see below for the probabilities of these events and the stimuli cues). Subjects were left free to direct their gaze during all task epochs except when instructed to fixate. Trials that took more than 7 seconds were considered inattentive and were not included in the analyses. Successfully rewarded trials were followed by visual *feedback* made of a white circle centered on the chosen offer stimulus.

The sequential presentation of offers in our task design has several important advantages over simultaneous offer presentations. First, if we employed simultaneous presentation of the two offers, the gaze patterns might have been much more variable, and therefore the dynamical analysis of neural firing aligned with gaze could have been more difficult to perform. Second, given our interest in studying the reactivation of first value encoding in the *delay 2* epoch, we needed such epoch to happen enough time after *offer 1*, eventually followed by a second offer (distractor), so that looking back to first offer side and encoding of the first offer value could not be purely explained by sensory encoding, and required gaze-dependent reactivation.

Visual stimuli were presented on a 24″ monitor with a resolution of 1024 pixels × 768 pixels, a physical width and 48.8 cm and a height of 36.6 cm, positioned at 57 cm from the eyes of the subjects. The visual offers consisted of two vertical rectangles 300 pixels tall and 80 pixels wide (14.3 cm height and 3.8 cm width). Their centers were displaced right or left from the center of the screen by 256 pixels (12.2 cm, 1/4 of the total screen width). The presentation sites were randomized so that the first offer could be presented with equal probability on either side of the screen and the second on the opposite side (the final count was that in 50.01% of the trials, the first offer was presented on the left side of the screen). The colors and height of the rectangles respectively indicated the magnitude *m* and probability *p* of the offered liquid reward. Reward magnitudes were pseudo-randomized across trials. Reward probabilities of the medium and large offers were independently drawn from a uniform probability distribution. The magnitude of the reward was indicated by the color of the bottom part of the vertical bar stimuli (gray: small, 125 µL; blue: medium, 165 µL; green: large, 240 µL). In all analyses, reward magnitudes *m* are reported and used in nominal units (1 = small, 2 = medium, 3 = large reward). The height of the bottom part of the bar stimuli indicated the (success) probability *p* of obtaining the liquid reward (with magnitude *m* as indicated by its color) if the offer was chosen, in a way that if the whole bar is of one single color, the probability of the reward is equal to one. Whenever *p* was < 1, the stimuli top fraction was colored in red, with its height indicating the complementary probability $1 - p$ for unsuccessful outcome. We define the expected value of an offer as the product of its probability times its magnitude: $EV = mp$. To consider the trial-to-trial variability of offer value, we also defined the variance $\sigma^2 = mp(1 - p)$, also referred to as offer risk. The definition of variance follows the intuition that risk can be modeled as the variance of *m* independent Bernoulli variables (since *m* is normalized to $m = 1, 2, 3$). We checked that the definition of variance as for a single Bernoulli variable ($\sigma^2 = p(1 - p)$) would not have qualitatively affected the results of our main behavioral and neural analyses. Offers having small rewards were always sure, $p = 1$ (safe option). Reward probabilities for the medium and large magnitude offers were randomly and independently drawn from a uniform distribution in the unit interval, whose presentation was limited in size only to the resolution of screen pixel size. Reward magnitudes (small, medium, or

large) of the first and second offers were randomized across trials so that 1/8 were safe, and for the remainder, 7/8, offers could be either medium or large in equal fractions (yielding a total of: 12.15% small, safe offers; 43.93% medium-sized gamble offers; 43.92% large sized gamble offers). Trials were interleaved with an 800 ms idle inter-trial interval when subjects were not given instructions and the screen was left blank.

Eye position was sampled at 1 kHz by an infrared eye-monitoring camera system (SR Research). Stimuli were controlled by a computer running Matlab (Mathworks) with Psychtoolbox[70] and Eyelink Toolbox[71].

**Neural data recordings.** Animals were accustomed to laboratory conditions and then trained to perform oculomotor tasks for liquid reward. We approached areas 11 and 13 through a standard Cilux recording chamber (Crist Instruments). A small prothesis for holding the head was used. Position was verified by magnetic resonance imaging by aligning white and gray matter scans to standard anatomical atlas with the aid of a Brainsight system (Rogue Research Inc.). Neuroimaging was performed using a 3 T MAGNETOM Trio Tim using 0.5 mm wide voxels. Animals received appropriate analgesics and antibiotics after all procedures. Throughout both behavioral and physiological recording sessions, the chamber was kept sterile with regular antibiotic washes and sealed with sterile caps. Multi-contact V-probes (Frederick Haer & Co., 32 contacts each, 75 µm inter-contact spacing, impedance range 0.8–4.0 MΩ) were lowered in acute sessions using a microdrive (NAN Instruments) until waveforms of between one and three neurons per contact were isolated. Individual action potentials were isolated on a Plexon system (Plexon, Inc.). Following a settling period, all active cells were recorded. Cells were sorted offline manually by trained electrophysiologists; no automated sorting tools were used. Neurons were selected for study solely based on the quality of isolation; they were never preselected based on task-related response properties. All collected neurons for which we managed to obtain at least 300 trials were analyzed; no neurons that surpassed our isolation criteria were excluded from analysis.

### Analyses of behavioral data

In all analyses, we pool trials where the first offer was on the left with trials where the first offer was on the right by mirroring in the latter case behavioral variables and eye tracking data along the vertical axis. Therefore, in all analyses and figures the first offer is anchored on the left presentation site, since we did not find appreciable difference between trials with opposite order of presentation (Supplementary Fig. S1B). For time resolved analyses, time epochs are aligned to task-related event times (e.g., *offer 1* time runs from stimulus onset up to the end of its presentation). For time epochs with random duration (e.g., for *delay 2*, due to trial-to-trial variation in fixation re-acquisition time), we limited the end of task-relevant time epochs to the duration of shortest trial across all data (in both monkeys and sessions), so that all analyzed data points were covered by all trials.

**Behavioral performance.** We quantify the behavioral performance on the value-based decision-making task by regressing the fraction of choices for right offer to the difference in the expected value of the two offers $EV_R - EV_L$ (Fig. 1B). Precisely, we fit choice data (*chR* = 1 if choice is right, 0 if choice is left) to the model: $\text{logit}(p_{chR}) = \beta_0 + \beta_1(EV_R - EV_L)$ of the choice probability $p_{chR}$, and assess the significance of linear interaction term $\beta_1$ via *F*-Statistic test to compare the model fit to a constant mean model (with $\beta_1 = 0$). The regression is performed via least squares estimation, ±95% C.I. is computed via the estimation of inverse cumulative t-distribution given the coefficients ($\beta_0, \beta_1$), covariance matrix and standard errors of the fit.

**Analyses of eye position and gaze shifts.** The spatial distribution of horizontal eye position across task epochs (Fig. 1C, top) is computed

by counting occurrences across trials with spatial resolution 0.14 cm at a temporal resolution of 1 ms. The counting is then applied by splitting trials where subjects mainly looked either left or right screen side (Fig. 1C, bottom). We call a trial *LookL* or *LookR* based on whether the average horizontal eye position in 10 ms bins is negative or positive, respectively. The 2D spatial distribution of eye position on the screen is reported as heatmaps around the center of the visual scene (Fig. 1D), binned and smoothed with a gaussian filter ($\sigma_x = \sigma_y$ = 5 bins, 0.7 cm). The 2D analysis is initially run for each task epoch and by pooling data across all sessions from the two subjects (Fig. 1D), then repeated by separating data from the two subjects (Supplementary Fig. S2A), for the two orders of presentation (first = L, or first = R; Supplementary Fig. S2B), for best offer *EV* (best L: $EV_L > EV_R$, or best R: $EV_R > EV_L$; Supplementary Fig. S2C), and for different choices (choice = L, or choice = R; Supplementary Fig. S2D). Results for different conditions are subtracted to highlight differences between best offer cases (best R − best L, Supplementary Fig. S2E) and chosen offers (choice R − choice L, Supplementary Fig. S2F).

We analyzed horizontal gaze shifts, which we defined as monotonic variations in the horizontal eye position with duration of at least 25 ms (Supplementary Fig. S1C). The shifts are labeled by horizontal direction (left/right) based on the sign of their first order discrete-time derivative (left if the eye position derivative is negative), at 1 ms resolution. We show the time histograms of horizontal gaze shifts both for shifts within each visual hemifield and by also including midline-crossing (Supplementary Fig. S1C).

**Probability of choice as a function of offer expected values and looking times.** For each task-related time epoch (*offer 1, delay 1*, etc.) and for each trial, we computed the fraction of time looking at the right screen side, $f_R = t_R/(t_R + t_L)$, where $t_R$ is the total time that the subject spends looking at the right screen side, and $t_L$ is the total time looking at left screen side in that epoch and trial.

We further analyzed the choice probability $p_{chR}$ (considering *chR*= 1 if choice is right, 0 if left) as a function of the difference in expected value of the two offers $EV_R - EV_L$ (Fig. 2A). The choice probability was fit using least squares method to the model $\mathrm{logit}(p_{chR}) = \beta_0 + \beta_1(EV_R - EV_L)$ as in Methods "Behavioral performance". First, we run the analysis including all trials, then we split trial pools where subject mainly look left ($f_R < 0.25$; light red in Fig. 2A), or right ($f_R > 0.75$; dark red in Fig. 2 A). The ±95% C.I. is computed via the estimation of inverse cumulative t-distribution given the coefficients ($\beta_0, \beta_1$), their covariance matrix and standard errors of the fit. The differences in slope show that the eye position shifts the logistic relationship between choice and *EV* difference, hence that eye position affects the choice beyond the *EV* differences.

Following a similar logic, we use the logistic regression model $\mathrm{logit}(p_{chR}) = \beta_0 + \beta_1(t_R - t_L)$ for the choice probability on a trial-by-trial bases by including as regressor the difference in right vs left screen looking times $t_R - t_L$ within each trial (Fig. 2B). The difference in looking times $t_R - t_L$ is normalized to the full epoch duration ($t_R + t_L$), and therefore the value of this regressor always falls within [−1, +1]. The ±95% C.I. is computed via estimation of inverse cumulative t-distribution as before. We find a significant relationship between choice and difference in looking times for the two screen sides in all task epochs. The same analysis is repeated for trial pools where first offer is best ($EV_L > EV_R$, light purple in Fig. 2B) and second offer is best ($EV_R > EV_L$, dark purple in Fig. 2.B). Note that we reference first offer to be always on the left side (data in trials where the first offer is presented on the right side are mirrored to left).

Further than this, we refine the above models by combining task variables in the logistic regression model $\mathrm{logit}(p_{chR}) = \beta_0 + \beta_1 EV_L + \beta_2 m_L + \beta_3 \sigma_L^2 + \beta_4 EV_R + \beta_5 m_R + \beta_6 \sigma_R^2 + \beta_7 s_{LR} + \beta_8 f_R$ of choice probability $p_{chR}$ again based on the choice variable *chR*= 1 if

choice is right, 0 if choice is left (Fig. 2C). We included the following trial-by-trial regressors: expected value of left and right offers $EV_L$ and $EV_R$ (computed as $EV = mp$); magnitudes $m_L$ and $m_R$; risk, or variance of left/right offers, $\sigma_L^2$ and $\sigma_R^2$ (computed as $\sigma^2 = mp(1 - p)$); order of offers presentation $s_{LR}$ (+1 if first offer is left, −1 first offer is right); and fraction of time looking at right screen side $f_R = t_R/(t_R + t_L)$. To compare the magnitudes of the regressors, we normalized $EV, m$ and $\sigma^2$ to their maximum value across all trials and sessions, and in both subjects, so for the analyses on the whole data set their trial-by-trial values ranged within unitary values (in the set [0, 1]). Comparisons between various logistic regression models of the choice probability $p_{chR}$ and choice *chR* prediction accuracy over *k*-fold cross-validation ($k = 4$) data subsets are shown in Supplementary Fig. S4. Results for the regression of *EV*, *m*, and $\sigma$ are reported in Supplementary Table ST3. Results are also shown separately for the two subjects (Supplementary Fig. S3A). To test the effect of the fraction of looking time ($f_R$) on the choice, we added to the previous logistic regression model the fractions of looking at the right side in each of the following relevant task epochs: *offer 1, delay 1, offer 2, delay 2* (Fig. 2D), showing that $f_R$ in *delay 2* has significant impact on the choice ($p = 0.029$, FDR corrected via Benjamini & Hochberg[47] method) even when other regressors are included. We performed an additional analysis by redefining the looking times $t_R$ and $t_L$ as the total amount of time when the gaze of the subjects is inside the physical locations of the offers (or when they are left empty) (Supplementary Fig. S3B). The locations of the visual stimuli are: left side, horizontal coordinates $-7.5 \pm 1.5$ cm; right side, horizontal coordinates $+7.5 \pm 1.5$ cm; Left and right sides: vertical coordinates between ± 6 cm. This allowed us to determine whether gaze spatial precision is important for the encoding of the stimulus (gazing to the offer vs looking to the screen side where the offer was located). We find no qualitative difference in our results when using screen side vs offer looking times (Supplementary Fig. S3B; compare with Fig. 2C).

## Analyses of neural data

Following the manual isolation of units, spiking activity data were preprocessed offline with Offline Sorter (Plexon, USA) and digitized at 1 kHz resolution. The resulting data have an average spike rate of $1.46 \pm 2.38$ (mean ± s.d.) spikes/s. Given the sparsity of the data, to improve the detectability of encoding of task variables, we always applied the analysis of neural data by using a time-resolved strategy based on boxcar sliding window.

**Neural spiking and the encoding of reward expected value.** For each of the recorded units, we computed the time-averaged spike count $\eta$ in time windows of 200 ms, in every 10 ms bins (consecutive time windows have 95% overlap; time windows start at every 10 ms bin). These settings were chosen because they provided a good trade-off between the temporal resolution of our analysis and large enough time windows to compute spike rates given the sparsity of OFC activity. These settings were in line with ranges used in previous studies of OFC activity[45]. The spike rate in each time window, for each neuron, and in each 10 ms bin was fit to the linear model $\eta = \beta_0 + \beta_1 EV$, where the *EV* was either of the left offer, $EV_L$, or of the right offer, $EV_R$. The linear model fit is implemented by Ordinary Least Squares estimation, yielding unbiased model weight estimates, whose variance is inversely proportional to the number of available trials, assuming independent trials. As it is often the case for this class of estimators, the assessment of significance of linear weights over low numbers of trials could lead to missed detection (type II errors, failing to reject the null hypothesis that $\beta_1 = 0$, i.e., testing non-significant the hypothesis that $\beta_1 \neq 0$ when this is true), but never generate false positives (rejecting $\beta_1 = 0$, type I errors, i.e., testing

significant the hypothesis that $\beta_1 \neq 0$, when this is not true) beyond the significance threshold used (we used 0.05).

The time-resolved, cell-by-cell linear model analysis was performed using three different sets of trials: (1) using all available trials, i.e. neglecting where the subject is looking; (2) using all trials within a session where the subjects mainly looked at the left side of the screen in a given bin (defined as trials where the average gaze position within the 10 ms time bin is negative); and (3) trials where the subjects mainly looked at the right side (average gaze position is positive). The sets of trials (2) and (3) allowed to study the modulatory effect of gaze on the encoding of the value of the offers by distinguishing trials where subjects mainly *'Look Left = LookL'* from trials where subjects mainly *'Look Right = LookR'*, respectively (Fig. 1C, bottom, shows the fraction of trials for *LookL / LookR* for each session, in each time bin). The number of available trials could vary across sessions ($n = 746.38 \pm 87.29$ mean $\pm$ s.e.m. trials per session, $n = 5971$ in total; Supplementary Table ST1 for exact numbers for each session) but coincided for simultaneously recorded cells. Given that both subjects shared the look-at-something and look-at-nothing gaze patterns for the two screen sides, we considered looking conditions as the opposite sides of the screen (*LookL / LookR*), which also allowed to maximize the usage of available trials. We did not consider looking conditions as based on perfect alignment of gaze position over stimuli presentation sites as the subjects did lack precision in reaching back to the exact locations of previously seen, but currently empty offer stimuli during delay times (e.g. *delay 2* in Fig. 1D, E; Supplementary Fig. S2). Such looking conditions drastically reduce the number of available trials, preventing us from using finer resolution for gaze in in neural encoding analyses, especially at delay times (at *delay 1* time the fraction of available trials is 9.79% of the total for *LookL*, 1.75% for *LookR*; at *delay 2* time it is 3.76% for *LookL*, 7.43% for *LookR*; see Supplementary Table ST2 for available trials at opposite screen sides).

To compute the fraction of significant cells, we pooled cells across all sessions for the two subjects ($n = 31 \pm 5.85$ mean $\pm$ s.e.m. cells per session, $n = 248$ in total; Supplementary Table ST1 for exact numbers per session). The significance of each cell in encoding the offer *EV* in each time bin is assessed via *F*-Statistic tests comparing the linear model vs constant model, thus assessing whether the linear coefficient $\beta_1$ is significantly different from zero ($p < 0.05$). From this, we estimated the fraction of significant cells over the total number of available cells (for both monkeys, Fig. 3B; for the two monkeys separately, Supplementary Fig. S9). In our results we do not find qualitative change if shorter (150 ms) or longer (250 ms) time windows are used for spike rates (Supplementary Fig. S10), nor if we applied baseline normalization (for each cell, for each trial, we subtracted the spike rate in each 10 ms bin by average spike rate in *pre-offer 1*, Supplementary Fig. S11).

To assess the where the fraction of cells was significantly above change, we used a *bin-wise trial-order* permutation test where we shuffled trials independently per bin (this destroys temporal correlations, but in this first analysis we are just interested in the significance of the fraction of cells per bin, not across time; see below for the statistical '*run length*' analysis across bins). Specifically, for each time bin, we shuffled the trial order of EVs independently for each cell. For each of the $n = 1000$ shuffles generated, we computed the fraction of cells that encoded either the left or right EV (including all trials, e.g., Fig.3B or conditioned to *LookL/LookR*, e.g., Fig. 3D). The 1000 fractions were used to build the null-hypothesis distribution (reported as shaded areas in all related figures). Finally, the 95$^{\text{th}}$ percentile (right tail) of the null-hypothesis distribution is used to test at 5% significance level if the observed fraction of significant cells is significantly larger than the one expected by chance. We chose a right-tailed test because we consider only meaningful fractions of significant cells larger than expected by chance, not smaller. In all figures where analysis is made based on *LookL* or *LookR* conditions, bin-wise trial-order shuffles are made

within each condition, such that the potential correlation between offer values and gaze are not destroyed and thus the resulting null hypothesis distribution from the shuffling constitutes a valid comparison. Note that since within a single trial, different bins can correspond to either the *LookL* or *LookR* conditions, the identity of trials used in the analysis across consecutive bins does not need to match.

Although we mostly showed results for the neural encoding in terms of fraction of significantly encoding cells, we also show the coefficient of determination $R^2$ for all the regressions that we implemented (Supplementary Fig. S12A). In addition, we repeated our significant tests based on *F*-tests by (non-parametric) permutation tests on the values of $R^2$ (Supplementary Fig. S12B). In the permutation test, the trial order of *EVs* and spike-counts are shuffled independently $n = 1000$ times in each 10 ms time offset bin (trial order shuffles were independent also across offset bins), for each cell, in each session, for the two subjects. Then, the empirical $R^2$ (based on the not shuffled data) is compared to the $n = 1000$ shuffled $R^2$ values, and the *p*-value is defined as the fraction of shuffles with $R^2$ larger than empirical $R^2$ [31]. Finally, we counted the cells with significant encoding ($p < 0.05$). The results for the permutation test (Supplementary Fig. S12B) qualitatively match results run via *F*-statistic tests (Fig. 3B, D).

In the previous bin-resolved analyses, it is possible to find significance fraction of cells in bins just by chance (i.e., the eventuality that the fraction of significant cells in any time bins crossed the 95th percentile significance threshold due to spurious fluctuations). To address this issue, we performed a cluster-based *run-length* analysis[38]. We defined the length of a run as the number of consecutive significant time bins (significant here means that in the current time bin we have a fraction of significant cells exceeding the 95th percentile of the fraction of cells from shuffled data). We considered that a run length was significantly longer than expected by chance if its length exceeded the 95th percentile of run lengths obtained from trial-order shuffled data. The time bins involved in significantly long runs are shown by the solid line at the bottom of each panel in Fig. 3B, D, G, F, in Fig. 4A, C, and in Supplementary Figs. S5-S12, S15. For this specific analysis, we used a fixed shuffling order so that all the bins within a trial had the same shuffled trial order (differently from the bin-wise trial-order permutation test described before, where trial order is randomized in each bin), yielding empirical 95th percentiles of 5 bins (50 ms) used as minimum significant run length in all solid lines at the bottom of Figs. 3B, D, G, F, 4B–E, 5A, C, Supplementary Figs. S5-S12, S15. This implementation was motivated by the interest in preserving the autocorrelation of spike rates across bins, although it could affect the correlations between looked side and expected value. Despite this, we found no qualitative differences between the resulting 95th percentiles of fractions of significant cells computed with this method and the bin-wise shuffling method (bin-wise shuffling 5-to-95th percentiles are shown as shaded areas in Figs. 3B, D, G, F, 4 and 5A, C, Supplementary Figs. S5-S12, S15).

Since the number of trials available for the *LookL* and *LookR* conditions could vary at each time bin (Fig. 1C), we repeated the analysis of Fig. 3D by controlling for the unbalanced number of trials in the two pools. We performed a sub-sampling trial method equating the number of trials for each condition before performing statistical comparisons (Supplementary Fig. S7). In each time bin, the number of trials for both *LookL* and *LookR* is set to $n(t) =$ min (number of trials *LookL*, number of trials *LookR*) in each session (e.g., $n(t) =$ min(60, 535)). The number of subsets at each time bin is given by $m(t) = \lceil N/n(t) \rceil$, with $N$ the total number of trials in each session (e.g., $m(t) = \lceil 595/60 \rceil = 10$). Then, the average fraction of significant cells is computed for each subset, and finally the fractions for all $m(t)$ subsets are averaged. In Supplementary Fig. S7 we show that the main results (Fig. 3D, E) are found by sub-sampling (Supplementary Fig. S7C, E),

yielding as well equivalent overall levels of $R^2$ for *LookL* and *LookR* (Supplementary Fig. S7D, to compare with S12A).

**Difference in encoding $EV_L$ versus $EV_R$ in task epochs, and difference in encoding $EV_L$ or $EV_R$ when looking at different looking sides: *LookL* versus *LookR*.** We tested whether in any task-related epoch (e.g., *offer 1*, *delay 1*, etc.), cells showed more encoding of $EV_L$ or $EV_R$. We first consider all the trials, regardless of where subjects directed their gaze (below, we conditioned the same analysis on the gaze position). For each task-related epoch, we first built two $n_{cells} \times n_{timebins}$ binary matrices, $M_L$ and $M_R$. In each matrix, the element $m_{i,j} = 1$ if cell $i$ at time bin $j$ has a significant encoding of the left or right $EV$ ($p < 0.05$, defined by $F$-tests on the linear term interaction coefficient described in Methods "Neural spiking and the encoding of reward expected value"), $m_{i,j} = 0$ otherwise.

We compute the fraction of time bins with significant encoding of either expected value by averaging the matrices $M_L$ and $M_R$ along time dimension in each task epoch, yielding vectors $u_L$ and $u_R$ with size $n_{cells} \times 1$. In Fig. 3C we report average ± s.e.m. across cells of the vectors $u_L$ and $u_R$. For each of the two cases, we computed the 95th percentile values from the distribution of the fraction of bins averaged across cells $u_L^{sh}$ and $u_R^{sh}$ for $n = 1000$ bin-wise shuffles of the trials order, used as significance threshold to be exceeded (in Fig. 3C, significance thresholds are reported as dotted lines, non-significant bars are reported in white, with colored frame). The two vectors $u_L$ and $u_R$ were compared via non-parametric Wilcoxon signed rank test ($p$-values were corrected for False Discovery Rate at FDR = 0.05 via Benjamini-Hochberg[47] procedure; *$p < 0.05$, **$p < 0.01$ and ***$p < 0.001$ in Fig. 3C).

After this, we analyzed the fraction of neurons that encoded the left and right expected values depending on the subject's gaze in each time bin within a task-related epoch. We divided trials into two groups: *LookL* trials, where average gaze position is <0 in the 10 ms time bin in each of those trials; and *LookR* trials, where average > 0. Focusing on either offer $EV_L$ and $EV_R$, we re-run the neural encoding analysis again to compute the fraction of cells significantly encoding for either looking side ($EV_L$ *LookL* and *LookR*, Fig. 3D, top; $EV_R$ for *LookL* and *LookR*, Fig. 3D, bottom). For both the $EV$s and for the two trial pools, we computed again the vectors $u_L$ and $u_R$ as above, called them $u_L^{LookL}$ and $u_R^{LookL}$ for the *LookL* trials, and $u_L^{LookR}$ and $u_R^{LookR}$ for the *LookR* trials. Like in Fig. 3C, we report average ± s.e.m. across cells of the vectors $u_L^{LookL}$, $u_L^{LookR}$ (Fig. 3E, top), and $u_R^{LookL}$, $u_R^{LookR}$ (Fig. 3E, bottom) (significance thresholds are reported as dotted lines, non-significant bars are reported in white, with colored frame). We compared the encoding of left expected value $EV_L$ when the subjects mostly looked at left versus the encoding when it mostly looked at right, that is, we compared $u_L^{LookL}$ vs $u_L^{LookR}$, again, via non-parametric tests (one-tailed Wilcoxon signed rank test, corrected for FDR via Benjamini-Hochberg[47] procedure; *$p < 0.05$, **$p < 0.01$ and ***$p < 0.001$ in Fig. 3E, top). This time we use one-tailed tests as we are interested in *LookL* significance being more prominent than in *LookR*, and not just significantly different in their mean value. The same is done for the effect of gaze on the encoding of right expected value $EV_R$, i.e., we applied non-parametric tests to compare $u_R^{LookL}$ vs $u_R^{LookR}$ (Fig. 3E, bottom), this time assessing whether *LookR* was larger than *LookL*, as we wanted to test stronger encoding of $EV_R$. The results in Fig. 3C, E are also shown after removing trials when gaze fell within screen midline segments (±4 cm, ±9.5 cm from midline; Supplementary Fig. S13).

**Neural encoding at delay times conditioned on previous looking side.** To determine whether memory traces and reactivation of activity in OFC are mostly due to trials where subjects did not change their gaze (e.g., the subjects look at the left side during *delay 2* but they did look at the right side during *offer 2*), we repeated the encoding analysis during the delay epochs but making sure that we used only trials where the animal actually looked at the imminent previous offer. We first focused on *delay 1* time and compared *LookL* vs *LookR* by conditioning on trials when subjects were previously looking at the left during the last 200 ms of *offer 1* (looking at the left is of course the most common gazing pattern). We find that the subjects *LookL* during the last 200 ms of *offer 1* on 80.21% of the total amount of trials, they keep *LookL* during the last 200 ms of *delay 1* on 47.95% of total amount of trials, while they switch to *LookR* during the last 200 ms of *delay 1* on 32.26% of the total trials. For the two cases we applied independently the regression of $EV_L$ and $EV_R$, as described in Methods "Difference in encoding $EV_L$ versus $EV_R$ in task epochs, and difference in encoding $EV_L$ or $EV_R$ when looking at different looking sides: *LookL* versus *LookR*" (Fig. 3F). We find that the encoding of the first offer $EV_L$ persisted if the subjects keep looking at the left side of the screen during *delay 1*. Next, we focused our analyses on *delay 2* time and we compared looking conditions for trials when subjects looked at right side during the last 200 ms of *offer 2*, which is the most common pattern (78.8% of the total amount of trials). Again, we applied the time-resolved regression of offer $EV$s to compare trials when subjects shifted to *LookL* (36.48% of total amount of trials) versus the trials when subjects kept *LookR* (42.32% of total amount of trials) during *delay 2* (Fig. 3G). We strikingly find significant increase in the encoding of left offer $EV_L$ when subjects look back to the first offer location during *delay 2*, despite the screen was left blank (left, dark orange line); and this reactivation occurred even when the second offer was looked at, indicating that reactivation is robust to distractors. For $EV_R$, just like for $EV_L$ in *delay 1*, we find that its encoding persists only if the subjects keep staring at ipsilateral screen location (right, light purple line). The analyses were also run for less common, complementary cases (*LookR* during *offer1* and *LookL* during *offer 2*), shown in Supplementary Figs. S5. The central result in Fig. 3G (left) of reactivation when looking back to the left side is also replicated when subsampling to equally sized trial pools (Methods "Neural spiking and the encoding of reward expected value") shown in Supplementary Fig. S8J.

In Fig. 4B, C we replicate reactivation results in Fig. 3G-F by time-locking data to the onset of first midline-crossing gaze shift following delay time start (shift detection is detailed in Methods "Analyses of eye position and gaze shifts"). To maximize trials usage, and to simplify the interpretation, we considered that the first midline-crossing gaze shifts could relate to value encoding reactivation even if they were transient (thus followed by further screen midline gaze shifts) due to variable response delays. Note that we did not consider gaze-shifts posterior to first gaze-shift, hence we used all timepoints starting at first shift time and until the end of current epoch. We checked that the exclusion of data in timepoints posterior to midline-crossing gaze-shifts happening later than first shift yielded to qualitatively match in the results. In Fig. 4B, C we analyze encoding starting 400 ms prior to each gaze shift (negative axis) to show the changes in encoding strength for pre- and post-shift times. In addition, in Fig. 4D, E we show patterns of encoding in trials where subjects only ever look at the same screen side during delay times. In this case data are aligned to delay start and only include trials where gaze did not cross screen midline, by which we intended to show exclusive encoding of immediately previous, ipsilateral offers.

**Neural encoding of offer subjective value and choice.** We further tested a more complete model where we applied the regression of both offer Subjective Values ($SV$) and the choice variable $ch$ ($= +1$ if choice is for right offer, $-1$ otherwise), for which we show results in Fig. 5. The $SV$s are computed by using behavioral data in each session, as $SV_L = w_{1L}EV_L + w_{2L}m_L + w_{3L}\sigma_L^2$ and $SV_R = w_{1R}EV_R + w_{2R}m_R + w_{3R}\sigma_R^2$, regressing the choice variable ($chR = 1$ if choice is right, 0 if left) in the choice probability model $\text{logit}(p_{chR}) = w_0 - w_{1L}EV_L - w_{2L}m_L - w_{3L}\sigma_L^2 + w_{1R}EV_R + w_{2R}m_R + w_{3R}\sigma_R^2$ (Supplementary Table ST3). The definition of $SV$s followed the testing of alternative models of choice, hence by picking the one with best choice prediction accuracy over $k$-fold cross-validation ($k = 4$) data subsets

(Supplementary Fig. S4). For the analysis of neural data, we fit the full linear model $\eta = \beta_0 + \beta_1 SV_L + \beta_2 SV_R + \beta_3 ch$ to the spike rate $\eta$ in each 10 ms bins, using a sliding time window of 200 ms. This time the choice variable $ch$ is defined as $ch = +1$ if choice is right, $-1$ if choice is left. In this analysis we followed the same methodology as for the time-resolved analysis previously run independently for the encoding of $EV_L$ and $EV_R$ described in Methods "Neural spiking and the encoding of reward expected value".

To prevent from introducing artefactual collinearity by using the same choice data both in the linear regressions to compute $SVs$ and in the linear regressions of spike rates, we randomly subsample data in two equally sized, disjoint subsets and perform independent regressions in each subset[72]. The first subset of trials, $S^{(1)}$ is used to the fit the logistic model of the choice to compute $SV$ weights. The second subset $S^{(2)}$ with remainder trials is used to fit $SV$ to the full spike rate model including $ch$. In symbols, we compute $SV$ weights on trial variables from $S^{(1)}$ indexed with (1) by fitting the model $\text{logit}(chR^{(1)}) = w_0^{(1)} - w_{1L}^{(1)} EV_L^{(1)} - w_{2L}^{(1)} m_L^{(1)} - w_{3L}^{(1)} \sigma_L^{2(1)} + w_{1R}^{(1)} EV_R^{(1)} + w_{2R}^{(1)} m_R^{(1)} + w_{3R}^{(1)} \sigma_R^{2(1)}$. Then we compute $SVs$ on trial variables from $S^{(2)}$ indexed with (2) as: $SV_L^{(2)} = w_{1L}^{(1)} EV_L^{(2)} + w_{2L}^{(1)} m_L^{(2)} + w_{3L}^{(1)} \sigma_L^{2(2)}$ and $SV_R^{(2)} = w_{1R}^{(1)} EV_R^{(2)} + w_{2R}^{(1)} m_R^{(2)} + w_{3R}^{(1)} \sigma_L^{2(2)}$. Once the weights $w_{\{1,2,3\}\{L,R\}}^{(1)}$ are computed on $S^{(1)}$ strials and used to compute $SVs$ on $S^{(2)}$ trials, denoted as $SV^{(2)}$, the same is done for $w_{\{1,2,3\}\{L,R\}}^{(2)}$ and $SV^{(1)}$ (Supplementary Table ST3). The $SV^{(2)}$ and $SV^{(1)}$ are used to perform regression of spike counts, including the respective variables $ch^{(2)}$ and $ch^{(1)}$ from the same subsets. This procedure ensures that $SV^{(2)}$ and $SV^{(1)}$, respectively computed using $chR^{(1)}$, $chR^{(2)}$ are respectively regressed with $ch^{(2)}$ and $ch^{(1)}$ in disjoint subsets.

To investigate the modulatory effects of gaze position, we repeated the spike rate regression analysis by either including all trials (Fig. 5A) and focusing on *LookL* or *LookR* trials (Fig. 5C). Interestingly, we found that our main results for the two $SVs$ qualitatively matched results for the encoding of the two $EVs$ in isolation, even when including the choice variable, which in turn is significantly encoded in the latest task epochs. We first assess the fraction of significant cells by comparing them with the 95[th] percentile of the same quantities computed over $n = 1000$ shuffles of the trial order, to be used as significance threshold (in Fig. 5B and 5D, significance thresholds are reported as dotted lines, non-significant bars are reported in white, with colored frame). Then, for the encoding of $SVs$, we compared the fraction of significant bins in *LookL* and *LookR* (Fig. 5D, top and middle) via one-tailed Wilcoxon signed-rank tests, to test that either $SV$ had higher encoding for ipsilateral looking condition (following the derivation of the vectors $u_L$ and $u_R$ described in Methods "Difference in encoding $EV_L$ versus $EV_R$ in task epochs, and difference in encoding $EV_L$ or $EV_R$ when looking at different looking sides: *LookL* versus *LookR*"). For the encoding of $ch$ (Fig. 5D, bottom), we compared *LookL* and *LookR* (again using $u_L$ and $u_R$ computed as in Methods "Difference in encoding $EV_L$ versus $EV_R$ in task epochs, and difference in encoding $EV_L$ or $EV_R$ when looking at different looking sides: *LookL* versus *LookR*") via two-tailed Wilcoxon signed-rank tests to test that the two conditions showed non-zero mean difference. All *p*-values were corrected for multiple comparisons with acceptance rate FDR = 0.05 via Benjamini-Hochberg[47] procedure.

**Analysis of regression residuals: preferred vs non-preferred choice.** To test the trial-to-trial relationship between spike rate, choice and offer value reactivation in OFC, we further analyze the residuals of linear regression model fits described in Methods "Neural encoding of offer subjective value and choice". by considering the above variables $SV_L$ and $SV_R$. For the *delay 2* epoch and for each cell, we compute the spike rate $\eta_{i,t}$ in each trial $i$ and at each time $t$ every 10 ms using a time window starting at bin start time and covering the following 200 ms. In each time

bin, and in each cell, we apply a simplified regression model $\eta_{i,t} = \beta'_{0,t} + \beta'_{1,t} SV_{L,i} + \beta'_{2,t} SV_{R,i}$, where $SV_{L,i}$ and $SV_{R,i}$ are the left and right subjective values in that trial. From here, we extract the residual activity $z_{i,t} = \eta_{i,t} - (\beta'_{0,t} + \beta'_{1,t} SV_{L,i} + \beta'_{2,t} SV_{R,i})$ for trial $i$ at time bin $t$. We study the relationship between choice and residuals as a function of the sign of time-averaged $\beta'_1 = \langle \beta'_{1,t} \rangle$ and $\beta'_2 = \langle \beta'_{2,t} \rangle$, performing this analysis in trials where subjects *LookL* and *LookR* separately. We found cells with positive time-averaged $SV$ tuning ($\beta'_1 > 0, \beta'_2 > 0$) and, respectively, negative time-averaged $SV$ tuning ($\beta'_1 < 0, \beta'_2 < 0$) in both *LookL* and *LookR* sets of trials. We pooled trials and cells based on *preferred* choice (i.e., positive $\beta'_1$ and choice=L, or negative $\beta'_1$ and choice=R for $SV_L$; positive $\beta'_2$ and choice=R, or negative $\beta'_2$ and choice=L for $SV_R$) or *non-preferred* choice (positive $\beta'_1$ and choice=R, or negative $\beta'_1$ and choice=L for $SV_L$; positive $\beta'_2$ and choice=L, or negative $\beta'_2$ and choice=R for $SV_R$). Thanks to this pooling, we could test whether each cell, being it positively or negatively related to $SV_L$ and $SV_R$, showed larger residuals for their *preferred* or *non-preferred* choice. Following the definition of $SVs$ in Methods "Neural encoding of offer subjective value and choice", we resort to disjoint subsets also for the analyses of regression residuals. Namely, the $SV^{(1)}$ defined for subset $S^{(1)}$ are used as regressors to fit the simplified model of the spike rate from trial $i$ at time bin $t$ as: $\eta_{i,t}^{(1)} = \beta_{0,t}'^{(1)} + \beta_{1,t}'^{(1)} SV_{L,i}^{(1)} + \beta_{2,t}'^{(1)} SV_{R,i}^{(1)}$ and to extract the residuals $z_{i,t}^{(1)}$, defined above. The whole procedure is repeated by swapping $S^{(1)}$ with $S^{(2)}$ to compute $z_{i,t}^{(2)}$ on $S^{(2)}$ trials (the weights to compute $SVs$ in $S^{(1)}$ come from behavioral fits using $S^{(2)}$, and vice versa, Methods "Neural encoding of offer subjective value and choice"). At this point, residuals for each cell are time-averaged in each trial, $z_i^{(1,2)} = \langle z_{i,t}^{(1,2)} \rangle$, and pooled in *preferred* choice and *non-preferred* choice for $SV_L$ and $SV_R$. For $SV_L$: $z_{pref} = [z_{\beta'_1>0,ch=L}^{(1)} || z_{\beta'_1<0,ch=R}^{(2)} || z_{\beta'_1>0,ch=L}^{(2)} || z_{\beta'_1<0,ch=R}^{(1)}]$, where $||$ denotes vector concatenation, and $z_{non-pref} = [z_{\beta'_1>0,ch=R}^{(1)} || z_{\beta'_1<0,ch=L}^{(2)} || z_{\beta'_1>0,ch=R}^{(2)} || z_{\beta'_1<0,ch=L}^{(1)}]$. The same definition holds for $z_{pref}$ and $z_{non-pref}$ tuned to $SV_R$, but replacing $\beta'_1$ with $\beta'_2$ and choice = L/R with R/L. In Supplementary Fig. S14B we show time- and trial-averaged residuals. Note that the length of residual vectors ($n$, at the bottom of each panel) could exceed the total of cells available since the sign of $\beta'_1$ and $\beta'_2$ could vary in the two disjoint subsets, but this was not problematic as collecting all available residuals always resulted in paired and equally sized *preferred* or *non-preferred* pools. The difference in *LookL* vs *LookR* was tested separately for data tuned to the weights of $SV_L$ (Supplementary Fig. S14B, top) and $SV_R$ (Supplementary Fig. S14B, bottom) via one-tailed, paired Wilcoxon signed rank tests assessing *preferred* choice > *non-preferred* choice.

**Analysis of regression residuals: Choice Probability.** We first define Choice Probability (CP) tuned to the $SV_L$ sign (Fig. 5E): if the cell has positive time-averaged tuning ($\beta'_1 > 0$), the CP is given by the fraction of choice = L (*preferred*) trials showing larger time-averaged residual magnitude than choice = R (*non-preferred*) trials; if the cell has negative time-averaged $SV_L$ tuning ($\beta'_1 < 0$), the CP is given by the fraction of choice = R (*preferred*) trials showing larger time-averaged residual magnitude than choice = L (*non-preferred*) trials. Through the convenient definition of time-averaged residual pools $z_{pref}$ and $z_{non-pref}$ in Methods "Analysis of regression residuals: preferred vs non-preferred choice", the above comparison is implemented by computing the fraction of trial occurrences where $z_{pref} > z_{non-pref}$. The number of trials available in *preferred* and *non-preferred* conditions could vary due to the unbalanced size of choice = L and choice = R trial pools, but CP computation is unaffected by this[50]. Indeed, the CP is computed as the fraction of occurrences where a randomly sampled residual from the

pool $z_{pref}$ is larger than a randomly sampled residual from the pool $z_{non-pref}$. We repeated the random sampling procedure data $n = 1000$ times to estimate the CP (the results would not yield qualitative difference by using larger $n$). This method is equivalent to the Receiver Operative Characteristic (ROC) method (Methods "Analysis of regression residuals: cell-wise and cells-combined ROC") and leads to numerical match in the results. Similarly, the definition of CP tuned to $SV_R$ sign and choice (*preferred*: $\beta'_2 > 0$ and choice = R or $\beta'_2 < 0$ and choice = L; *non-preferred*: $\beta'_2 > 0$ and choice = L or $\beta'_2 < 0$ and choice = R) is based on residuals magnitude comparison (Fig. 5F). CPs are separately computed for *LookL* and *LookR* trials for each cell, then CPs for the two looking conditions are compared via paired, one-tailed Wilcoxon signed rank tests to assess that each cell showed larger $SV$-sign-tuned CP for ipsilateral looking condition (*LookL* > *LookR* for $SV_L$, *LookR* > *LookL* for $SV_R$).

**Analysis of regression residuals: cell-wise and cells-combined ROC.** The performance of the above-described choice detection based on the magnitude of residuals across trials and extracting CPs is equivalent to the Receiver Operating Characteristic (ROC) method. The ROC analysis consists in extracting the rate of correct detection (TPR, True Positives Rate, y-axis in ROC plots; TPR = TP/(TP + FN) where TP = true positive and FN = false negative); incorrect detection (FPR, False Positives Rate, x-axis in ROC plots; FPR = FP/(FP + TN) where FP = false positive and TN = true negative); and in computing the Area Under the Curve (AUC) by cumulating the area under ROC. We applied ROC analysis '*cell-wise*', i.e., computing TPR and FPR for each cell by predicting choice via comparison of trial-by-trial time-averaged residuals magnitude (*preferred* vs *non-preferred* defined as in Methods "Analysis of regression residuals: preferred vs non-preferred choice", Supplementary Fig. S14A). We extracted TPR and FPR by counting the occurrences of trials where choice is correctly or incorrectly detected (hence computing TP, FN, FP, TN factors) according to the respective choice predictions and actual choices. This led to *cell-wise* AUCs numerically equal to CPs in each cell (thus median CPs in Fig. 5E-F match median AUCs in Supplementary Fig. S14A).

### Reporting summary
Further information on research design is available in the Nature Portfolio Reporting Summary linked to this article.

## Data availability
Data is available at the https://doi.org/10.12751/g-node.evlnq5.

## Code availability
Code is available at the https://doi.org/10.12751/g-node.evlnq5.

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

## Acknowledgements

We appreciate Ramon Nogueira for discussion about core methods. This project was supported by grants from the Howard Hughes Medical Institute (HHMI; Ref: 55008742), The Bial Foundation (Ref: 106/2022), Ministerio de Ciencia e Innovación (Ref: PID2020-114196GB-I00/AEI) and ICREA Academia (2022) to R.M.B.

## Author contributions

D.F., R.M.B. conceptualized and designed the analyses. B.H. ideated the task. D.F. analyzed the data. B.H., T.C.P., M.Z.W. collected the data. D.F., B.H., R.M.B. wrote the manuscript.

## Competing interests

The authors declare no competing interests.
