## [Peer Review File · Nature Communications]

REVIEWER COMMENTS

Reviewer #1 (Remarks to the Author):

Ferro and colleagues report behavior and neural recordings in the OFC of macaque monkeys performing a value-based decision task used in many prior studies from this group. In each trial, the monkey is presented with two decision offers that vary according to the magnitude and probability of juice reward. The offers are presented one at a time on opposite sides of a computer screen, after which the monkeys are prompted to indicate their chosen offer with an eye movement. Most relevant to this study is the fact that between offer1 and offer2, and between offer2 and the choice prompt, there are 600ms delay periods in which the screen is blank. The main claim of the study is that during the delay periods, the monkeys frequently look at the locations where the offer stimuli previously appeared, and that doing so “reactivates” neural encoding of the values of the offers previously at those locations. An additional claim is that this gaze-triggered activity is a form of “reevaluation” of the offers, given that during the delay periods neural activity also predicts upcoming choices. If true, these claims would suggest a new and important function for OFC during decision-making, i.e., the computation of spatially-based memories for value.

However, it is not clear from the presentation of the data that these claims fully hold up to scrutiny. These issues could potentially be resolved with additional analyses and/or adjustments to existing analyses, as suggested below:

1) The first and strongest claim of the study is that viewing an “empty” offer location during the delay reactivates a representation of the value of the offer. Figure 3D and E show this by dividing trials, bin-by-bin, according to whether the gaze was detected on the left or right of the screen, and then counting the fraction of cells significantly encoding the value of the offer in a linear model. This approach appears to have two weaknesses.

The first is that whether or not a cell is significant depends in part on the number of trials used to fit the model. Because gaze location varies as a function of time in the trial (Figure 1C), the dynamics in the fraction of significant cells could be attributable in part to the changes in gaze location, rather than changes in neural encoding. The control analysis in Supplementary Figure S9 addresses this issue by mandating that equal numbers of trials are used in each time bin. However, it appears that several of the results are less robust when this control is performed, e.g. the dark green line during delay1, and the light blue line during delay 2. It would be helpful to see a version of Figure 3E that uses equal-trial-number data, in order to assess which effects hold up under this control. Additionally, there needs to be an explicit discussion of this issue, and acknowledgment of any results that differ between the main analysis and this control.

The second weakness is that other variables, such as EV or risk, could be correlated with gaze allocation; this is not an unreasonable assumption, given prior studies showing that gaze is influenced by the values of visible targets[1–3]. If this were true, stratifying trials into LookL/LookR would also be stratifying by another variable. As a result, even when the trial numbers are equal, it would be difficult to rule out the possibility that the effects of gaze direction are explained by other variables.

2) Figures 3F and G are extremely important for the main claim that gaze onto empty offer locations activates representations of value. The idea behind these analyses is to divide trials into those where the gaze did vs. did not transition away from the offer during the delay period. However, it would appear that this analysis seems to suffer from the same issue regarding the unequal numbers of trials in the two groups, given that the analysis performed here is the same as in Figure 3D. These analyses could be repeated with the same procedure as in Figure S9, to resolve this issue.

3) Related to the above: Figure 3F and G imply that when gaze transitions from one side to the other, encoding in OFC neurons changes to reflect the gaze location. However, this is only implied and is not directly shown, because at no point are any analyses performed with data time referenced to the gaze transitions (i.e. saccades from one side of the screen to the other).

One way to correct this issue, and to provide strong support for the main claims of the study, would be to show that changes in OFC encoding are reliably time-locked to gaze transitions, as has been shown for visible targets in other studies[4]. Note that this kind of analysis will also cleanly address the two weaknesses in comment 1 above, because the critical comparison will be the encoding that occurs before vs. after the gaze transition over the same fixed set of trials – rather than the encoding observed between different sets of trials (LookL/R) that could be unequal in number and that could differ by some confounding variable.

4) The second major claim is that “variability in the size of [gaze-based] reactivation predicts choice” (line 421-422). It is not clear which data support this claim. Figure 4A shows that, after accounting for variability due to the subjective values, variability in spiking also predicts choice. And Figure 4C shows that after splitting trials according to gaze position, this effect still holds, though perhaps at a weaker magnitude given that fewer significant cells are detected. But these data show only that gaze-based reactivation and choice-predictive activity are both detectable in the same model at around the same time. To show that variability in reactivation predicts choice requires an analysis that computes some kind of single-trial measure of reactivation, and then shows that this measure correlates with choices on a trial-to-trial basis. For an example of an

analysis with similar goals in single cells, see the choice probability analyses of Conen and Padoa-Schoppa (2015)[5]; for an example in neural populations, see McGinty & Lupkin (2021)[6].

5) A 95th percentile significance threshold seems too lenient, even given the a priori expectation of greater-than-chance encoding. This is evident in the false positives in the pre-offer1 epochs in both Figures 3 and 4. I understand that some sort of cluster-based threshold was used to find minimum run lengths for significant effects; but this also seems too lenient, given the many extremely short epochs of significant encoding in Figure 3. The use of this lenient threshold for the time series data might be acceptable, in order to show the dynamics and general trends in the data. But for the data in Figure 3C and E, and Figure 4B and D, more needs to be done to confirm the robustness of the effects. I would recommend first documenting the raw p-values of the filled bars in these graphs (i.e. the fraction of shuffled instances greater than the observed). In addition, to address multiple comparison issues, I would strongly recommend applying some form of p-value correction to these, and reporting corrected p-values alongside the raw p-values.

6) The criteria used to designate LookL and LookR trials are not robust, and could easily be improved. The current criteria are simply whether the average gaze position was left or right of the screen center. This means that instances where the fixation is virtually at the center, or where the monkey is shifting his gaze across the center are going to be arbitrarily assigned as LookL or LookR – thereby adding variability to the results. This might be an appropriate criterion for the brief re-fixate and choice-go epochs, because gaze mostly seems to be at the center at these times (Figure S2); but this criterion seems inappropriate for the other (much longer) offer and delay epochs, where gaze is concentrated on the left and right offer locations. For finding LookL/R trials during the offer and delay epochs, it would be best to set some non-zero positive and negative boundaries around the center, so that at-center fixations are not assigned to LookL/R, and are either discarded or analyzed separately.

7) Line 211/Figure 2B: The claim is that the slope of the lines in Figure 2B are higher for the EV_R > EV_L condition, because these are trials in which the right offer is more valuable. But wouldn't having a higher right offer value make the *intercept* higher, reflecting an overall greater probability of choosing right? The increased slope means that in R>L trials, relative gaze time has a greater effect on the choice outcome – it doesn't mean that right choices are more likely. Related: In Figure 2B, isn't the expectation that the lines during the choice-hold period be non-overlapping, reflecting the different relative values of the offers, and therefore the different probabilities of choosing right?

7) Some additional details on the neural recording, sorting, and pre-processing would be welcome. First, can the authors confirm the use of single contact electrodes? If so, how many were used per session? If 1-3 neurons are isolated per electrode, this would imply using ~30 electrodes to record 59 neurons (monkey 1, session 2), which seems like an extremely large number of individual electrodes to insert/remove in each session. Were these chronic electrodes? Is it possible that

multi-contact probes were used in some sessions? More detail is needed. Second, electrode impedances are specified in mega-ohms or kilo-ohms, not in “MU” as written on line 642. Third, what were the isolation criteria used to accept/reject neurons for analysis, e.g. minimum unit amplitude relative to the noise, fraction of short ISIs, etc.? Fourth, line 652 mentions preprocessing steps including “thresholding and quantization”; what exactly does this mean? Does thresholding refer to a minimum firing rate threshold? Does quantization refer to temporal binning? More detail is needed.

8) In an attempt to simplify the data presentation, right-first/left-second trials have been re-coded as left-first/right second trials, so that dissimilar trials can be analyzed in the same way. This is potentially confusing, and glosses over the fact that the OFC neurons have both spatial-based responses[7] as well as order-based responses in tasks with serial offers^{4,6}. To avoid confusion, and make the results more aligned with conventions of previous studies, I think it would be better to be transparent and simply designate the stimuli as 1st offer/2nd offer, or something similar.

9) The authors report that the monkeys only look at the offers in ~80% of trials. What happens on the 20% of trials where the monkey does not look at the offer initially, but then goes on to look at the empty offer location later. Is the same “reactivation” signal seen, even though the offer wasn’t viewed in the first place? Likewise, are reactivation effects stronger when these 20% of trials are removed from the main analysis?

10) In supplementary Figure 8C, why do the shaded areas differ for the two conditions? Presumably this is due to differences in the number of trials in each condition and bin? This needs to be explained in detail to avoid any confusion as to what is happening here.

12) There is potentially an alternative interpretation of the behavioral and neural effects in the delay₂ epoch. The data show that during this epoch, there is an offer 1 signal that appears while the subjects look at the location of the previously viewed target (which is not present). This is interpreted as being related to a reevaluation of the first offer as part of the ongoing decision process. However, there is an alternative explanation. There are some indications from recent studies in humans and NHPs suggesting that once an implicit decision has been reached, the eyes are often drawn to the location of the to-be-chosen target. See recent studies from Manohar[8] and Westbrook[9]. The idea is that late in a trial, gaze begins to reflect the latent outcome of the choice process. If such an effect were present here, then the gaze behavior and neural signals appearing late in the delay₂ epoch could reflect representations of the chosen value, rather than reevaluation of a previously visited offer. This is a subtlety that should be addressed in the discussion section.

13) The idea that the OFC “only encodes the value of fixated offers” is not universally supported. The discussion on lines 453-466 overlooks several studies showing that OFC encodes both the

currently viewed and, to some extent, previously viewed offers simultaneously in tasks with serial designs and/or free gaze. See studies from Hunt[4], Ballesta[10], Yoo[11], and McGinty[6].

14) Figure 4A says EV, but caption says SV.

15) Supplementary Figure 2 caption refers to Figure 2E; but Figure 2 has only panels A-D.

REFERENCES

1. Yasuda, M., Yamamoto, S. & Hikosaka, O. Robust Representation of Stable Object Values in the Oculomotor Basal Ganglia. *J. Neurosci.* 32, 16917–16932 (2012).
2. Cavanagh, S. E., Malalasekera, W. M. N., Miranda, B., Hunt, L. T. & Kennerley, S. W. Visual fixation patterns during economic choice reflect covert valuation processes that emerge with learning. *Proc. Natl. Acad. Sci.* 116, 22795–22801 (2019).
3. McGinty, V. B., Rangel, A. & Newsome, W. T. Orbitofrontal Cortex Value Signals Depend on Fixation Location during Free Viewing. *Neuron* 90, 1299–1311 (2016).
4. Hunt, L. T. et al. Triple dissociation of attention and decision computations across prefrontal cortex. *Nat. Neurosci.* 21, 1471–1481 (2018).
5. Conen, K. E. & Padoa-Schioppa, C. Neuronal variability in orbitofrontal cortex during economic decisions. *J. Neurophysiol.* 114, 1367–1381 (2015).
6. McGinty, V. B. & Lupkin, S. M. Value signals in orbitofrontal cortex predict economic decisions on a trial-to-trial basis. 2021.03.11.434452 Preprint at <https://doi.org/10.1101/2021.03.11.434452> (2021).
7. Yoo, S. B. M., Slezzer, B. J. & Hayden, B. Y. Robust Encoding of Spatial Information in Orbitofrontal Cortex and Striatum. *J. Cogn. Neurosci.* 30, 898–913 (2018).
8. Manohar, S. G. & Husain, M. Attention as foraging for information and value. *Front. Hum. Neurosci.* 7, 711 (2013).
9. Westbrook, A. et al. Dopamine promotes cognitive effort by biasing the benefits versus costs of cognitive work. *Science* 367, 1362–1366 (2020).
10. Ballesta, S. & Padoa-Schioppa, C. Economic Decisions through Circuit Inhibition. *Curr. Biol.* 29, 3814-3824.e5 (2019).
11. Yoo, S. B. M. & Hayden, B. Y. The Transition from Evaluation to Selection Involves Neural Subspace Reorganization in Core Reward Regions. *Neuron* 0, (2019).

Reviewer #2 (Remarks to the Author):

Summary

This study addresses a crucial and ongoing question in decision-making concerning the role of eye position, or gaze, in gating which visual offers (i.e., stimuli) are evaluated—and correspondingly which stimulus values are represented in the brain—thereby determining the ultimate decision. This work builds upon a 10- to 15-year body of research establishing the role of gaze in evaluating simple binary choices and showing, in monkeys, that gaze modulates the putative value representations in orbitofrontal cortex (OFC). The current study extends this work by testing this same role of gaze on choices and OFC activity, but now asking if/how eye position modulates the decision process after the stimuli have been removed from view, but when subjects nonetheless are presumably still deliberating. The authors integrate past work on the interaction between gaze and visuospatial memory to posit that eye position plays a similar role after stimuli have been removed as when the stimuli are present: gaze directed at the former location of a stimulus both indicates that the subject is evaluating that stimulus (from memory) and facilitates that reevaluation process insofar as gating which stimulus value OFC is presently representing, and thereby influencing which stimulus some downstream circuit will ultimately choose.

The authors have designed and implemented a study that is well suited to test this hypothesis, and offer evidence that is suggestive of their core idea. However, the most novel aspect of the hypothesis—that gaze facilitates retrieval of a value memory and gates OFC encoding (“reactivation”) of that memory—is not sufficiently supported by the manuscript in its current form. I believe that the findings in general, and this key finding in particular, would likely become both more convincing and accessible in a significantly revised manuscript.

My key concerns involve the rigor/design of the analyses, the mismatch between the strength of interpretation and evidence to support the claims, and the density and length of the exposition, which would benefit from significant distillation and prioritization. Finally, the manuscript would benefit from greater theoretical integration regarding the import of their findings on past literature and for models of visually guided decision-making. I have offered several suggestions for all of these areas and would be glad to review a revised manuscript.

Major Points

Analysis

Separate regressions

Throughout, separate regressions are used for each variable of interest instead of a single model including all variables.

This practice does not allow different regressors (e.g., gaze sidedness, epoch, offer values) to compete for variance, and thus cannot make claims about independence that, I believe, the authors wish to make, such as, “gaze in epoch A explains choice above and beyond that in epoch B”.

In addition, having so many separate models is confusing and risks overwhelming the reader.

This practice of using separate regressions is used throughout the behavioral and neural analyses, and is too frequent to enumerate each case. In general, I would recommend using a single regression in which terms of interest compete for variance and interaction terms are used to test the dependency between 2 or more variables (e.g., hypothesis that slope of $t_R - t_L$ depends on dEV).

Fig 2D is a welcome exception to the above in that includes multiple epochs in a single model. Can this type of approach be used throughout?

Fraction of neurons analyses

The neural analyses are based on fraction of significant neurons. This approach has several limitations that limit interpretability of the results.

First, this approach does not allow us to track neurons across epochs in the trial. That is, we do not gain insight into if or how quickly neurons are changing their encoding. For instance, in Fig 3 (though the concept applies to multiple figures), when the proportion EV_L goes down during offer2, is that because the former EV_L neurons are becoming silent and new neurons are coming on to encode EV_R, or are the EVL neurons switching to becoming EVR neurons?

Moreover, it seems the authors (understandably) want to make observations about how a given (set of) neuron(s) change over the trial (e.g., line 376, “a significant fraction ... carries signals through ... and became prominent), but the present analysis really can't say anything about a given set of neurons over time.

We also are blind to how some present feature (e.g., gaze-dependent firing rate) depends on some past feature (e.g., recent gaze). This is critical for interpreting delay-period activity, such as whether

the sustained gaze or OFC activity during delay² and related to offer² is greater than we'd expect from the temporal "inertia" (i.e., autocorrelation) of fixation or firing rate, respectively.

Notably, Fig 3F and 3G get at the spirit of dynamics by conditioning on the previous eye position.

Independent time-bin analyses

A second concern is that neurons are labeled "significant" based on independent analyses performed in finely spaced, overlapping time bins.

By computing "significance" independently in bins separated by 10ms, the authors are implying that what the neuron is encoding can change every 10ms. Is that the authors' belief? If so, it should be justified. My sense is that the authors want a finer temporal scale to correlate with eye position. However, eye position also has a longer time constant than 10ms, so again, the short interval requires justification, or consider the use of a longer interval.

A more serious concern is statistical. The analysis does not appear to adequately address the temporal dependency between time bins, both because the bins are overlapping (and so clearly are not independent!) and due to the temporal smoothness/autocorrelation in the spike trains. This likely contributes to a biased estimation of significance. It is not clear whether their permutation testing addresses this, since it seems that the permutation testing preserves the overlap and temporal autocorrelations. If I am mistaken, and the analysis adequately addresses these statistical concerns, that should be made more explicit and addressed in the Results section.

In recent years, several approaches for describing the above population-level dynamics (aka, "stability")—and statistical methods for testing those dynamics—have been proposed and should be considered for the present paper. To name a few:

- Murray, J. D. et al. Stable population coding for working memory coexists with heterogeneous neural dynamics in prefrontal cortex. *Proc. Natl Acad. Sci. USA* 114, 394–399 (2017).
- Kimmel, D. L., Elsayed, G. F., Cunningham, J. P., & Newsome, W. T. (2020). Value and choice as separable and stable representations in orbitofrontal cortex. *Nat. Comm.*, 11(1), 1–19.
- Najafi, F., Elsayed, G. F., Cao, R., Pnevmatikakis, E., Latham, P. E., Cunningham, J. P., & Churchland, A. K. (2020). Excitatory and Inhibitory Subnetworks Are Equally Selective during Decision-Making and Emerge Simultaneously during Learning. *Neuron*, 105(1), 165-179.e8.
- This paper concerns longer timescales between trials. The other papers concern within-trial dynamics.

Morcos, A. S. & Harvey, C. D. History-dependent variability in population dynamics during evidence accumulation in cortex. *Nat. Neurosci.* 19, 1672–1681 (2016).

Coarse analysis of eye position

The analysis of eye position is surprisingly coarse given the authors' stated hypotheses and interpretations.

A central theory of the manuscript is that gaze of a given stimulus gates cognitive deliberation about—and OFC encoding of—that stimulus, ultimately driving choices. However, analyses of eye position primarily concern whether subject is looking to the Left vs Right side of screen. (Gazing 51/49% is treated the same as 99/1%!). Why can't the analyses depend on whether subject is fixating the stimulus? Or if gaze position is highly variable WRT the stimulus, then use some continuous predictor, like distance between gaze and stimulus.

Moreover, where exactly were subjects looking relative to the offers? Was there anything systematic? Did choice and/or OFC activity depend on precise eye position? The authors state that "subjects preferentially look at the physical rectangular shapes of the offers" (line 130). But where on the (relatively large) shape? For instance, how does eye position vary with the height of the gray portion of the bar? What about during the delay period viewing? For instance, during the delay, did proximity to the former position of the bar predict greater choice accuracy (i.e., sensitivity to offer value)?

The authors state "subjects gaze did lack precision in reaching exact stimuli locations" (line 679). So how did this imprecision (variance) in eye position relate to choice and OFC activity. (As an aside, the imprecision was surprising given the large size of the stimuli.)

Another metric is based on the time fixating offer 1 vs 2, which at least has the advantage of being a continuous measure. But it is revealing a slightly different aspect of gaze—time vs. precision. How should we understand these 2 metrics? That is, what are they each telling us? And therefore, in the neural analyses, when should we use one over the other?

These questions about eye position are not incidental. Rather, the authors explicitly invoke precise eye position in making their central claim: "moving the eyes to another location, even if empty, washes out the neural memory of the previously seen offer in OFC: only the value of the offer that falls, or immediately fell, in the fovea is strongly encoded." (Line 449-51). This statement emphasizes a strong hypothesis that precise eye position—indeed foveation—modulates OFC's encoding of offer value from memory. And yet the analyses of eye position focus on the coarsest of

metrics—like left vs. right side of the screen—which seem totally at odd with the hypothesis. That is, if the difference in gaze location can result in the difference between "washing out" vs. "reactivation", then presumably OFC is exquisitely sensitive to gaze location; the analyses should be similarly precise.

Interpretation

The authors take a strong position that looking time during the delay period represents a "reevaluation" process that modulates OFC activity, which in turn reflects "reactivation" of memory for an offer, and ultimately has some (causal) impact on choice. While the authors are mindful of the limitations of a correlational study—and do not outright claim causal evidence—in my view they a) present this interpretation too strongly and without qualification, and b) do not adequately acknowledge or address a series of alternative interpretations of the results. Here are some of those alternatives that I believe should be ruled-out, discussed, and/or acknowledged.

Looking time and reevaluation

Is looking time reflecting a reevaluation process (that impacts choice) or it is merely predicting choice independently (i.e., revealing the outcome of a decision process that does/did not depend on looking time)? Making a case for this distinction requires decoupling the linear dependency between looking time and choice. For instance, earlier work used a model (DDM) non-linearly linking viewing time and stimulus value to choice. Another option—though likely beyond the scope of this study—would be to systematically manipulate the stimulus during (or before) the viewing time.

In addition, it would be useful to assess the extent to which looking time during the delay is a function of various factors other than choice, including looking time during the offers, last fixation, absolute offer values, offer magnitude and/or probability (which impact visual appearance of offer), etc.

OFC activity as memory reactivation

The fact that the offer stimuli and choice targets share the same positions creates a potential confound: pre-choice gaze and OFC firing may reflect a plan/anticipation of the upcoming choice, rather than some deliberative process about the stimulus. This would be a minor point—indeed most previous studies of OFC have this same feature—except that the authors make repeated and strong claims that the OFC activity represents a "reactivation" of a memory for the stimulus. (In fact, this is a major thrust of the paper.) By interpreting the activity as "memory", the authors are committing to the retrospective, deliberative process over the prospective, anticipatory one. In my view, either the interpretation should be made significantly more circumspect, or the authors should offer additional evidence and/or logical argument to support this strong interpretation.

As an aside, in future experiments, one might consider orthogonalizing stimulus and choice targets. For instance, the choice targets could appear above and below the fixation point, and/or the mapping choice target to spatial location could be randomized, such that if offer1 is presented on, say, the left, the choice target for offer1 would be randomized to either the left or right of (or above or below) the fixation point. This way the animal could not prepare an eye movement, or anticipate the value of a spatial location, prior to the choice phase. An additional step would be to use a button/joystick to effect the choice rather than an eye movement and spatial location.

OFC activity as reevaluation and input to the decision process

How do we know reactivation reflects reevaluation? The authors suggest as evidence the correlation between spiking activity and choice. This particular form of correlation has been studied widely (under the banner of “choice predictivity”) and hotly debated as to the extent to which it reflects a causal role in the decision process vs. reflecting feedback from a decision process that occurs elsewhere and independently. (For instance, see Nienborg, H., Cohen, M. R., & Cumming, B. G. (2012). Decision-related activity in sensory neurons: correlations among neurons and with behavior. *Annual Review of Neuroscience*, 35, 463–483.) If the claim of (putative) causality is to be included, the ambiguity around this interpretation of choice predictive activity should also be featured prominently.

Putting aside the interpretation of choice predictive correlation, I did not see what I expected to be the essential correlative analysis to support the hypothesis that gaze-dependent OFC activity represents deliberation/re-evaluation: controlling for the known, observable factors—task conditions (EVR/L, dEV, EVR+L), gaze (during offer1/2, epoch 1/2), pre-delay firing rate—is the remaining (i.e., unexplained) variance in choice explained by OFC activity during delay 2? And—to support the strongest form of the hypothesis—is this relationship conditional on concurrent gaze?

Unfortunately, to my read, the data as presented contradict the strong form of the above hypothesis. Specifically, in Figure 4D, bottom row, delay2 epoch, the percentage of neurons significantly encoding choice is not statistically different for R vs. L choices. If choice predictive activity (CP) during epoch2 reflects gaze-dependent reevaluation, then we should see evidence of that here: CP for R should be greater for LookR than LookL, but it's not. Perhaps I am missing something?

The authors state “variability in the size of this [gaze-dependent] reactivation predicts choice” (Line 421), which contradicts my above interpretation that CP is NOT modulated by gaze. Clarifying the basis for this interpretation is crucial.

The strong statements about reevaluation and reactivation occur throughout the manuscript and, in my view, should be qualified appropriately. A couple, but by no means all, examples include:

- Line 25: “This reactivation reflects a reevaluation process...”.
- Line 413: “gaze can reactivate neural responses that reflect the values of previously shown offers, presumably to facilitate their re-evaluation for the formation of a choice.”

****Clarity****

To the authors credit, the writing was generally clear and (I believe) I have a clear sense of what was done. However, I felt that it too long to get to the key and most compelling results, which risk being buried and underappreciated by the reader.

To me, the critical analysis concerns the eye movements and OFC activity related to offer1 that occurs during delay2 (e.g., Fig 3D, top panel, “LookL”, but even more precisely shown in Fig 3G, and discussed lines 318-322—a full 2 text-only pages into the OFC results!). As the authors correctly point out, this period is special because it is not contiguous with the viewing of offer1, and therefore cannot be explained by a simple form of “inertia” (i.e., autocorrelation) in eye position or OFC activity, to which the other analysis periods (e.g., delay1 related to offer1, delay2 related to offer2) are subject.

Why not bring this point out front and center? At present, the reader may miss this point at best, and at worst may become skeptical of the authors’ overall interpretation since it is a much weaker argument when focusing on the other periods.

Another example are the various regression models (often of limited interpretability—see comment re: use of separate models for each term) that gradually build until arriving at the key, appropriate model. For example, the behavioral analysis goes through many models before the full logistic model (lines 232-233). This is also true for the figures. Why not just show the critical and appropriate analysis in 2D, and omit the separate regressions in 2C?

****Theory****

Putting the questions of analysis and interpretation aside, in the end, I would have appreciated greater assimilation with existing literature and greater discussion of a theory for gaze-gated memory retrieval and evaluation. Why is it important to sequentially evaluate offers, and why use gaze as the means to do so? What problems does it cause/solve? How does gaze-gated memory retrieval interact with other circuits related to memory, evaluation, choice, and eye movements, such as ACC, HPC/MTL, dlPFC/FEF?

Minor Points

- The authors discuss why they used sequential presentation of offers, but it was still unclear to me the advantage over simultaneous presentation. If the key theory is that gaze during delay period gates evaluation of a specific offer, why not present both simultaneously, release fixation, eventually remove the offers, and then analyze gaze and neural activity both during the offer and delay periods? To be clear, I'm not suggesting that the sequential presentation is problematic, but since the authors (rightly) highlight it as a distinguishing feature, the pros/cons of this feature should be clear, as well.

- The task involves risky offers with variable magnitude m and probability p . The analyses all assume that subjects are computing the value of the offers as $m \cdot p$, or EV, and basing their decision on the difference in value, $dEV = EV_R - EV_L$. It would be useful to show that this is indeed the case.

- o That is, is there an influence of m or p linearly independent of EV? For instance, when m is high, do subjects favor the higher m offer, even when offer EVs are equivalent? For example, this kind of question could be addressed in Fig 1B by showing that dEV explains choices across a range of absolute m and p .

- o Even if we assume offer value is linear with EV, another question is how the 2 offers are compared. The authors assume choices are based on dEV . I would be useful to rule-out some alternatives.

☐ First, it has been shown widely that choices depend on the absolute value of the offers, in addition to the difference in values. Would be useful to show choices as function of dEV for a range of absolute EVs. Also, in the regression models, it would be useful to consider a model that includes terms for absolute EV (e.g., $\{EV_R, EV_L\}$ or $EV_R + EV_L$) in addition to dEV .

Surprisingly, in Fig 2C and related model, absolute values for EV_R and EV_L are used, but dEV is not included. Why is this?

☐ Second, given a fixed range of offers, there are certain offers that will very likely be higher/lower than the alternative, making consideration of the other offer unnecessary. And given a fixed range, these cases will correlate strongly with dEV , meaning that dEV may appear to be a significant factor, but it is merely correlated with single offer magnitude. (Note that this is not an either/or hypothesis. Subjects can consider both offers, but weight them disproportionately.) I am not implying this is the case, but it would be very useful to rule out this case. And not just for completeness. Indeed, the primary thesis is that subjects direct gaze at the 2 offers (or their former location) so as to consider both offers; therefore, it is crucial to show behaviorally the extent to which subjects are basing decisions based on each offer.

o Despite using EV for most analyses, the analyses in Fig 4 introduce a new term, SV, that depends on EV and offer variance. It is unclear why the authors switch to using SV for this analysis.

o Also, the SV term depends on fitting 2 parameters, w_1 and w_2 , to the data, which introduces additional uncertainty. Moreover, because SV is a fit term, for unbiased results, cross validation should be used: w_1 and w_2 should be fit on one set of trials and the betas (B1-3) should be estimated on a separate set of trials. Again, it may be simpler to stick with EV (if indeed EV is justified; see above).

- For all behavioral logistic choice curves (e.g., Fig 1B), it is unclear what the points represent.

o Are these individual trials? If so, for which session and which monkey? And if so, why are there so few? And if so, they we need some summary across behavioral sessions, such as a histogram of logistic fit parameters.

o If these are summary data, then it would be useful to show (some) individual sessions and a histogram across sessions to get a sense of the between-session variance.

- I was unclear what to learn from Fig 1D that wasn't shown in Fig 1E (which is the more useful of the 2). Moreover, I wonder if 1D could actually lead to confusion. That is, since all trials contribute, we don't know if what appears to be fixation of both offers is occurring within a single trial (as I think the authors would like to claim), or whether different trials (e.g., chose L vs chose R trials) have systematically different eye positions (e.g., fixation of only 1 of the 2 offers). This is where 1E, or an alternative (see below), is useful.

- I was interested in a slight variation of Fig 1E where the eye position is split by trials in which the subject ultimately chose L vs chose R, and ideally matched for dEV (or matched based on $p(\text{Chose R}) \pm \text{some epsilon}$). This would tell us about how eye position relates to the ultimate choice (i.e., the internal process), rather than how it relates to dEV (i.e., the external parameters). I'm not sure if my suggestion would be in addition to or in place of 1E.

- Some colors are difficult to distinguish, including

o Fig 2B, lines and points

o Fig 2C, σ_L and σ_R .

- The Discussion highlights 3 major findings re: OFC activity, but I wasn't sure which results supported the 2nd finding: "(2) gazing to an empty region where no stimulus has been presented washes out that value encoding" (line 419). (Same topic comes up again on Line 449—see Major Points above.) Apologies if I simply missed this. Or consider making the link more clear?

Reviewer #3 (Remarks to the Author):

Ferro and colleagues

This is an interesting manuscript that tests a clearly-specified and striking hypothesis. Ferro and colleagues consider how choice values are compared during decision making. First, they identify neurons in orbitofrontal cortex (OFC) in which activity covaries with choice value. They show that these neurons encode the value of each of two potential choices when they are shown to macaques but that they do so while the macaque is looking at them. Intriguingly, they re-encode those specific choice values when the animals are returning their gaze to the location in which the choice offer had been presented even if it is now no longer visible. The authors argue that the animals are reactivating one choice representation at a time in a manner that is guided by attention where the focus of attention continues to be specific and relatively exclusive even when the stimulus is no longer visible.

The results and the data are analysed carefully. A number of additional alternative analyses are presented in the supplementary materials. In general, the analyses support the authors' arguments. I was, however, unsure about one critical piece of evidence (although I think that are ways in which this might be addressed)

Main points

1. Lines 370-383 and figure 4a and 4c. Ferro and colleagues use a clever test to assess whether neural activity that occurs during periods when animals are looking at an empty screen, where an offer had been presented but is no longer visible, is related to the choice that animals will make at the end of the session. This is an important test because it is asking whether the neural activity encodes the choice the animal will make over and above the value of the potential choice; if this is the case then the neural activity is likely to be contributing to the decision process. The authors can therefore use this approach to ask whether activity that occurs when animals look at one empty location or another is related to the choice the animals will later make.

The activity in panel 4a suggests that the OFC activity during “empty” intervals does predict the choice the animal will later make (above chance level of the black line labelled ChR in 4a in the delay interval). However, it is only in figure 4c’s bottom panel that the activity is analysed as a function of where the animal is gazing. The grey line reflects encoding of whether a choice will be made to the right (ChR) when the animal is gazing to the right. However, this grey line seems no higher than the black line which corresponds to trials when the animals were looking in the opposite direction. If I understand correctly this is what a strong interpretation of the authors’ hypothesis would predict.

I think there might several ways of dealing with this. One might simply be to tone down the claims made and to be clearer in the text if there is a limitation in the strength of the evidence. Perhaps delay-related activity does predict choice but not in a manner that is related to the direction of gaze. Perhaps the authors could reconsider the manner in which they separate out trials into the black and grey traces in the bottom panel of 4c. If I understand correctly they are separated on the basis of whether the animal looks more one way than another but perhaps a more exacting criterion could be used such as focussing on trials when animals spend more than 50% as much time looking one way than the other. Or, alternatively, perhaps the authors could focus on the final part of the last delay and condition the analysis on that period.

Minor point

“Further, to test whether evaluation is sequential by default, we let subjects to move their eyes at will, and expect tracking the internal deliberation process by using the look-at-nothing effect ...” sounds a bit awkward. Maybe it should be something like: “Further, to test whether evaluation is sequential by default, we let subjects move their eyes at will, and expect to be able to track the internal deliberation process by using the look-at-nothing effect ...”

RESPONSES TO REVIEWER COMMENTS

We would like to thank all the reviewers for providing very highly valuable suggestions and for dedicating their time in reviewing our manuscript.

To simplify the reading of following responses, we will recall here the structure and numbers for different sections of the main manuscript text:

Introduction

Results

- R1. Performance and look-at-nothing gazing in a gambling task*
- R2. Gaze position modulates choice during look-at-something and look-at-nothing*
- R3. Look-at-something and look-at-nothing value encoding*
- R4. Reactivation of value encoding during look-at-nothing correlate with [...] look-at-something*

Discussion

Methods

- M1. Behavioral task*
- M2. Behavioral data analyses*
 - M2.1 Behavioral performance*
 - M2.2 Analyses of eye position and gaze shifts*
 - M2.3 Probability of choice as a function of offer expected values and looking times*
- M3. Neural data recordings*
- M4. Analyses of neural data*
 - M4.1 Neural spiking and the encoding of reward expected value*
 - M4.2 Difference in encoding EV_L versus EV_R as a function of time, and [...] LookL versus LookR.*
 - M4.3 Neural encoding at delay times conditioned on previous looking side*
 - M4.4 Neural encoding of offer subjective value and the choice*
 - M4.5 Correlation of SV regression weights*
 - M4.6 Analysis of regression residuals*

Data availability

Code availability

References

Acknowledgements

Author contributions

Declaration of interests

All responses are marked by a left vertical bar and indent spacing to the left side. We have included some figures in this document for direct response, and we have indexed them as Comment Figures.

Reviewer #1 (Remarks to the Author):

Ferro and colleagues report behavior and neural recordings in the OFC of macaque monkeys performing a value-based decision task used in many prior studies from this group. In each trial, the monkey is presented with two decision offers that vary according to the magnitude and probability of juice reward. The offers are presented one at a time on opposite sides of a computer screen, after which the monkeys are prompted to indicate their chosen offer with an eye movement. Most relevant to this study is the fact that between offer1 and offer2, and between offer2 and the choice prompt, there are 600ms delay periods in which the screen is blank. The main claim of the study is that during the delay periods, the monkeys frequently look at the locations where the offer stimuli previously appeared, and that doing so “reactivates” neural encoding of the values of the offers previously at those locations. An additional claim is that this gaze-triggered activity is a form of “reevaluation” of the offers, given that during the delay periods neural activity also predicts upcoming choices. If true, these claims would suggest a new and important function for OFC during decision-making, i.e., the computation of spatially-based memories for value.

However, it is not clear from the presentation of the data that these claims fully hold up to scrutiny. These issues could potentially be resolved with additional analyses and/or adjustments to existing analyses, as suggested below:

We would like to thank Reviewer 1 for the so many insightful comments and proposals. We have addressed all of them in detail below, thanks to which we believe have improved the quality of the manuscript.

1) The first and strongest claim of the study is that viewing an “empty” offer location during the delay reactivates a representation of the value of the offer. Figure 3D and E show this by dividing trials, bin-by-bin, according to whether the gaze was detected on the left or right of the screen, and then counting the fraction of cells significantly encoding the value of the offer in a linear model. This approach appears to have two weaknesses.

The first is that whether or not a cell is significant depends in part on the number of trials used to fit the model. Because gaze location varies as a function of time in the trial (Figure 1C), the dynamics in the fraction of significant cells could be attributable in part to the changes in gaze location, rather than changes in neural encoding.

We agree with the reviewer that the number of trials can affect the assessment of significant encoding of each cell, and thus the number of significant cells in each time bin. However, our conclusions are not affected by this, as described in the following.

First, we provide results of an analysis demonstrating that the different numbers of trials do not spuriously cause our results. The analysis appeared in the earlier version of the manuscript but eventually hidden in the supplementary information (Supplementary Figure S7A-D; was previously Supplementary Figure S9). This analysis is analogous to Fig. 3B, D but with an important

difference: while in Fig. 3B, D the number of trials used to compute the fraction of significant cells varies based on whether subjects look most to the right or left, in Supplementary Figure S7 the same number of trials is used in both conditions. As it is evident, the patterns in both figures are the same, especially during delay 2 (the most relevant epoch for reactivation), but also in the other epochs (though with shorter significant encoding runs), indicating that the effects that we described in Fig. 3B, D are not an artifactual consequence of using an unbalanced number of trials.

Second, we note that using an unbalanced number of trials works against our hypothesis, not for it. In other words, this analysis risks Type II errors, but not Type I errors. Specifically, for the analysis shown in Fig. 3B, 3D the reactivation of encoding of EV_L during the *delay 2* epoch is found when the subjects look at the left screen side (*LookL*), which is precisely when the least number of a trials is available (as the subjects tend to look more on the right during that period, since it is the side where the second, most recent offer has been presented). In parallel, when the animal looks at the right, we do not see encoding of the EV_L , even though in this latter case we use more trials. Therefore, these two core results cannot be accounted for by an unbalanced number of trials – though stronger results could presumably be obtained if we had an equally large number of trials in both conditions. We have addressed this further in the 7th paragraph of Results Sec. *R3* in the main text.

Finally, we performed epoch-wise comparisons in the fraction of significant cells encoding EV_L depending on whether the subjects look at the left or at the right (Fig. 3E). This analysis does not need to have equal number of trials per condition: testing a difference between two fractions of significant cells does not require that each fraction is computed using the same number of trials (e.g., a t-test on the difference between two means does not require equal number of trials to compute the means and test their difference; in other words, an equal number of trials affects the variances, but not the means, thus not introducing false positives). We have now commented on trials availability and significance assessment in Methods *M4.1*. In the following of this response, we provide further elaborations about this aspect, including Comment Figure 1, but it can be safely omitted in a first reading or if the previous description is clear.

In conclusion, our core results are robust to analyses where we equate the number of trials (by subsampling, factually leading to results relying on poorer statistical power) and are independent on the unequal number of trials available for the two looking sides.

Supplemental information about significance of linear model weights and number of trials

In Fig. 3 and Fig. 4, we assess the fraction of significant cells based on *F*-Statistics of the linear model of each cell (Methods *M4.1*). Specifically, we test the hypothesis that the spike count η (in 200 ms boxcar windows) fit to a linear model for EV (in Fig. 3, SV in Fig. 4) of each offer has slope $\beta_1 \neq 0$ against the null hypothesis that $\beta_1 = 0$ (constant model).

The fit of the model $\eta = \beta_0 + \beta_1 x$ (with $x = EV_L$ or EV_R respectively), is applied through Least Squares (LS) estimation, i.e. deriving unbiased the estimator $\hat{\beta}_1 = \frac{\sum_i (\eta_i - \bar{\eta})(x_i - \bar{x})}{\sum_i (x_i - \bar{x})^2}$, with mean matching true mean $E[\hat{\beta}_1] = \beta_1$, and whose variance $V[\hat{\beta}_1] = \frac{\sum_i (\eta_i - \bar{\eta})^2}{(n-2) \sum_i (x_i - \bar{x})^2}$ decreases with n , the number of trials. We observe that using this kind of estimator over lower numbers of available trials can increase the chance of getting larger *p*-values, hence leading to higher probability of failing to reject the null hypothesis that β_1 is zero when the alternative hypothesis is true, i.e., when β_1 is significantly different from zero (type II errors, false negative or missed detection). At the same time, this estimator does not suffer from false positive (type I) errors. This can be shown by applying multiple runs of the same regression to variable numbers of observations (Comment Figure 1).

Comment Figure 1: Summary statistics of linear interaction slope (β_1) with constant term (β_0) for different numbers of observations n (shown in log-scale on the x-axis). **A.** From above, we show mean $E(\beta_1)$, Standard Error $SE(\beta_1)$, t-statistics $t = E(\beta_1)/SE(\beta_1)$, p-value and assessments of significance as occurrences of $p < 0.05$. In all cases, we report mean over 1000 least-squares estimation runs and 5-95th percentiles of their distributions (shaded areas). Here we set $\eta = \beta_0 + \beta_1 x + \varepsilon$, with $\beta_0 = 1, \beta_1 = 0, x \sim N(0, 1)$ and $\varepsilon \sim N(0, 10)$. Note that in the bottom panel, the fraction of runs with false positives (where we erroneously detect $p < 0.05$) is fixed at the threshold value 0.05 even for lower number of trials. **B.** Same as A, but for $\beta_1 = 1$ to show the occurrences of false negatives, in bottom panel. **A-B.** It can be shown that the trends shown do not vary for different values of std. of x and ε , nor with different number of runs. We show a disadvantaged case with std. of noise 10 times larger than the std. of the linear term.

As is evident from the bottom panels in B, this model is susceptible to false negatives (missing detection of actual linear interactions) in lower numbers of available trials. Yet, as for bottom panel in A, the false positives (detecting spurious interactions) do not vary with the number of observations.

We have added considerations about false negatives / false positives at the end of the first paragraph of Methods *M4.1*.

The control analysis in Supplementary Figure S9 addresses this issue by mandating that equal numbers of trials are used in each time bin. However, it appears that several of the results are less robust when this control is performed, e.g. the dark green line during delay1, and the light blue

line during delay 2. It would be helpful to see a version of Figure 3E that uses equal-trial-number data, in order to assess which effects hold up under this control. Additionally, there needs to be an explicit discussion of this issue, and acknowledgment of any results that differ between the main analysis and this control.

As suggested by the reviewer, we have reproduced Fig. 3E with an equal number of trials (subsampling to even trial numbers) in Supp. Fig. S7E-F (previous Supp. Fig. S9 mentioned by the reviewer to which we have added panels E-F). The results are in alignment with Fig. 3E (not updated, same as before revision): significant reactivation of EV_L when LookL during delay 2, encoding of EV_R only when LookR and not when LookL during offer 2 and delay 2. The main differences are that here the encoding of EV_L is weaker in offer 1, LookL (B top, n.s.), and that comparison of EV_L encoding between LookL and LookR is no longer significant in delay 2 (B top, n.s.), although the expected tendency is observed and only there is significant EV_L encoding in the LookL condition. Both differences could be previously expected by inspecting Supplementary Figure S7A-D (same as before revision) after combining all time bins and can be attributed to a lower number of trials used to perform the analysis with equal number of trials per looking condition.

The second weakness is that other variables, such as EV or risk, could be correlated with gaze allocation; this is not an unreasonable assumption, given prior studies showing that gaze is influenced by the values of visible targets[1–3]. If this were true, stratifying trials into LookL / LookR would also be stratifying by another variable. As a result, even when the trial numbers are equal, it would be difficult to rule out the possibility that the effects of gaze direction are explained by other variables.

We thank the reviewer for this comment. Note that when we separate trials by *LookL* and *LookR* into two sets, within each of the sets we have a finite range of available EV_L and EV_R to perform our analysis conditioning on looking side, although this range can of course be unequal in each condition due to the correlation between *EV* and gaze that the reviewer mentions. Our point is that thanks to the sizable value ranges we can still study the significance in linear encoding of EV_L and EV_R in the activity of OFC neurons conditioned to gaze location. Note also that when we use subjective values instead of *EV*, we actually include the effect of risk, among other factors.

In addition, in all cases where we apply comparisons between *LookL* and *LookR* we compare empirical fractions of significant cells to bin-wise trial-order shuffled fractions of significant cells (better detailed in fourth paragraph of Methods *M4.1*). By doing so, we always compare the empirical significance of *LookL* and *LookR* trial pools within each bin to the significance of the same trial pools but shuffled in order in that specific bin. This ensures that though there is a finite range of available EV_L or EV_R , their significance is still assessed based on the same range both for empirical data and shuffled data.

We added citation to [1] Yasuda et al., *J. Neurosci.* (2012); [2] Cavanagh et al., *PNAS* (2019) and [3] McGinty et al., *Neuron* (2016) as we deem very relevant their findings about value-based visual guidance in the first paragraph of Introduction.

2) Figures 3F and G are extremely important for the main claim that gaze onto empty offer locations activates representations of value. The idea behind these analyses is to divide trials into those where the gaze did vs. did not transition away from the offer during the delay period.

However, it would appear that this analysis seems to suffer from the same issue regarding the unequal numbers of trials in the two groups, given that the analysis performed here is the same as in Figure 3D. These analyses could be repeated with the same procedure as in Figure S9, to resolve this issue.

We thank the reviewer for this important comment. Many of the same arguments as above apply here as well. Specifically, the difference in number of trials can create Type II errors, but, given the structure of our analysis, does not lead to Type I errors.

In any case, we have repeated the analysis of Figures 3F and 3G by subsampling to achieve even trial sizes (as in previous Supp. Fig. S9, now Supp. Fig. S7) and reported the results in Supp. Fig. S8 F-J. As expected, with reduced number of trials, some effects are less evident (this is necessarily true), but, importantly, the crucial result that there is reactivation of encoding of EV_L in *LookL* during delay 2 after subsampling (i.e., by using the same number of trials as in *LookR*), is still present (panel J, left). Some other effects are affected by the lower availability of trials: we find fewer encoding of EV_L for *LookL* in delay 1 and fewer encoding of EV_R for *LookR* in delay 2, but this is due to reduced trials availability (we have almost zero trials to use at the beginning of the delay, as shown in panels G-H).

This result could be expected when considering the effect of trials availability on our data: since we found encoding of EV_L in *delay 2* for *LookL*, which is already the looking condition with fewer trials available, equating the number of available trials in *LookR* to the number of trials available in *LookL* leaves our *LookL* results almost unchanged (compare Supp. Fig. C with H, and E with J).

Further considerations about trials availability and encoding of EV s during delay epochs are also provided in response to next Comment 3.

We have added Supp. Fig. 8, and we describe these results in the 7th paragraph of Results R3.

3) Related to the above: Figure 3F and G imply that when gaze transitions from one side to the other, encoding in OFC neurons changes to reflect the gaze location. However, this is only implied and is not directly shown, because at no point are any analyses performed with data time referenced to the gaze transitions (i.e. saccades from one side of the scree to the other).

One way to correct this issue, and to provide strong support for the main claims of the study, would be to show that changes in OFC encoding are reliably time-locked to gaze transitions, as has been shown for visible targets in other studies[4]. Note that this kind of analysis will also cleanly address the two weaknesses in comment 1 above, because the critical comparison will be the encoding that occurs before vs. after the gaze transition over the same fixed set of trials – rather than the encoding observed between different sets of trials (*LookL/R*) that could be unequal in number and that could differ by some confounding variable.

We deeply appreciate the comment and the analysis suggested by the reviewer, which is indeed very sharp. Although the timing of gazes is extremely variable and the gazing patterns are highly heterogeneous, we have been able to perform the proposed analysis in Supp. Fig. S14. We are pleased to report that it goes in the predicted direction.

Specifically, we have estimated the timing of gaze shifts (defined in last paragraph in Methods M2.2, shown in Supp. Fig. S1C) from one side of the screen to the other (implying midline crossing). Thus, we

computed encoding of EV time-locked to the start of the first gaze-shift following delay time start, depending on the gaze shift side (from L to R, or vice versa) in both *delay 1* and *delay 2* periods. The results support our claims and reproduce our previous results:

- (1) In *delay 1*, before the gaze shift from L to R (Supp. Fig. 14D left, light orange line), there is encoding of EV_L during brief periods of time, presumably indicating that subjects previously visually sampled / encoded the L offer, but soon after the shift to the R side, encoding decays to non-significant range;
- (2) In *delay 1*, there is no encoding of EV_L if subjects shifted the gaze from R to L (Supp. Fig. 14D left, dark orange line), presumably because subjects did not previously visually sample / encode the L offer;
- (3) In *delay 2*, there is no encoding of EV_L before the gaze is shifted from R to L, because the subjects are eventually not yet retrieving the L offer EV (Supp. Fig. S14E left, light orange line);
- (4) In *delay 2*, as expected, after the gaze is shifted from R to L there is encoding (reactivation) of EV_L (Sup. Fig. S14E left, dark orange line).
- (5) In *delay 2*, before the gaze shift from R to L (Supp. Fig. S14E right, dark purple line), there is encoding of EV_R , which then decays as soon as subjects gaze away from most recent offer presentation site.

In conclusion, we show that gaze-shift time-locked analyses can reproduce the main results shown in the main text (Fig. 3F-G).

Finally, note that the number of trials is not perfectly balanced before and after the gaze shift (panel G, H) in all time-bins for this proposed analysis, because the gaze can happen late in *delay 1* or *delay 2* periods, reducing the number of available trials time-locked to gaze shifts as time progresses (*delay 2* expires and subjects reacquire fixation). Going beyond this analysis by subsampling trials once again would inexorably lead to loss of statistical power.

We have included the gaze-shift alignment in Methods *M4.3*, described these results in the last paragraph of Results *R3* and added the Supp. Fig. S14, where we show delay time results time-locked to gaze-shift onset for trials with gaze shifts (Supp. Fig. S14A-E) and time-locked to delay start for trials without delay time gaze-shifts (Supp. Fig. 14F-J).

We now mentioned [4] Hunt et al, Nat. Neuro. 2018 in the second paragraph of the Introduction for the relevance of their findings on choice-related prefrontal cortex activity time-locked to gaze transitions (the same work was already previously cited in current third paragraph of Discussion).

4) The second major claim is that “variability in the size of [gaze-based] reactivation predicts choice” (line 421-422). It is not clear which data support this claim. Figure 4A shows that, after accounting for variability due to the subjective values, variability in spiking also predicts choice. And Figure 4C shows that after splitting trials according to gaze position, this effect still holds, though perhaps at a weaker magnitude given that fewer significant cells are detected. But these data show only that gaze-based reactivation and choice-predictive activity are both detectable in the same model at around the same time. To show that variability in reactivation predicts choice requires an analysis that computes some kind of single-trial measure of reactivation, and then shows that this measure correlates with choices on a trial-to-trial basis. For an example of an analysis with similar goals in single cells, see the choice probability analyses of Conen and Padoa-Schoppa (2015)[5]; for an example in neural populations, see McGinty & Lupkin (2021)[6].

We thank the reviewer for this important comment. The fact that we find encoding of the choice variable (Fig. 4C, bottom) regardless of whether animals *LookR* or *LookL* tells us that fluctuations of activity predict choice – otherwise, fluctuations of activity during *delay 2* beyond what can be predicted by EVs would have not shown significant correlations. However, the reviewer here makes a very good point: is the sign of the fluctuations (reactivation) in the expected direction, that is, aligned with tuning of EV encoding cells?

To address this important question, we have performed two additional analyses in the revised manuscript (i, ii).

i) If reactivation implies reevaluation in *LookL* trials, then when the animal chooses L, we should see more fluctuation in activity of neurons with positive tuning to SV_L , and less activity in neurons with negative tuning. In other words, a positive fluctuation of activity of a neuron should increase the probability of the L choice if the tuning of the neuron is such that an increase of in SV_L leads to an increase of rate (positive modulation). We define fluctuations are the residuals $z_i = \eta_i - (\beta'_0 + \beta'_1 SV_{L,i} + \beta'_2 SV_{R,i})$ at the i^{th} trial, and study the relationship between β'_1 and choice. Note that SV s are now defined as $SV = \beta_1 EV + \beta_2 m + \beta_3 \sigma^2$; following suggestion by Reviewer 2, we have now included the magnitude among relevant regressors, corresponding to the best behavioral model to predict choice (Supp. Fig. S4).

Our prediction is confirmed by the results (Fig. 4F). We predict that neurons with positive tuning and positive activity fluctuations better predict L choices, and that neurons with negative tuning better predict opposite, R choices, when there are positive activity fluctuations. To simplify the visualization, we combined data based on the choice side (L, R) and the sign of β'_1 by defining ‘preferred choice’ (positive β'_1 and choice=L or negative β'_1 and choice=R) and ‘non-preferred choice’ units (positive β'_1 and choice=R or negative β'_1 and choice=L), analogous to what is done in the choice-probability literature. By comparing residual magnitudes in the two data pools, we find significantly larger residuals in preferred choice pools, confirming that the sign of fluctuations of residual activity during reactivation correlate with choice, and in the predicted direction. Since for every neuron we have both preferred and non-preferred choices, we used paired Wilcoxon tests, yielding significant difference ($p=0.00021$) for *LookL*.

In Fig. 4F (right) we show as a control, that repeating the same analysis in *LookR* trials leads to a non-significant relationship between noise fluctuations when aligning SV_L tuning with choice, in line with the lack of reactivation during *LookR*.

ii) Following the same idea as before, we can ask the same question but correlating the regressors of the linear model $\eta = \beta_0 + \beta_1 SV_L + \beta_2 SV_R + \beta_3 ch$ for each unit, where choice is encoded as $ch = +1$ for R, and -1 for L. Then, following the same reasoning as above, we expect that in *LookL* trials in *delay 2*, a neuron with positive β_1 would be associated with a negative β_3 (that is, more positive firing modulation to SV_L correlates to more negative modulation with choice). To avoid spurious correlations between regressors trained on the same dataset (Arandia et al., Neuron 2016)^[b], we estimated $\beta_1^{(1)}, \beta_3^{(1)}$ by using the first half of the trials, and $\beta_1^{(2)}, \beta_3^{(2)}$, by using the second half of the data, and then computed the correlation between $[\beta_1^{(1)} \beta_1^{(2)}]$ and $[\beta_3^{(2)} \beta_3^{(1)}]$ (combing β_1, β_3 in disjoint sets). The results are shown in Fig. 4E, confirming the prediction of a significant negative correlation in *LookL* trials, which is absent in *LookR* trials, as expected.

We have described these new important results in Fig. 4E-F, in the last paragraph of Results *R4* and described the Methods in *M4.5, M4.6*.

We have added suggested references:

[5] Conen, K. E., & Padoa-Schioppa, C. (2015). Neuronal variability in orbitofrontal cortex during economic decisions. *Journal of neurophysiology*, 114(3), 1367-1381.
 [6] McGinty, V. B., & Lupkin, S. M. (2021). Behavioral read out of population value signals in primate orbitofrontal cortex. *BioRxiv*, 2021-03.
 In the last paragraph of Results R4.

And added new references:

[a] Britten, K. H., Newsome, W. T., Shadlen, M. N., Celebrini, S., & Movshon, J. A. (1996). A relationship between behavioral choice and the visual responses of neurons in macaque MT. *Visual Neuroscience*, 13, 87–100.
 [b] Arandia-Romero, I., Tanabe, S., Drugowitsch, J., Kohn, A., & Moreno-Bote, R. (2016). Multiplicative and additive modulation of neuronal tuning with population activity affects encoded information. *Neuron*, 89(6), 1305-1316.
 In the last paragraph of Results R4 and in Methods M4.6.

5) A 95th percentile significance threshold seems too lenient, even given the a priori expectation of greater-than-chance encoding. This is evident in the false positives in the pre-offer1 epochs in both Figures 3 and 4. I understand that some sort of cluster-based threshold was used to find minimum run lengths for significant effects; but this also seems too lenient, given the many extremely short epochs of significant encoding in Figure 3. The use of this lenient threshold for the time series data might be acceptable, in order to show the dynamics and general trends in the data. But for the data in Figure 3C and E, and Figure 4B and D, more needs to be done to confirm the robustness of the effects. I would recommend first documenting the raw p-values of the filled bars in these graphs (i.e. the fraction of shuffled instances greater than the observed). In addition, to address multiple comparison issues, I would strongly recommend applying some form of p-value correction to these, and reporting corrected p-values alongside the raw p-values.

We thank the reviewer a lot for the proposed adjustments. We report the raw *p*-values, along with multiple comparison correction in the following two tables CT1-2. The adjustment method is based on False Discovery Rate (FDR) correction at acceptance level 0.05 by Benjamini-Hochberg, 1995, now included in Methods.

In addition, we now report dotted lines showing 95th percentile of fractions of significant cells from data with bin-wise shuffled trial order in Fig. 3C, 3E and Fig. 4B, 4D. The final numerical *p*-values (FDR corrected) for Fig. 3C, 3E are now in Supp. Table ST6, and for Fig. 4B, 4D, updated to further changes in SV computation (now defined by including magnitude regressors) in Supp. Table ST7.

We have also improved the statistical assessment of run length analyses (6th paragraph of Methods M4.1).

	p -values (Wilcoxon signrank)					Benjamini-Hochberg (FDR=0.05) corrections				
	offer1	delay1	offer2	delay2	ch-hold	offer1	delay1	offer2	delay2	ch-hold
EV_L vs EV_R	7.62e-4 ***	7.72e-8 ***	0.2519 -	3.70e-4 ***	0.2466 -	0.0013 **	3.86e-7 ***	0.2519 -	9.25e-4 ***	0.2519 -
EV_L LL vs LR	6.91e-4 ***	1.08e-7 ***	0.1419 -	0.0314 *	0.0141 *	0.0017 **	5.40e-7 ***	0.1419 -	0.0392 *	0.0235 *
EV_R LL vs LR	0.1902 -	0.3638 -	8.10e-6 ***	6.88e-7 ***	4.82e-4 ***	0.2378 -	0.3638 -	2.03e-5 ***	3.44e-6 ***	8.04e-4 ***

Comment Table CT1. Exact p-values for Fig 3C, 3E. *LL* stands for *LookL*, *LR* for *LookR*. The *p*-values are corrected via Benjamini-Hochberg correction over each row. Changes in stars of *p*-values stars are highlighted in blue.

	p -values (Wilcoxon signrank)					Benjamini-Hochberg (FDR=0.05) corrections				
	offer1	delay1	offer2	delay2	ch-hold	offer1	delay1	offer2	delay2	ch-hold
SV_L LL vs LR	0.0014 **	4.85e-6 ***	0.0635 -	0.0024 **	0.0017 **	0.0028 **	2.42e-5 ***	0.0635 -	0.0030 **	0.0028 **
SV_R LL vs LR	0.0257 *	0.0743 -	1.76e-4 ***	4.84e-8 ***	3.41e-5 ***	0.0321 *	0.0743 -	2.93e-4 ***	2.42e-7 ***	8.52e-5 ***
ch. LL vs LR	0.4563 -	0.3293 -	0.7563 -	0.5528 -	0.3425 -	0.6910 -	0.6910 -	0.7563 -	0.6910 -	0.6910 -

Comment Table CT2. Exact p-values for Fig 4D. *LL* stands for *LookL*, *LR* for *LookR*. The p-values are corrected via Benjamini-Hochberg correction over each row. No changes in stars of *p*-values stars are found. Note: in most updated version of Fig. 4 and in Supp. Table ST7 these values have been updated as the SV have been updated from $SV = \beta_1 EV + \beta_2 \sigma^2$ to $SV = \beta_1 EV + \beta_2 m + \beta_3 \sigma^2$.

6) The criteria used to designate LookL and LookR trials are not robust, and could easily be improved. The current criteria are simply whether the average gaze position was left or right of the screen center. This means that instances where the fixation is virtually at the center, or where the monkey is shifting his gaze across the center are going to be arbitrarily assigned as LookL or LookR – thereby adding variability to the results. This might be an appropriate criterion for the brief re-fixate and choice-go epochs, because gaze mostly seems to be at the center at these times (Figure S2); but this criterion seems inappropriate for the other (much longer) offer and delay epochs, where gaze is concentrated on the left and right offer locations. For finding LookL/R trials during the offer and delay epochs, it would be best to set some non-zero positive and negative boundaries around the center, so that at-center fixations are not assigned to LookL/R, and are either discarded or analyzed separately.

We thank the reviewer for the suggestion. As the reviewer notes, this is not an issue that is likely to cause us to draw erroneous conclusions – if we were overly generous in assigning gaze directions, it would lead to Type II but not Type I errors. In any case, we can confirm that this analysis approach does not cause much of an effect either way. Specifically, we have now performed the analysis suggested by the reviewer: repeating the analysis of Fig. 3C and 3E by removing an area around the midline of the screen. As we show in Supp. Fig. S13, our results are robust to this removal, despite losing some trials in this process. We have included this important control analysis as Supp. Fig. S13, mentioned it in the main text (in the last paragraph of Results R3) and described the Methods in M4.2.

7) Line 211/Figure 2B: The claim is that the slope of the lines in Figure 2B are higher for the EV_R > EV_L condition, because these are trials in which the right offer is more valuable. But wouldn't having a higher right offer value make the *intercept* higher, reflecting an overall greater probability of choosing right? The increased slope means that in R>L trials, relative gaze time has a

greater effect on the choice outcome – it doesn't mean that right choices are more likely. Related: In Figure 2B, isn't the expectation that the lines during the choice-hold period be non-overlapping, reflecting the different relative values of the offers, and therefore the different probabilities of choosing right?

We apologize for the eventual confusion created by our previously proposed referencing in Fig.2B, and we agree that reviewer's proposal is more reasonable (we previously did not flip EV1 to the left side in trials where first offer is on the right side). We have updated all panels in Fig. 2B with the proposed representation. All other figures are unchanged as this more intuitive referencing was already used in all figures (including in the remainder panels of Fig. 2). We have updated the second paragraph of Results *R2* and updated to this change the second paragraph of Methods *M2.3*.

7) Some additional details on the neural recording, sorting, and pre-processing would be welcome. First, can the authors confirm the use of single contact electrodes? If so, how many were used per session? If 1-3 neurons are isolated per electrode, this would imply using ~30 electrodes to record 59 neurons (monkey 1, session 2), which seems like an extremely large number of individual electrodes to insert/remove in each session. Were these chronic electrodes?

We thank the reviewer for pointing at this very important clarification. We used multi-contact V-probes (32 contacts each, 75 μm inter-contact spacing). The recordings were performed in acute sessions (not chronic), the number of cells isolated per contact could range 1-3, and the number of electrodes was either 1 or 2.

We have now detailed this information in Methods *M3*.

Is it possible that multi-contact probes were used in some sessions? More detail is needed. Second, electrode impedances are specified in mega-ohms or kilo-ohms, not in "MU" as written on line 642. Third, what were the isolation criteria used to accept/reject neurons for analysis, e.g. minimum unit amplitude relative to the noise, fraction of short ISIs, etc.?

Yes, multi-contact electrodes were used for all sessions. We now indicate this in the revised manuscript.

We acknowledge that "MU" was a typo, we have now fixed to $\text{M}\Omega$ (Methods *M3*).

We have used a standard laboratory pipeline for isolation, exactly as described more in detail in previous works (Strait et al., *Neuron* 2014; Wang and Hayden, *Nature Comm.* 2017; Yoo and Hayden, *Neuron* 2020) ^[c-e]. We never adopted a fixed quantitative criterion for unit amplitude vs noise because of variability across sessions. The sorting was made through Offline Plexon Sorter software (Plexon, USA) by expert researchers with at least three years of training apprenticed to senior lab members for at least two full projects. Note the very same approach to unit isolation and identification is taken on what is now about 20 electrophysiology papers from the Hayden lab in the past 12 years.

References to previous research from the lab:

- [c] Strait, C.E., Blanchard, T.C. and Hayden, B.Y., 2014. Reward value comparison via mutual inhibition in ventromedial prefrontal cortex. *Neuron*, 82(6), pp.1357-1366.
- [d] Wang, M. Z., & Hayden, B. Y. (2017). Reactivation of associative structure specific outcome responses during prospective evaluation in reward-based choices. *Nature communications*, 8(1), 15821.
- [e] Yoo, S. B. M., & Hayden, B. Y. (2020). The transition from evaluation to selection involves neural subspace reorganization in core reward regions. *Neuron*, 105(4), 712-724.

Fourth, line 652 mentions preprocessing steps including “thresholding and quantization”; what exactly does this mean? Does thresholding refer to a minimum firing rate threshold? Does quantization refer to temporal binning? More detail is needed.

We propose improved word usage in Methods *M4* first two lines by specifying that preprocessing was performed through Plexon software (Plexon, USA). The thresholding applied is for cells with lower firing rate than 0.6 spikes/s. We apologize for the eventual misleading usage of the word “quantization”, referring to temporal binning and digitization into binary sequences.

8) In an attempt to simplify the data presentation, right-first/left-second trials have been re-coded as left-first/right second trials, so that dissimilar trials can be analyzed in the same way. This is potentially confusing, and glosses over the fact that the OFC neurons have both spatial-based responses[7] as well as order-based responses in tasks with serial offers [4,6]. To avoid confusion, and make the results more aligned with conventions of previous studies, I think it would be better to be transparent and simply designate the stimuli as 1st offer/2nd offer, or something similar.

We thank the reviewer for this suggestion. We have debated the two possible naming schemes and reached to the conclusion that the first-L / second-R terminology is the simplest. This is mostly because our central analysis is based on moving gaze towards or away from a target, and to picture such a target, one always must visually locate it somewhere (either R or L). Therefore, we decided to aid to this by referencing as “presenting” the first offer always on the left side. We agree that this might cause confusion about visuospatial selectivity (that we do not find, as shown in Supp. Table ST2: first=L, first=R columns, Supp. Fig. S1, S2B), but we think this is the simplest way to convey our in-depth gaze-dependent analysis using some physical reference.

9) The authors report that the monkeys only look at the offers in ~80% of trials. What happens on the 20% of trials where the monkey does not look at the offer initially, but then goes on to look at the empty offer location later. Is the same “reactivation” signal seen, even though the offer wasn’t viewed in the first place? Likewise, are reactivation effects stronger when these 20% of trials are removed from the main analysis?

We thank the reviewer once again for this very insightful proposal. We now include Supp. Fig. S6, in which we condition whether the subjects look or did not look at the left offer side during offer 1 or delay 1. As expected, if subjects did not *LookL* during *offer 1* or *delay 1*, there is no encoding of EV_L (Supp. Fig. S6C, top). More importantly, and as predicted, this coincided with lack of encoding of EV_L regardless of whether the subjects looked R or L during *delay 2*. However, when we use only the trials where the subjects always looked at L during *offer 1*, we do see a strong encoding of EV_L during *offer 1* and *delay 1* (Supp. Fig. S6D, top). Further, during *delay 2* even strong reactivation is seen by removing the trials where subjects did not initially look L (and there was no initial encoding of EV_L).

We have reported these results in Supp. Fig S6 and included this in the main text (6th paragraph of Results R3). In the analysis, we remove trials when subjects did not look at the left during the last 200ms of offer 1 or delay 1 epochs. Similar results (not shown) are obtained when we remove trials when the subjects do not look left during the last 200 ms of *offer 1*, but we use this stronger conditioning to avoid possible memory effects during *delay 1*.

10) In supplementary Figure 8C, why do the shaded areas differ for the two conditions? Presumably this is due to differences in the number of trials in each condition and bin? This needs to be explained in detail to avoid any confusion as to what is happening here.

Indeed, the differences are due to the different number of trials used in LookR vs LookL conditions. We have clarified this in the caption of Supp. Fig. S12.

12) There is potentially an alternative interpretation of the behavioral and neural effects in the delay2 epoch. The data show that during this epoch, there is an offer 1 signal that appears while the subjects look at the location of the previously viewed target (which is not present). This is interpreted as being related to a reevaluation of the first offer as part of the ongoing decision process. However, there is an alternative explanation. There are some indications from recent studies in humans and NHPs suggesting that once an implicit decision has been reached, the eyes are often drawn to the location of the to-be-chosen target. See recent studies from Manohar[8] and Westbrook[9]. The idea is that late in a trial, gaze begins to reflect the latent outcome of the choice process. If such an effect were present here, then the gaze behavior and neural signals appearing late in the delay2 epoch could reflect representations of the chosen value, rather than reevaluation of a previously visited offer. This is a subtlety that should be addressed in the discussion section.

We thank again the reviewer for the comment. Given the new analysis provided above in this response, and the previous results in the main text, we find it difficult to reconcile that the activation observed during delay 1 encoding of EV_L when *LookL* and not when *LookR* is merely due to a motor preparation, although we acknowledge that it is impossible to argue that there is full absence of elements related to motor preparation or encoding of the value of the to-be-chosen offer. Our arguments about the fact that look-at-nothing reactivation does not merely consist in motor preparation build on the following observations: (1) The reactivation is about re-encoding of EV_L (or SV_L), which is not visually displayed at *delay 2* time (despite the presence of a later, 'distracting' offer is shown in the meantime) – suggesting that there is a memory component in the reactivation that is

independent of the motor plan. If *delay 2* reactivation was merely related to motor planning, it could only consist of the categorical encoding of side to be gazed at or chosen, not of the offer value. (2) Fluctuations of activity during the reactivation at *delay 2* time are correlated with choice, indicating that there is a reevaluation stage.

We have added references for the suggested sources [8-9] and [f-g] in the Discussion.

[8] Manohar, S. G. & Husain, M. Attention as foraging for information and value. *Front. Hum. Neurosci.* 7, 711 (2013).

[9] Westbrook, A. et al. Dopamine promotes cognitive effort by biasing the benefits versus costs of cognitive work. *Science* 367, 1362–1366 (2020).

Added reference:

[f] Hare, T. A., Schultz, W., Camerer, C. F., O'Doherty, J. P., & Rangel, A. (2011). Transformation of stimulus value signals into motor commands during simple choice. *Proceedings of the National Academy of Sciences*, 108(44), 18120-18125.

[g] Milstein, D. M., & Dorris, M. C. (2007). The influence of expected value on saccadic preparation. *Journal of Neuroscience*, 27(18), 4810–4818. <https://doi.org/10.1523/JNEUROSCI.0577-07.2007>

13) The idea that the OFC “only encodes the value of fixated offers” is not universally supported. The discussion on lines 453-466 overlooks several studies showing that OFC encodes both the currently viewed and, to some extent, previously viewed offers simultaneously in tasks with serial designs and/or free gaze. See studies from Hunt[4], Ballesta[10], Yoo[11], and McGinty[6].

We thank the reviewer for this observation. We had included references to [10] and [11].

In the third paragraph of Discussion, we mention that our findings are not aimed to align with task paradigms that foresee gaze fixation as in [10]. While in [11], the authors consider asynchronous encoding, and do not enter in the detail of gaze alignment.

We are now mentioning that for asynchronous encoding we find overlap in the identity of neural encoding cells (of approximately 41% of cells encoding either offer value at its presentation time), while for simultaneous encoding we do find much less overlap (approximately 1/3 the cells encoding the two offer values asynchronously). We have detailed these results in the second paragraph of Results R3.

We have further mentioned these aspects and cited the suggested paper [6] in the second paragraph of Discussion, and [4] in the third paragraph of Discussion.

In case of further interest in aspects related with the intersection of significantly encoding populations, we have now included supplementary information below in Comment Tables CT3-CT6.

Supplementary Information about the overlap of encoding neural populations

SV_L cells sig. (fractions%)	AND delay1	AND offer2	AND delay2	AND ch-hold	NOT offer1	NOT delay1	NOT offer2	NOT delay2	NOT ch-hold
offer1 43 ^[18] (17.3%)	16 ^[2] (6.5%)	5 ^[2] (2.0%)	10 ^[2] (4.0%)	10 ^[2,5] (4.0%)		27 ^[17.5] (10.9%)	38 ^[17.5] (15.3%)	33 ^[18] (12.3%)	33 ^[17] (13.3%)
delay1 34 ^[17] (13.7%)		6 ^[2] (2.4%)	13 ^[2] (5.2%)	16 ^[2] (6.5%)	18 ^[17] (7.3%)		28 ^[16.5] (11.3%)	21 ^[16.5] (8.5%)	18 ^[17] (7.3%)
offer2 32 ^[19] (12.9%)			9 ^[2] (3.7%)	9 ^[2] (3.7%)	27 ^[18] (10.9%)	26 ^[18] (10.5%)		23 ^[17] (9.3%)	23 ^[18] (9.3%)
delay2 39 ^[18] (15.7%)				15 ^[2] (2.4%)	29 ^[18] (11.7%)	26 ^[17] (10.5%)	30 ^[16.5] (12.1%)		24 ^[17] (9.7%)
ch-hold 35 ^[16] (14.1%)					25 ^[16] (10.1%)	19 ^[16] (7.7%)	26 ^[16] (10.5%)	20 ^[16] (8.1%)	

Comment Table CT3. Number of cells ^[95th percentile from shuffled data] (fractions of the total number of cells n=248) showing significant encoding for SV_L in each epoch following offer1 presentation (first column), showing significant encoding of SV_L in pairs of epochs (columns 2-5), and showing significant encoding of SV_L exclusively in either epoch (columns 6-10). As shown, all numbers of cells are above 95th percentiles.

SV_R cells sig. (fractions%)	AND delay1	AND offer2	AND delay2	AND ch-hold	NOT offer1	NOT delay1	NOT offer2	NOT delay2	NOT ch-hold
offer1 6 ^[19] (2.4%)	1 ^[2] (0.4%)	0 ^[2] (0%)	2 ^[2] (0.8%)	0 ^[2] (0%)		5 ^[18] (2.4%)	6 ^[18.5] (2.0%)	4 ^[18.5] (1.6%)	6 ^[18] (2.4%)
delay1 8 ^[18] (3.2%)		1 ^[2] (0.4%)	0 ^[2] (0%)	0 ^[2] (0%)	7 ^[16.5] (2.8%)		7 ^[17] (2.8%)	8 ^[17.5] (3.2%)	8 ^[17] (3.2%)
offer2 32 ^[18] (12.9%)			4 ^[3] (1.6%)	5 ^[2] (2.0%)	32 ^[17] (12.9%)	31 ^[17] (12.5%)		28 ^[17] (11.3%)	27 ^[17] (10.9%)
delay2 44 ^[18] (17.7%)				10 ^[2] (4.0%)	42 ^[17] (16.9%)	44 ^[18] (17.7%)	40 ^[18] (16.1%)		34 ^[17.5] (13.7%)
ch-hold 27 ^[18] (10.9%)					27 ^[17] (10.9%)	27 ^[17] (10.9%)	22 ^[17.5] (8.9%)	17 ^[18] (6.8%)	

Comment Table CT4. Number of cells ^[95th percentile from shuffled data] (fractions of the total number of cells n=248) showing significant encoding for SV_R in each epoch following offer2 presentation (first column), showing significant encoding of SV_R in pairs of epochs (columns 2-5), and showing significant encoding of SV_R exclusively in either epoch (columns 6-10). Numbers of cells above 95th percentiles are in black. Numbers below percentiles, as expected before offer2 presentation, are in gray.

β'_1 corr. (signif.)	delay1	offer2	delay2	ch-hold	β'_2 corr. (signif.)	delay2	ch-hold
offer1	0.67 (***)	0.40 (***)	0.53 (***)	0.51 (***)	offer2	0.05 (-)	0.36 (***)
delay1		0.59 (***)	0.70 (***)	0.70 (***)	delay2		0.44 (***)
offer2			0.54 (***)	0.55 (***)			
delay2				0.58 (***)			

Comment Table CT5. Autocorrelation of SV_L weights (β_1 , columns 2-5) and SV_R weights (β_2 , columns 7-8) between epochs following respective offer presentation. P-values FDR corrected via Benjamini-Hochberg, 1995 method (- n.s., *** $p < 0.001$). Note that though correlations among weights may show significant, few cells may be significant in both epochs when taken in pairs (to be compared to Comment Table CT3, columns 2-5; Comment Table CT4, columns 2-5).

Cells signif. (%)	AND SV_R off2	AND SV_R del2	AND SV_R ch-hd	corr. (β'_1, β'_2) (signif.)	β'_2 off2	β'_2 del2	β'_2 ch-hd
SV_L off1	18 ^[2] (7.3%)	10 ^[2.5] (4.0%)	4 ^[2] (1.6%)	β'_1 Offer1	0.84 (***)	0.10 (-)	0.37 (***)
SV_L del1	10 ^[2] (4.0%)	12 ^[2] (4.8%)	7 ^[2] (2.8%)	β'_1 delay1	0.52 (***)	0.29 (***)	0.53 (***)
SV_L off2	6 ^[2] (2.4%)	6 ^[2] (2.4%)	3 ^[2] (1.2%)	β'_1 offer2	0.24 (***)	0.01 (-)	0.35 (***)
SV_L del2	7 ^[2] (2.8%)	13 ^[2] (5.2%)	5 ^[2] (2.0%)	β'_1 delay2	0.42 (***)	-0.01 (-)	0.34 (***)
SV_L ch-hd	6 ^[2] (2.4%)	11 ^[2] (4.4%)	8 ^[2] (3.2%)	β'_1 ch-hold	0.44 (***)	0.23 (***)	0.56 (***)

Comment Table CT6. Number of cells ^[95th percentile from shuffled data] (fractions of the total number of cells $n=248$) showing significant encoding of both offers SV s (columns 2-4) between epochs following respective offer presentation. Correlation between regression weights (β'_1, β'_2) in epoch pairs (columns 6-8). P-values FDR corrected via Benjamini-Hochberg, 1995 method (- n.s., *** $p < 0.001$). Note that though correlations among weights may show significant, few cells may be significant in both epochs when taken in pairs (at most 7.3%, to be compared to columns 2-4). As shown, all numbers of cells are above 95th percentiles.

14) Figure 4A says EV, but caption says SV.

We thank the reviewer for spotting this typo, fixed in updated version.

15) Supplementary Figure 2 caption refers to Figure 2E; but Figure 2 has only panels A-D.

We thank the reviewer for spotting this typo, intended to be Fig. 1E, fixed in updated version.

REFERENCES

1. Yasuda, M., Yamamoto, S. & Hikosaka, O. Robust Representation of Stable Object Values in the Oculomotor Basal Ganglia. *J. Neurosci.* 32, 16917–16932 (2012).
2. Cavanagh, S. E., Malalasekera, W. M. N., Miranda, B., Hunt, L. T. & Kennerley, S. W. Visual fixation patterns during economic choice reflect covert valuation processes that emerge with learning. *Proc. Natl. Acad. Sci.* 116, 22795–22801 (2019).
3. McGinty, V. B., Rangel, A. & Newsome, W. T. Orbitofrontal Cortex Value Signals Depend on Fixation Location during Free Viewing. *Neuron* 90, 1299–1311 (2016).
4. Hunt, L. T. et al. Triple dissociation of attention and decision computations across prefrontal cortex. *Nat. Neurosci.* 21, 1471–1481 (2018).
5. Conen, K. E. & Padoa-Schioppa, C. Neuronal variability in orbitofrontal cortex during economic decisions. *J. Neurophysiol.* 114, 1367–1381 (2015).
6. McGinty, V. B. & Lupkin, S. M. Value signals in orbitofrontal cortex predict economic decisions on a trial-to-trial basis. 2021.03.11.434452 Preprint at <https://doi.org/10.1101/2021.03.11.434452> (2021).
7. Yoo, S. B. M., Sleezer, B. J. & Hayden, B. Y. Robust Encoding of Spatial Information in Orbitofrontal Cortex and Striatum. *J. Cogn. Neurosci.* 30, 898–913 (2018).
8. Manohar, S. G. & Husain, M. Attention as foraging for information and value. *Front. Hum. Neurosci.* 7, 711 (2013).
9. Westbrook, A. et al. Dopamine promotes cognitive effort by biasing the benefits versus costs of cognitive work. *Science* 367, 1362–1366 (2020).
10. Ballesta, S. & Padoa-Schioppa, C. Economic Decisions through Circuit Inhibition. *Curr. Biol.* 29, 3814–3824.e5 (2019).
11. Yoo, S. B. M. & Hayden, B. Y. The Transition from Evaluation to Selection Involves Neural Subspace Reorganization in Core Reward Regions. *Neuron* 0, (2019).

Reviewer #2 (Remarks to the Author):

Summary

This study addresses a crucial and ongoing question in decision-making concerning the role of eye position, or gaze, in gating which visual offers (i.e., stimuli) are evaluated—and correspondingly which stimulus values are represented in the brain—thereby determining the ultimate decision. This work builds upon a 10- to 15-year body of research establishing the role of gaze in evaluating simple binary choices and showing, in monkeys, that gaze modulates the putative value representations in orbitofrontal cortex (OFC). The current study extends this work by testing this same role of gaze on choices and OFC activity, but now asking if/how eye position modulates the decision process after the stimuli have been removed from view, but when subjects nonetheless are presumably still deliberating. The authors integrate past work on the interaction between gaze and visuospatial memory to posit that eye position plays a similar role after stimuli have been removed as when the stimuli are present: gaze directed at the former location of a stimulus both indicates that the subject is evaluating that stimulus (from memory) and facilitates that reevaluation process insofar as gating which stimulus value OFC is presently representing, and thereby influencing which stimulus some downstream circuit will ultimately choose.

The authors have designed and implemented a study that is well suited to test this hypothesis, and offer evidence that is suggestive of their core idea. However, the most novel aspect of the hypothesis—that gaze facilitates retrieval of a value memory and gates OFC encoding (“reactivation”) of that memory—is not sufficiently supported by the manuscript in its current form. I believe that the findings in general, and this key finding in particular, would likely become both more convincing and accessible in a significantly revised manuscript.

My key concerns involve the rigor/design of the analyses, the mismatch between the strength of interpretation and evidence to support the claims, and the density and length of the exposition, which would benefit from significant distillation and prioritization. Finally, the manuscript would benefit from greater theoretical integration regarding the import of their findings on past literature and for models of visually guided decision-making. I have offered several suggestions for all of these areas and would be glad to review a revised manuscript.

We would like to thank the reviewer for the so many insightful comments and proposals. We have addressed all of them in detail below, thanks to which we think that we have strongly improved the quality of the manuscript.

Major Points

Analysis

Separate regressions

Throughout, separate regressions are used for each variable of interest instead of a single model including all variables.

This practice does not allow different regressors (e.g., gaze sidedness, epoch, offer values) to

compete for variance, and thus cannot make claims about independence that, I believe, the authors wish to make, such as, “gaze in epoch A explains choice above and beyond that in epoch B”.

We thank the reviewer for this question. In Fig. 2C-D we use logistic regression with multiple regressors, just as the reviewer proposes, thus following a standard approach at the behavioral level (note that in the new analysis shown in the revised Fig. 2C-D, proposed below by the reviewer, we have also added a new regressor, the magnitude of reward for each offer). In our neural data analysis, indeed, in Fig. 3 we use simple linear regression with just one regressor at a time. However, we do use multiple linear regression in Fig. 4 by combining linearly the subjective values (in turn including expected value, magnitude, variance) and choice. As can be appreciated by comparing Fig. 3 and Fig. 4, we do not find qualitative difference in any of the reported effects. Therefore, we consider our results are robust to the level of linear analysis, simple or multiple.

In addition, having so many separate models is confusing and risks overwhelming the reader. This practice of using separate regressions is used throughout the behavioral and neural analyses, and is too frequent to enumerate each case. In general, I would recommend using a single regression in which terms of interest compete for variance and interaction terms are used to test the dependency between 2 or more variables (e.g., hypothesis that slope of $t_R - t_L$ depends on dEV).

We appreciate this comment, related to the above one. Our strategy has been to initially dissect the effect of single variables in a model-independent manner, and only later include all of them in either multiple logistic regression for the behavior results or multiple linear regression for the neural results. Note that starting with a model-independent analysis is very typical in behavioral sciences (first start with psychometric curves, general trends, etc.). Also, in terms of presenting our findings, it was useful to first describe the basic effects of e.g., $EV_R - EV_L$ and $t_R - t_L$ in independent plots (Fig. 2A,B), so that we can immediately next use logistic regression analysis (Fig. 2C), including all relevant regressors, as the reviewer proposes. By doing things this way, we get the best of both worlds – clarity of exposition for a first-time reader, but a completeness that is not prone to analyses biases of the type in reviewer’s concerns.

Fig 2D is a welcome exception to the above in that includes multiple epochs in a single model. Can this type of approach be used throughout?

As noted above, in Fig. 4 we have also used multiple linear regression, and in Fig. 2C we have also used multiple logistic regression, so Fig. 2D is not the only case where we combine regressors.

Fraction of neurons analyses

The neural analyses are based on fraction of significant neurons. This approach has several limitations that limit interpretability of the results.

First, this approach does not allow us to track neurons across epochs in the trial. That is, we do not

gain insight into if or how quickly neurons are changing their encoding. For instance, in Fig 3 (though the concept applies to multiple figures), when the proportion EV_L goes down during offer2, is that because the former EV_L neurons are becoming silent and new neurons are coming on to encode EV_R, or are the EVL neurons switching to becoming EVR neurons?

This question is very relevant. In our data, we find that approximately 41% of the peak number of cells that show significant encoding of either offer SV also asynchronously participate in encoding both offer SV s at their respective presentation time ($n=18$ out of 43-44 cells, 7.3% of total cells). By computing correlation epoch-wise between β'_1 and β'_2 (from linear rate model $\eta = \beta'_0 + \beta'_1 SV_L + \beta'_2 SV_R$ as in Methods M4.6), we find that linear regression weights of cells encoding the two offers at their respective presentation times are significantly correlated ($\rho = 0.84, p < 0.001$). This result is in line with previous reports (Strait et al., Neuron 2014) [c].

When considering simultaneous encoding of both offers, we find that approximately 14% of cells encoding the first offer at first offer presentation time also encode the second offer soon after it is presented ($n=6$ out of 43-44 cells, 2.4% of total cells). Note that this number is 1/3 of the above number of cells showing asynchronous encoding ($n=6$ instead of $n=18$). The linear regression weights of the two SV s at offer 2 are again significantly correlated ($\rho = 0.24, p < 0.001$).

Taken together, these two results suggest that the two offers are in part encoded by same cells, but that only a fraction of them encodes both SV s simultaneously.

At first offer presentation time, 17.3% of total cells encode the first offer SV , then as soon as the second offer is presented, some of them (7.3% of total cells) also encode the second offer. Of this 7.3%, only 1/3 ($\approx 2.4\%$ of total) cells encoding first offer encodes second offer while still encoding first offer at offer 2 time. The remainder 2/3 ($\approx 4.9\%$ of total) cells that was encoding offer 1 and that shows asynchronous encoding of offer 2 switched to exclusively encoding second offer at offer 2 time.

In addition to all this, to discuss the continuity of the encoding between presentation epochs and delay 2 (reactivation time), we compute the correlation between regression weights of subjective values at different looking conditions in such epochs.

In the following analyses, we always fixed offer 2 at LookR, hence whenever we report delay 2 LookL, results imply gaze-shifts; in contrast, delay 2 LookR implies staying on most recently presented offer side. For ipsilateral configurations (encoding of SV_L and LookL; encoding of SV_R and LookR) we found significant, positive correlations ($\rho = 0.55$ for SV_L LookL in offer 1 and LookL in delay 2; $\rho = 0.22$ for SV_R LookR in offer 2 and LookR in delay 2; $p < 0.001$ in both cases).

For contralateral configurations we find negative correlations ($\rho = -0.21$ for SV_L LookL in offer 1 and LookR in delay 2, $p < 0.001$; $\rho = -0.12$ for SV_R LookR in offer 2 and LookL in delay 2, n.s.).

We find that 46 cells (18.5% of the total) encode SV_L at offer 1 LookL, 24 cells (9.7% of the total) encode SV_L at delay 2 LookL, and 5 cells (2.0% of the total) encode SV_L in both cases. This indicates that $\approx 1/5$ of same cells active during offer 1 LookL re-activate at delay 2 LookL, and that the remainder $\approx 4/5$ of cells involved in delay 2 LookL reactivation of SV_L are not the same as the ones involved in initial offer value encoding.

We checked that all results discussed in this response would qualitatively agree with equivalent results using EV s instead of SV s.

We have now included the most important results from the above in the 2nd paragraph of Results R4.

[c] Strait, C.E., Blanchard, T.C. and Hayden, B.Y., 2014. Reward value comparison via mutual inhibition in ventromedial prefrontal cortex. *Neuron*, 82(6), pp.1357-1366.

In case of further interest in the specific aspect of encoding significance / weights correlations across epochs, we provide more exhaustive analyses about simultaneous and asynchronous fractions of cells significant in task-relevant configurations, and correlations among weights in Comment Tables CT3-6.

(Copy of response to Reviewer1, Comment 13)

Supplementary Information about the overlap of encoding neural populations

SV_L cells sig. (fractions%)	AND delay1	AND offer2	AND delay2	AND ch-hold	NOT offer1	NOT delay1	NOT offer2	NOT delay2	NOT ch-hold
offer1 43 ^[18] (17.3%)	16 ^[2] (6.5%)	5 ^[2] (2.0%)	10 ^[2] (4.0%)	10 ^[2.5] (4.0%)		27 ^[17.5] (10.9%)	38 ^[17.5] (15.3%)	33 ^[18] (12.3%)	33 ^[17] (13.3%)
delay1 34 ^[17] (13.7%)		6 ^[2] (2.4%)	13 ^[2] (5.2%)	16 ^[2] (6.5%)	18 ^[17] (7.3%)		28 ^[16.5] (11.3%)	21 ^[16.5] (8.5%)	18 ^[17] (7.3%)
offer2 32 ^[19] (12.9%)			9 ^[2] (3.7%)	9 ^[2] (3.7%)	27 ^[18] (10.9%)	26 ^[18] (10.5%)		23 ^[17] (9.3%)	23 ^[18] (9.3%)
delay2 39 ^[18] (15.7%)				15 ^[2] (2.4%)	29 ^[18] (11.7%)	26 ^[17] (10.5%)	30 ^[16.5] (12.1%)		24 ^[17] (9.7%)
ch-hold 35 ^[16] (14.1%)					25 ^[16] (10.1%)	19 ^[16] (7.7%)	26 ^[16] (10.5%)	20 ^[16] (8.1%)	

Comment Table CT3. Number of cells ^[95th percentile from shuffled data] (fractions of the total number of cells n=248) showing significant encoding for SV_L in each epoch following offer1 presentation (first column), showing significant encoding of SV_L in pairs of epochs (columns 2-5), and showing significant encoding of SV_L exclusively in either epoch (columns 6-10). As shown, all numbers of cells are above 95th percentiles.

SV_R cells sig. (fractions%)	AND delay1	AND offer2	AND delay2	AND ch-hold	NOT offer1	NOT delay1	NOT offer2	NOT delay2	NOT ch-hold
offer1 6 ^[19] (2.4%)	1 ^[2] (0.4%)	0 ^[2] (0%)	2 ^[2] (0.8%)	0 ^[2] (0%)		5 ^[18] (2.4%)	6 ^[18.5] (2.0%)	4 ^[18.5] (1.6%)	6 ^[18] (2.4%)
delay1 8 ^[18] (3.2%)		1 ^[2] (0.4%)	0 ^[2] (0%)	0 ^[2] (0%)	7 ^[16.5] (2.8%)		7 ^[17] (2.8%)	8 ^[17.5] (3.2%)	8 ^[17] (3.2%)
offer2 32 ^[18] (12.9%)			4 ^[3] (1.6%)	5 ^[2] (2.0%)	32 ^[17] (12.9%)	31 ^[17] (12.5%)		28 ^[17] (11.3%)	27 ^[17] (10.9%)
delay2 44 ^[18] (17.7%)				10 ^[2] (4.0%)	42 ^[17] (16.9%)	44 ^[18] (17.7%)	40 ^[18] (16.1%)		34 ^[17.5] (13.7%)
ch-hold 27 ^[18] (10.9%)					27 ^[17] (10.9%)	27 ^[17] (10.9%)	22 ^[17.5] (8.9%)	17 ^[18] (6.8%)	

Comment Table CT4. Number of cells ^[95th percentile from shuffled data] (fractions of the total number of cells n=248) showing significant encoding for SV_R in each epoch following offer2 presentation (first column), showing significant encoding of SV_R in pairs of epochs (columns 2-5), and showing significant encoding of SV_R exclusively in either epoch (columns 6-10). Numbers of cells above 95th percentiles are in black. Numbers below percentiles, as expected before offer2 presentation, are in gray.

β'_1 corr. (signif.)	delay1	offer2	delay2	ch-hold	β'_2 corr. (signif.)	delay2	ch-hold
offer1	0.67 (***)	0.40 (***)	0.53 (***)	0.51 (***)	offer2	0.05 (-)	0.36 (***)
delay1		0.59 (***)	0.70 (***)	0.70 (***)	delay2		0.44 (***)
offer2			0.54 (***)	0.55 (***)			
delay2				0.58 (***)			

Comment Table CT5. Autocorrelation of SV_L weights (β'_1 , columns 2-5) and SV_R weights (β'_2 , columns 7-8) between epochs following respective offer presentation. P-values FDR corrected via Benjamini-Hochberg, 1995 method (- n.s., *** $p < 0.001$). Note that though correlations among weights may show significant, few cells may be significant in both epochs when taken in pairs (to be compared to Comment Table CT3, columns 2-5; Comment Table CT4, columns 2-5).

Cells signif. (%)	AND SV_R off2	AND SV_R del2	AND SV_R ch-hd	corr. (β'_1, β'_2) (signif.)	β'_2 off2	β'_2 del2	β'_2 ch-hd
SV_L off1	18 ^[2] (7.3%)	10 ^[2.5] (4.0%)	4 ^[2] (1.6%)	β'_1 Offer1	0.84 (***)	0.10 (-)	0.37 (***)
SV_L del1	10 ^[2] (4.0%)	12 ^[2] (4.8%)	7 ^[2] (2.8%)	β'_1 delay1	0.52 (***)	0.29 (***)	0.53 (***)
SV_L off2	6 ^[2] (2.4%)	6 ^[2] (2.4%)	3 ^[2] (1.2%)	β'_1 offer2	0.24 (***)	0.01 (-)	0.35 (***)
SV_L del2	7 ^[2] (2.8%)	13 ^[2] (5.2%)	5 ^[2] (2.0%)	β'_1 delay2	0.42 (***)	-0.01 (-)	0.34 (***)
SV_L ch-hd	6 ^[2] (2.4%)	11 ^[2] (4.4%)	8 ^[2] (3.2%)	β'_1 ch-hold	0.44 (***)	0.23 (***)	0.56 (***)

Comment Table CT6. Number of cells ^[95th percentile from shuffled data] (fractions of the total number of cells $n=248$) showing significant encoding of both offers SV 's (columns 2-4) between epochs following respective offer presentation. Correlation between regression weights (β'_1, β'_2) in epoch pairs (columns 6-8). P-values FDR corrected via Benjamini-Hochberg, 1995 method (- n.s., *** $p < 0.001$). Note that though correlations among weights may show significant, few cells may be significant in both epochs when taken in pairs (at most 7.3%, to be compared to columns 2-4). As shown, all numbers of cells are above 95th percentiles.

Moreover, it seems the authors (understandably) want to make observations about how a given (set of) neuron(s) change over the trial (e.g., line 376, "a significant fraction ... carries signals through ... and became prominent), but the present analysis really can't say anything about a given set of neurons over time.

We also are blind to how some present feature (e.g., gaze-dependent firing rate) depends on some past feature (e.g., recent gaze). This is critical for interpreting delay-period activity, such as whether the sustained gaze or OFC activity during *delay 2* and related to *offer 2* is greater than we'd expect from the temporal "inertia" (i.e., autocorrelation) of fixation or firing rate, respectively.

Notably, Fig 3F and 3G get at the spirit of dynamics by conditioning on the previous eye position.

As the reviewer already noted, Fig. 3F and 3G already addressed whether the encoding of EV_L and EV_R during *delay 2* is merely due to the inertia of fixation or the temporal autocorrelation of the firing rate, arguing it is neither of the two. Specifically, the sustained encoding of EV_L during *delay 2* in *LookL* occurs when subjects were previously looking at the right (during *offer 2*) and implies shift of gaze to the left. This clearly means that the OFC response that we see during *delay 2* cannot be due to a previous fixation to the L offer side, nor to the autocorrelation of firing rate, since this effect is not present when subjects *LookR* during *delay 2* and the encoding of EV_L is lower at *offer 2* time whenever subjects *LookR*.

We have now enriched the investigation of EV encoding at delay times by performing such analyses time-locked to midline-crossing gaze shifts (Supp. Fig. S14) where we also show the presence of the EV_L reactivation effect at *delay 2* time when subjects *LookL* despite they were looking to opposite screen side at pre-shift time (Supp. Fig. S14J).

We understand that this aspect might have been less clear before, and we have now included lines with our above comments at the end of last paragraph of Results R3.

Independent time-bin analyses

A second concern is that neurons are labeled "significant" based on independent analyses performed in finely spaced, overlapping time bins.

By computing "significance" independently in bins separated by 10ms, the authors are implying that what the neuron is encoding can change every 10ms. Is that the authors' belief? If so, it should be justified. My sense is that the authors want a finer temporal scale to correlate with eye position. However, eye position also has a longer time constant than 10ms, so again, the short interval requires justification, or consider the use of a longer interval.

We thank the reviewer for the comment. We have considered short intervals of 10 ms in Fig. 3B,D and in Fig. 4A,C but the settings used are not simply considering the 10 ms for both neural spiking rate and eye position. We have used 10 ms as we deemed this setting fine enough to capture rather fast changes in gaze position, but sampling every 10 ms is only used for eye data. For neural data, due to the sparseness of firing, 10 ms is used as offset across sliding boxcar windows covering 200 ms aligned to the start of the 10 ms offset. We provide checks that using a shorter or longer boxcar window would not affect our main results in Supp. Fig. S10A for 150 ms and in Supp. Fig. S10B for 250 ms. Further than this, we show average significance of the fractions of cells in larger time scales before significance assessment in Fig. 3C,E and Fig. 4B,D, hence already providing results in terms of "what is a neuron encoding" on longer time scales addressing the reviewer's concern. Anyways, we consider using bins of 10ms as totally valid, and standard in other papers, to study the temporal evolution of the fraction of encoding cells. We understand that the usage of windows of 190 ms longer than the 10

ms bins used introduce temporal correlations between consecutive fractions, but as we discuss in the next point, our statistical analysis considers this choice in the settings of the analyses

A more serious concern is statistical. The analysis does not appear to adequately address the temporal dependency between time bins, both because the bins are overlapping (and so clearly are not independent!) and due to the temporal smoothness/autocorrelation in the spike trains. This likely contributes to a biased estimation of significance. It is not clear whether their permutation testing addresses this, since it seems that the permutation testing preserves the overlap and temporal autocorrelations. If I am mistaken, and the analysis adequately addresses these statistical concerns, that should be made more explicit and addressed in the Results section.

We understand the reviewer's concern and we provide a detailed description of the methods used, with further adjustments to consider all aspects mentioned in the above comment.

The bin-wise shuffling procedure (Methods *M4.1*) destroys temporal correlations between consecutive bins because we apply independent shuffles of the trial order across time bins. The adoption of this method was made necessary by the interest in dynamically selecting LookL/LookR trial pools at each time bin. However, consecutive bins in the empirical *LookL/LookR* data may as well lack temporal correlations, as they in part come from different trials – although there is partial overlap of trials across contiguous bins (the gazed side could not drastically change from bin to bin).

While this method was deemed sufficient to assess the significance of the fractions of cells in each bin (defined by the 95th percentile of shuffled fractions of cells, shaded areas in Figs. 3B, D, F, G, 4A, C and in all related Supp. Figs. S5-S12, S14), we understand that this might not be accurate enough when performing run length analyses (shown by solid lines at the bottom of Figs. 3B, D, F, G, 4A, C, and all related Supp. Figs. S5-S12, S14), i.e., to assess the significance of consecutive runs of significant bins.

Since shuffling in our methods consists in picking EVs from randomized trials and apply linear fit to empirical spike data (for EV analyses, or a random SV and random choice for SV and choice analyses), we observe that applying a fixed trial-order shuffle in all time bins would not prevent from selecting LookL/LookR spike data, but would not allow to dynamically pick random Evs from pools where subjects LookL/LookR as trials in the LookL/LookR pools vary in time.

To fix for the lack of temporal autocorrelations in our previous run length analyses (Methods *M4.1*, solid lines at the bottom of Figs. 3B, D, F, G, 4A, C, Supp. Figs. S5-S12, S14) we have now performed shuffling by randomly selecting EVs across trials, using the same randomization for all the bins in each trial. In cases where we already use 'all trials' (Fig. 3B, Fig. 4A, and related Supp. Figs. S6, S7, S9-S12) and do not consider *LookL/LookR*, this was a straightforward fix since we were not interested in selecting EVs based on the looking side.

By applying these fixes, we did not find major differences in our results, but we found that the 95th percentile of run lengths for significance assessment went from 2 bins (20 ms, for trial order shuffles independent across bins) to 5 bins (50 ms, for trial order shuffles fixed across bins).

Re-running all analyses led to results almost identical to previous results, but for the run length assessment where we simply do not report significant runs shorter than 5 bins anymore.

We have now updated main Figs. 3B, D, F, G, 4A, C, Supp. Figs. S5-S12, S14, 4th and 6th paragraph of Methods *M4.1*.

In recent years, several approaches for describing the above population-level dynamics (aka, “stability”)—and statistical methods for testing those dynamics—have been proposed and should be considered for the present paper. To name a few:

- Murray, J. D. et al. Stable population coding for working memory coexists with heterogeneous neural dynamics in prefrontal cortex. *Proc. Natl Acad. Sci. USA* 114, 394–399 (2017).
- Kimmel, D. L., Elsayed, G. F., Cunningham, J. P., & Newsome, W. T. (2020). Value and choice as separable and stable representations in orbitofrontal cortex. *Nat. Comm.*, 11(1), 1–19.
- Najafi, F., Elsayed, G. F., Cao, R., Pnevmatikakis, E., Latham, P. E., Cunningham, J. P., & Churchland, A. K. (2020). Excitatory and Inhibitory Subnetworks Are Equally Selective during Decision-Making and Emerge Simultaneously during Learning. *Neuron*, 105(1), 165-179.e8. This paper concerns longer timescales between trials. The other papers concern within-trial dynamics.
- Morcos, A. S. & Harvey, C. D. History-dependent variability in population dynamics during evidence accumulation in cortex. *Nat. Neurosci.* 19, 1672–1681 (2016).

We thank the reviewer for providing relevant references. However, the size of our simultaneously recorded neural populations is relatively small and heterogeneous across sessions ($n=248$ units, 31 ± 5.85 mean \pm sem per session, Supp. Table ST1), which limits the intention to perform a proper and interesting population analysis that would address, e.g., encoding of variables at the population level (Moreno-Bote et al., 2014) ^[i]. This is why we have focused on single neuron analysis in this paper. We are trying to get additional data with large populations to move in that interesting direction.

Fortunately, we believe that the data we have are sufficient to test our hypotheses, and that our hypotheses are both novel and important.

[i] Moreno-Bote, R., Beck, J., Kanitscheider, I., Pitkow, X., Latham, P., & Pouget, A. (2014). Information-limiting correlations. *Nature neuroscience*, 17(10), 1410-1417.

Coarse analysis of eye position

The analysis of eye position is surprisingly coarse given the authors’ stated hypotheses and interpretations.

A central theory of the manuscript is that gaze of a given stimulus gates cognitive deliberation about—and OFC encoding of—that stimulus, ultimately driving choices. However, analyses of eye position primarily concern whether subject is looking to the Left vs Right side of screen. (Gazing 51/49% is treated the same as 99/1% !). Why can’t the analyses depend on whether subject is fixating the stimulus? Or if gaze position is highly variable WRT the stimulus, then use some continuous predictor, like distance between gaze and stimulus.

We agree with the reviewer that our analysis uses coarse gaze separation. We have made three new analyses to provide further elaboration about our settings selection.

First, we have repeated the analysis in Fig. 3C, E by removing data points at times when the horizontal coordinate of gaze position was closer than 4 cm or 9.5 cm from screen midline. This allowed a better separation between left and right screen looking definitions, and considers data points when gaze was more finely directed towards the stimuli sites. Our results are robust to this removal, despite reducing trials availability. We have included this important control analysis in Supp. Fig. S13 and highlighted the results in the last paragraph of Results *R3* and Methods *M4.2*.

Second, to address the concern that in our analysis in Fig. 2A is based on $f_R < 0.5$ and $f_R > 0.5$ (and that doing so trials with 51/49% are treated the same as trials with 99/1%), we have now repeated the analysis using the condition $f_R < 0.25$ and $f_R > 0.75$ instead of $f_R < 0.5$ and $f_R > 0.5$, to increase the discriminability about where subjects are looking, and generate an updated Fig. 2A. To be extra clear about such methods, we include below Comment Figure 2, showing how our investigations were always guided by the observation that f_R (in turn based on *LookL* if eye position < 0 ; *LookR* if eye position > 0) showed quite noticeable bimodality in all task epochs.

Comment Figure 2 distribution of f_R in all task epochs. Data pooled for both subjects. Vertical dotted lines show median values.

Finally, although the timing of gazes is extremely variable and the gazing patterns are highly heterogeneous, we have been able to perform a gaze-dependent analysis by aligning the EV encoding analysis to midline-crossing gaze shifts Supp. Fig. S14. We have estimated (last paragraph in Methods *M2.2*) the timing of gaze shifts from one side of the screen to the other (which implies a midline crossing), and computed encoding of EV depending on the gaze shift side (from L to R, or vice versa) in both delay 1 and delay 2. Besides supporting our claims and reproducing our previous delay 2 re-activation of EV_L by *LookL* results, we further show that:

(Copy of response to Reviewer 1, Comment 3)

- (1) In *delay 1*, before the gaze shift from L to R (Supp. Fig. 14D left, light orange line), there is encoding of EV_L during brief periods of time, presumably indicating that subjects previously visually sampled / encoded the L offer, but soon after the shift to the R side, encoding decays to non-significant range;
- (2) In *delay 1*, there is no encoding of EV_L if subjects shifted the gaze from R to L (Supp. Fig. 14D left, dark orange line), presumably because subjects did not previously visually sample / encode the L offer;
- (3) In *delay 2*, there is no encoding of EV_L before the gaze is shifted from R to L, because the subjects are eventually not yet retrieving the L offer EV (Supp. Fig. S14E left, light orange line);
- (4) In *delay 2*, as expected, after the gaze is shifted from R to L there is encoding (reactivation) of EV_L (Supp. Fig. S14E left, dark orange line).
- (5) In *delay 2*, before the gaze shift from R to L (Supp. Fig. S14E right, dark purple line), there is encoding of EV_R , which then decays as soon as subjects gaze away from most recent offer presentation site.

We have described these and related aspects in the last paragraph of Results *R3* and provide the new Supp. Figure S13, S14.

Moreover, where exactly were subjects looking relative to the offers? Was there anything systematic? Did choice and/or OFC activity depend on precise eye position? The authors state that “subjects preferentially look at the physical rectangular shapes of the offers” (line 130). But where on the (relatively large) shape? For instance, how does eye position vary with the height of the gray portion of the bar? What about during the delay period viewing? For instance, during the delay, did proximity to the former position of the bar predict greater choice accuracy (i.e., sensitivity to offer value)?

These are interesting questions, but due to high variability of eye data across trials, we did not search for visual behavioral patterns by precisely targeting stimuli locations. We anyway illustrate all systematic behavioral patterns that we found in the eye data. Regarding whether subjects were directly looking at the stimulus and at the size of the bars, we show in below Comment Figure 3 in offer 1 and up to delay 2, and in choice-hold time, subjects tend to be more precise along horizontal axis than along the vertical one. However, during the delay times, when subjects shift their gaze to screen side opposite to most recent offer side, they most often reached the lower portion of the previously shown, but currently empty rectangular bar shapes. It is in this latter scenario that gaze seems to have lowest precision, both vertically and horizontally. We have no direct explanation about why this is the case, even considering that if the previous offer was most valuable, its stimulus would most likely have a higher vertical bar. It is possible that there is some strategic element in trying to reach the diametrically opposite location, though we did not investigate this specific aspect. Interestingly, the subjects most often looked at upper portions of the vertical stimuli bars at choice-hold time, in line with the observation that chosen offers were most valuable, hence had higher vertical bars.

As a further clarification, we want to specify that there is never a gray ‘portion’ of the bar, whenever offers are gray (safe) the bar is fully gray. When this happens, the offer has unitary reward probability, and given that the height of the bar matches unitary probability, we have full gray bar.

In relation to the probability and bar height, in below Comment Figure 4 we correlate vertical eye position at different task epochs with offer reward probability. We find that during respective offer presentation (i.e. when the stimuli are in vision), during delay and at choice time the vertical coordinate of subject’s eye position has significant correlation with the height of the colored portion of vertical stimuli (cueing at their probability). We confirmed this result both for left (Comment Figure 4, top) and right offer (Comment Figure 4, bottom). This would lead to the conclusion that despite subjects look at lower stimuli height locations during delay times, their vertical gaze coordinate is still significantly correlated to the height of the bar, and that this allows correct task performance.

Comment Figure 3. Same as Fig. 1D but showing offer presentation locations.

Comment Figure 4. Vertical eye position (normalized) vs probability of the offers (p_L , p_R probability matches height of the bars). The convention used is to always align the first offer on the left screen side (data in trials where first offer was on right screen side were horizontally mirrored). The top panels show probability of left (first) offer vs vertical eye position, while bottom panels show probability of right (second) offer vs vertical eye position, in different task epochs. Pairwise correlations (ρ) between probability and vertical eye position are shown in blue when significant ($*p < 0.05$, $***p < 0.001$, data pooled for all sessions in both subjects, $n = 5971$ trials).

The authors state “subjects gaze did lack precision in reaching exact stimuli locations” (line 679). So how did this imprecision (variance) in eye position relate to choice and OFC activity. (As an aside, the imprecision was surprising given the large size of the stimuli.)

We thank the reviewer for the comment, and we are sorry for the eventual confusion. What we meant is that as shown in the previous comment figure, gaze is not always precisely matching the rectangular shape of the stimuli during the offer periods (see above Comment Figure 3), and less so during the delay periods (though it could be expected). We have detailed the sentence in the second paragraph of Methods *M4.1* to clarify this.

This imprecision can appear surprising, yet it does less so when considering that we do not instruct the subjects to directly look at the offer stimuli during offer presentation, nor we did this (they were free to direct gaze at their will) during empty screen, delay times.

Another metric is based on the time fixating offer 1 vs 2, which at least has the advantage of being a continuous measure. But it is revealing a slightly different aspect of gaze—time vs. precision. How should we understand these 2 metrics? That is, what are they each telling us? And therefore, in the neural analyses, when should we use one over the other?

Note that both $t_R - t_L$ and f_R are continuous quantities, and they are related, as $f_R = t_R / (t_R + t_L)$. They are both valid over epochs of fixed duration, and only have different ranges of variability.

We have updated Fig. 2B and Methods M2.3 so that now we use $t_R - t_L$ normalized by the full epoch duration $t_R + t_L$ instead of the maximum over trials, making the two metrics more easily related.

We remind that to classify trials as *LookL/R* in neural data analyses we use the average gaze position, hence space-based metric instead of time-based metric. We argue that when using a fine resolution (e.g., 10 ms bins), considering average eye position >0 as *LookR*, and respectively <0 as *LookL* is a better choice than using f_R or any other time-based metric. Considering fractions of time or time differences is a better selection when interested in epoch-wise temporal effects at the expense of a coarser spatial resolution (as we do not consider gaze coordinate amplitude but its sign only). In contrast, the use of average gaze position over full epoch times may provide blurred discriminants, making us blind to relevant time-locked effects.

These questions about eye position are not incidental. Rather, the authors explicitly invoke precise eye position in making their central claim: “moving the eyes to another location, even if empty, washes out the neural memory of the previously seen offer in OFC: only the value of the offer that falls, or immediately fell, in the fovea is strongly encoded.” (Line 449-51). This statement emphasizes a strong hypothesis that precise eye position—indeed foveation—modulates OFC’s encoding of offer value from memory. And yet the analyses of eye position focus on the coarsest of metrics—like left vs. right side of the screen—which seem totally at odd with the hypothesis. That is, if the difference in gaze location can result in the difference between “washing out” vs. “reactivation”, then presumably OFC is exquisitely sensitive to gaze location; the analyses should be similarly precise.

The criticism and recommendation are very well taken. Thus, we have rewritten all those sentences (fourth paragraph of Discussion) to weaken any implication of gaze precision in our results.

We have updated (third paragraph, Discussion):

“moving the eyes to another location” to “shifting gaze from one side to the other side of the screen”.
“washes out the neural memory” to “coincides with a decrease in the neural encoding strength”;

Deleted: “value of the offer that falls in the fovea”.

Interpretation

The authors take a strong position that looking time during the delay period represents a “reevaluation” process that modulates OFC activity, which in turn reflects “reactivation” of memory for an offer, and ultimately has some (causal) impact on choice. While the authors are mindful of the limitations of a correlational study—and do not outright claim causal evidence—in my view they a) present this interpretation too strongly and without qualification, and b) do not adequately acknowledge or address a series of alternative interpretations of the results. Here are some of those alternatives that I believe should be ruled-out, discussed, and/or acknowledged.

Looking time and reevaluation

Is looking time reflecting a reevaluation process (that impacts choice) or it is merely predicting

choice independently (i.e., revealing the outcome of a decision process that does/did not depend on looking time)? Making a case for this distinction requires decoupling the linear dependency between looking time and choice. For instance, earlier work used a model (DDM) non-linearly linking viewing time and stimulus value to choice. Another option—though likely beyond the scope of this study—would be to systematically manipulate the stimulus during (or before) the viewing time.

If we understand well the reviewer's question, we are asked whether activity during *delay 2* reflects re-evaluation of the offer or it merely represent the outcome of the choice to be indicated later in the task. Note that we observe that there is encoding of EV_L during delay 2 when the animal looks at the left: this encoding would not be present if the neural activity were a mere representation of the to-be-made choice.

§ However, to further address this important question, we have performed two additional analyses (i, ii), which will be denoted by the (§) symbol in the reminder of the response to reviewer.

(copy from Reviewer 1, Comment 4)

i) If reactivation implies reevaluation in *LookL* trials, then when the animal chooses L, we should see more fluctuation in activity of neurons with positive tuning to SV_L , and less activity in neurons with negative tuning. In other words, a positive fluctuation of activity of a neuron should increase the probability of the L choice if the tuning of the neuron is such that an increase of in SV_L leads to an increase of rate (positive modulation). We define fluctuations are the residuals $z_i = \eta_i - (\beta'_0 + \beta'_1 SV_{L,i} + \beta'_2 SV_{R,i})$ at the i^{th} trial, and study the relationship between β'_1 and choice. Note that SV s are now defined as $SV = \beta_1 EV + \beta_2 m + \beta_3 \sigma^2$; following suggestion by Reviewer 2, we have now included the magnitude among relevant regressors, corresponding to the best behavioral model to predict choice (Supp. Fig. S4).

Our prediction is confirmed by the results (Fig. 4F). We predict that neurons with positive tuning and positive activity fluctuations better predict L choices, and that neurons with negative tuning better predict opposite, R choices, when there are positive activity fluctuations. To simplify the visualization, we combined data based on the choice side (L, R) and the sign of β'_1 by defining 'preferred choice' (positive β'_1 and choice=L or negative β'_1 and choice=R) and 'non-preferred choice' units (positive β'_1 and choice=R or negative β'_1 and choice=L), analogous to what is done in the choice-probability literature. By comparing residual magnitudes in the two data pools, we find significantly larger residuals in preferred choice pools, confirming that the sign of fluctuations of residual activity during reactivation correlate with choice, and in the predicted direction. Since for every neuron we have both preferred and non-preferred choices, we used paired Wilcoxon tests, yielding significant difference ($p=0.00021$) for *LookL*.

In Fig. 4F (right) we show as a control, that repeating the same analysis in *LookR* trials leads to a non-significant relationship between noise fluctuations when aligning SV_L tuning with choice, in line with the lack of reactivation during *LookR*.

ii) Following the same idea as before, we can ask the same question but correlating the regressors of the linear model $\eta = \beta_0 + \beta_1 SV_L + \beta_2 SV_R + \beta_3 ch$ for each unit, where choice is encoded as $ch = +1$ for R, and -1 for L. Then, following the same reasoning as above, we expect that in *LookL* trials in delay 2, a neuron with positive β_1 would be associated with a negative β_3 (that is, more positive firing

modulation to SV_L correlates to more negative modulation with choice). To avoid spurious correlations between regressors trained on the same dataset (Arandia et al., Neuron 2016)^[b], we estimated $\beta_1^{(1)}, \beta_3^{(1)}$ by using the first half of the trials, and $\beta_1^{(2)}, \beta_3^{(2)}$, by using the second half of the data, and then computed the correlation between $[\beta_1^{(1)} \beta_1^{(2)}]$ and $[\beta_3^{(2)} \beta_3^{(1)}]$ (combining β_1, β_3 in disjoint sets). The results are shown in Fig. 4E, confirming the prediction of a significant negative correlation in LookL trials, which is absent in LookR trials, as expected.

We have described these new important results in Fig. 4E-F, in the last paragraph of Results R4 and described the Methods in M4.5, M4.6.

We have added suggested references:

- [5] Conen, K. E., & Padoa-Schioppa, C. (2015). Neuronal variability in orbitofrontal cortex during economic decisions. *Journal of neurophysiology*, 114(3), 1367-1381.
- [6] McGinty, V. B., & Lupkin, S. M. (2021). Behavioral read out of population value signals in primate orbitofrontal cortex. *BioRxiv*, 2021-03.

In the last paragraph of Results R4.

And added references:

- [a] Britten et al., 1996 Britten, K. H., Newsome, W. T., Shadlen, M. N., Celebrini, S., & Movshon, J. A. (1996). A relationship between behavioral choice and the visual responses of neurons in macaque MT. *Visual Neuroscience*, 13, 87–100.
- [b] Arandia-Romero, I., Tanabe, S., Drugowitsch, J., Kohn, A., & Moreno-Bote, R. (2016). Multiplicative and additive modulation of neuronal tuning with population activity affects encoded information. *Neuron*, 89(6), 1305-1316.

In the last paragraph of Results R4 and in Methods M4.6.

In addition, it would be useful to assess the extent to which looking time during the delay is a function of various factors other than choice, including looking time during the offers, last fixation, absolute offer values, offer magnitude and/or probability (which impact visual appearance of offer), etc.

We thank the reviewer for this question. Note that our strategy has been to use linear model to predict choice in terms of looking time, value of the offers, etc., not the other way around. This simply states that our goal has always been to predict choice vs other regressors (most importantly EVs and looking times), and we have not attempted to predict looking time from choices and all other possible combinations of regressors.

OFC activity as memory reactivation

The fact that the offer stimuli and choice targets share the same positions creates a potential confound: pre-choice gaze and OFC firing may reflect a plan/anticipation of the upcoming choice, rather than some deliberative process about the stimulus. This would be a minor point—indeed most previous studies of OFC have this same feature—except that the authors make repeated and strong claims that the OFC activity represents a “reactivation” of a memory for the stimulus. (In

fact, this is a major thrust of the paper.) By interpreting the activity as “memory”, the authors are committing to the retrospective, deliberative process over the prospective, anticipatory one. In my view, either the interpretation should be made significantly more circumspect, or the authors should offer additional evidence and/or logical argument to support this strong interpretation.

We thank the reviewer again for this important comment. We agree that orthogonalized choice targets and stimulus locations would have eventually provided a better experimental design. However, we would like to restate that the activity observed during delay 2 when animals look at the L better fits the interpretation of memory reactivation of the L offer. This is because a purely intentional signal would only predict the stimulus L itself, without any reference to its value. However, what we find is that there is reactivation of the EV_L (or SV_L) encoding. Clearly, this signal is not purely a motor signal of categorical response. We have discussed this in the second paragraph of the Discussion.

Also, please check above the response (denoted with symbol ξ) about the correlation between choice and fluctuations of activity, which speaks again in favor of a deliberation process involving memory during delay 2 – this of course does not exclude the possibility that the signal is also reflecting motor preparation, which is now also discussed in the second paragraph of the Discussion.

As an aside, in future experiments, one might consider orthogonalizing stimulus and choice targets. For instance, the choice targets could appear above and below the fixation point, and/or the mapping choice target to spatial location could be randomized, such that if offer1 is presented on, say, the left, the choice target for offer1 would be randomized to either the left or right of (or above or below) the fixation point. This way the animal could not prepare an eye movement, or anticipate the value of a spatial location, prior to the choice phase. An additional step would be to use a button/joystick to effect the choice rather than an eye movement and spatial location.

This is indeed a very wise suggestion, which we will follow in next designs.

OFC activity as reevaluation and input to the decision process

How do we know reactivation reflects reevaluation? The authors suggest as evidence the correlation between spiking activity and choice. This particular form of correlation has been studied widely (under the banner of “choice predictivity”) and hotly debated as to the extent to which it reflects a causal role in the decision process vs. reflecting feedback from a decision process that occurs elsewhere and independently. (For instance, see Nienborg, H., Cohen, M. R., & Cumming, B. G. (2012). Decision-related activity in sensory neurons: correlations among neurons and with behavior. *Annual Review of Neuroscience*, 35, 463–483.) If the claim of (putative) causality is to be included, the ambiguity around this interpretation of choice predictive activity should also be featured prominently.

We thank the reviewer for the question. As said in the previous point and in choice correlation analysis (denoted with §), we now address the question of how choice correlates with fluctuations of activity while at the same time taking into account the tuning of the encoding of each neuron. We have reported considerations about these aspects in the second paragraph of Discussion, where we specify that we do not claim any causal interplay between of reactivation and choice.

Putting aside the interpretation of choice predictive correlation, I did not see what I expected to be the essential correlative analysis to support the hypothesis that gaze-dependent OFC activity represents deliberation/re-evaluation: controlling for the known, observable factors—task conditions (EVR/L, dEV, EVR+L), gaze (during offer1/2, epoch 1/2), pre-delay firing rate—is the remaining (i.e., unexplained) variance in choice explained by OFC activity during delay 2? And—to support the strongest form of the hypothesis—is this relationship conditional on concurrent gaze?

We have now analyzed these aspects in detail in above response (denoted with §).

Unfortunately, to my read, the data as presented contradict the strong form of the above hypothesis. Specifically, in Figure 4D, bottom row, delay2 epoch, the percentage of neurons significantly encoding choice is not statistically different for R vs. L choices. If choice predictive activity (CP) during epoch2 reflects gaze-dependent reevaluation, then we should see evidence of that here: CP for R should be greater for LookR than LookL, but it's not. Perhaps I am missing something?

We believe we have now addressed this point in above response (denoted with §). Note also that we do not need/expect to have any difference of encoding of choice during delay 2 when LookL vs LookR conditions. Instead, what we expect is that the sign of the fluctuations is aligned with the preferred choice of every neuron, and this is now shown in previous analyses (denoted with §).

Probably, part of the confusion was naming our choice variables as “chR” (we did not mean to use only trials with choice towards the right here), and hence we have changed it to “ch =choice”, to make clear that both R and L choice trials are included in the regression in each gazing condition (LookR and LookL).

The authors state “variability in the size of this [gaze-dependent] reactivation predicts choice” (Line 421), which contradicts my above interpretation that CP is NOT modulated by gaze. Clarifying the basis for this interpretation is crucial.

We have now included analyses about neural activity fluctuations and choice in above response (denoted with §).

The strong statements about reevaluation and reactivation occur throughout the manuscript and, in my view, should be qualified appropriately. A couple, but by no means all, examples include:

- Line 25: “This reactivation reflects a reevaluation process...”.
- Line 413: “gaze can reactivate neural responses that reflect the values of previously shown offers, presumably to facilitate their re-evaluation for the formation of a choice.”

We apologize again for this, as we did not intend to imply any causal effects. For how we wrote “reflect” was not intended to imply any causal link between activity and choice. We have rewritten “gaze correlates with a reactivation of neural responses ...” and checked all over the manuscript.

We have mentioned that we do not claim any causal interplay between gaze, spike rate and choice in the interpretation of results in the second paragraph of Discussion.

****Clarity****

To the authors credit, the writing was generally clear and (I believe) I have a clear sense of what was done. However, I felt that it too long to get to the key and most compelling results, which risk being buried and underappreciated by the reader.

To me, the critical analysis concerns the eye movements and OFC activity related to offer1 that occurs during delay2 (e.g., Fig 3D, top panel, “LookL”, but even more precisely shown in Fig 3G, and discussed lines 318-322—a full 2 text-only pages into the OFC results!). As the authors correctly point out, this period is special because it is not contiguous with the viewing of offer1, and therefore cannot be explained by a simple form of “inertia” (i.e., autocorrelation) in eye position or OFC activity, to which the other analysis periods (e.g., delay1 related to offer1, delay2 related to offer2) are subject.

Why not bring this point out front and center? At present, the reader may miss this point at best, and at worst may become skeptical of the authors’ overall interpretation since it is a much weaker argument when focusing on the other periods.

We thank the reviewer a lot for this valuable comment. However, we find difficult to explain the central results of Fig. 3 without going first to describe the simpler results. However, we have shortened the results sections by removing the least interesting results, and move them to Supplementary Information (e.g. previous Fig. 1E is now Supp. Fig. S2C), so that the central results appear more quickly.

Another example are the various regression models (often of limited interpretability—see comment re: use of separate models for each term) that gradually build until arriving at the key, appropriate model. For example, the behavioral analysis goes through many models before the full logistic model (lines 232-233). This is also true for the figures. Why not just show the critical and appropriate analysis in 2D, and omit the separate regressions in 2C?

We found necessary and clearer to describe the simpler behavioral effects based on traditional psychometric curves (Fig. 2A,B), as going directly to the multiple logistic regression would be too much of a jump. Further, we initially considered the results in Fig. 2D not much more important than the results in Fig. 2C, while we found the latter ones much simpler to introduce.

****Theory****

Putting the questions of analysis and interpretation aside, in the end, I would have appreciated greater assimilation with existing literature and greater discussion of a theory for gaze-gated memory retrieval and evaluation. Why is it important to sequentially evaluate offers, and why use gaze as the means to do so? What problems does it cause/solve? How does gaze-gated memory retrieval interact with other circuits related to memory, evaluation, choice, and eye movements, such as ACC, HPC/MTL, dIPFC/FEF?

All findings that we presented support the idea that offers are typically evaluated sequentially. Well-established, now standard methods in neuroeconomics make use of simultaneous offers presentation of offers, a practice inherited from behavioral economics. However, it does not necessarily follow that offers are evaluated simultaneously (Hayden, 2018; Hayden and Moreno Bote, 2018; Orquin Loose, 2013) ^[i-1].

At the most basic level, offers are best identified during foveation, implying that visually presented offers should be first identified in sequence. Beyond this, there is good reason to think that evaluation, which requires abstraction, is a demanding process and therefore is best performed within the spotlight of attention. That in turn would mean that it ought to be a sequential process, as performing two evaluations at once may be difficult or impossible. Indeed, evaluation requires integration of information from memory, and may even involve multiple different memories over time; these would also be attentionally demanding processes that would benefit from focal processing. Our data are not the first to demonstrate this idea; indeed, there is now a good deal of evidence that evaluation preferentially takes place in sequence (Krajbich et al., 2010; Rich and Wallis 2016; Xie Nie Yang, 2018) ^[m-o].

Why would sequential evaluation require fixation? It is not clear why, but there is plentiful evidence that gaze fixation facilitates memory retrieval. Wynn et al. (2019)^[p] show that reinstating gaze patterns during free viewing facilitates subsequent memory retrieval in cognitively demanding tasks. At the neural level, prefrontal cortex (Goldman-Rakic, 1995)^[q] has been associated to memory maintenance through sustained firing in post-presentation delay times, hippocampus has been shown to participate in memory-based decision formation (Bakkour et al., 2019; Biderman et al., 2020) ^[r-s]. Among most related, Bone et al. (2019)^[t] found that the encoding and retrieval of mental images had significantly similar gaze patterns and that, consistent with findings about 'looking at nothing', neural reactivation correlated with gaze reinstatement. In this work we propose that the use of gaze to enhance retrieval and therefore evaluation may reflect the benefits of reinstating the encoding state at the neural level, or may simply activate related memories, facilitating retrieval. Overall, determining why retrieval benefits from refixation remains an important and fascinating open question.

We have added text from the above response to the 4th paragraph of the Discussion.

Previous references cited:

- [j] Hayden, B. Y. (2018). Economic choice: the foraging perspective. *Current Opinion in Behavioral Sciences*, 24, 1-6.
- [k] Hayden, B. Y., & Moreno-Bote, R. (2018). A neuronal theory of sequential economic choice. *Brain and Neuroscience Advances*, 2, 2398212818766675.
- [l] Orquin, J. L., & Loose, S. M. (2013). Attention and choice: A review on eye movements in decision making. *Acta psychologica*, 144(1), 190-206.
- [m] Krajbich, I., Armel, C., & Rangel, A. (2010). Visual fixations and the computation and comparison of value in simple choice. *Nature neuroscience*, 13(10), 1292-1298.
- [n] Rich, E. L., & Wallis, J. D. (2016). Decoding subjective decisions from orbitofrontal cortex. *Nature neuroscience*, 19(7), 973-980.
- [o] Xie, Y., Nie, C., & Yang, T. (2018). Covert shift of attention modulates the value encoding in the orbitofrontal cortex. *Elife*, 7, e31507.

New references added:

- [p] Wynn, J. S., Shen, K., & Ryan, J. D. (2019). Eye movements actively reinstate spatiotemporal mnemonic content. *Vision*, 3(2), 21.
- [q] Goldman-Rakic, P. S. (1995). Cellular basis of working memory. *Neuron*, 14(3), 477-485.
- [r] Bakkour, A., Palombo, D. J., Zylberberg, A., Kang, Y. H., Reid, A., Verfaellie, M., ... & Shohamy, D. (2019). The hippocampus supports deliberation during value-based decisions. *elife*, 8, e46080.
- [s] Biderman, N., Bakkour, A., & Shohamy, D. (2020). What are memories for? The hippocampus bridges past experience with future decisions. *Trends in Cognitive Sciences*, 24(7), 542-556.
- [t] Bone, M. B., St-Laurent, M., Dang, C., McQuiggan, D. A., Ryan, J. D., & Buchsbaum, B. R. (2019). Eye movement reinstatement and neural reactivation during mental imagery. *Cerebral Cortex*, 29(3), 1075-1089.

Minor Points

- The authors discuss why they used sequential presentation of offers, but it was still unclear to me the advantage over simultaneous presentation. If the key theory is that gaze during delay period gates evaluation of a specific offer, why not present both simultaneously, release fixation, eventually remove the offers, and then analyze gaze and neural activity both during the offer and delay periods? To be clear, I'm not suggesting that the sequential presentation is problematic, but since the authors (rightly) highlight it as a distinguishing feature, the pros/cons of this feature should be clear, as well.

We thank the reviewer for the question. We believe that sequential presentation had important advantages in our analysis. First, the gaze patterns during simultaneous presentations would have been much more variable, and therefore the conditioning analysis of Fig. 3F would have been more difficult to perform, a priori. Second, the delay 2 period happens long time later after offer 1 is presented, and therefore reactivation of the offer 1 value (EV_L or SV_L) cannot be explained by stimulus encoding without using gaze-centered vision. The fact that offer 1 disappears requires that there is memory retrieval of it. We agree with the reviewer that probably similar analyses could have been made with the simultaneous presentation of offers, but with the two caveats that we have just described. We have improved the first paragraph of Methods *M1* based on this response and included more details about the sequential nature of offer evaluation in the 4th paragraph of the Discussion.

- The task involves risky offers with variable magnitude m and probability p . The analyses all assume that subjects are computing the value of the offers as $m \cdot p$, or EV , and basing their

decision on the difference in value, $dEV = EV_R - EV_L$. It would be useful to show that this is indeed the case.

- That is, is there an influence of m or p linearly independent of EV ? For instance, when m is high, do subjects favor the higher m offer, even when offer EV s are equivalent? For example, this kind of question could be addressed in Fig 1B by showing that dEV explains choices across a range of absolute m and p .
- Even if we assume offer value is linear with EV , another question is how the 2 offers are compared. The authors assume choices are based on dEV . It would be useful to rule-out some alternatives.
 - ♣ First, it has been shown widely that choices depend on the absolute value of the offers, in addition to the difference in values. Would be useful to show choices as function of dEV for a range of absolute EV s. Also, in the regression models, it would be useful to consider a model that includes terms for absolute EV (e.g., $\{EV_R, EV_L\}$ or EV_R+EV_L) in addition to dEV . Surprisingly, in Fig 2C and related model, absolute values for EV_R and EV_L are used, but dEV is not included. Why is this?

We thank the reviewer for this important question and suggestion. We have now applied a logistic regression model with regressor EV , m (magnitude), p (probability) and σ^2 (for R and L), and compared it with a model that only uses EV s. We find that the best model is the one that includes, EV , magnitude m , and variance σ^2 , while the one that we have used so far included only EV and σ^2 (the best model has a 5% choice prediction performance advantage; see Supp. Fig. S4). Therefore, we have repeated all the analysis in Fig. 2C-D, Fig. 4 and related Supplementary Figures using a new definition of subjective value based on the improved model (last paragraph of Methods 2.3). We find that all main results are still present after using the improved behavioral model. We have described the new model and a model comparison analysis in Methods and Supp. Fig. S4. Note that with the new definition of subjective value (now including magnitude of reward) we do see higher and stronger reactivation of SV_L in LookL at delay 2 (see updated Fig. 4).

Finally, we want to recall that $EVR+EVL$ is always included in the model, as we consider any arbitrary linear combination of EVR and EVL in our regressions.

- ♣ Second, given a fixed range of offers, there are certain offers that will very likely be higher/lower than the alternative, making consideration of the other offer unnecessary. And given a fixed range, these cases will correlate strongly with dEV , meaning that dEV may appear to be a significant factor, but it is merely correlated with single offer magnitude. (Note that this is not an either/or hypothesis. Subjects can consider both offers, but weight them disproportionately.) I am not implying this is the case, but it would be very useful to rule out this case. And not just for completeness. Indeed, the primary thesis is that subjects direct gaze at the 2 offers (or their former location) so as to consider both offers; therefore, it is crucial to show behaviorally the extent to which subjects are basing decisions based on each offer.

Note that our analysis does not require that the animals look at the 2 offers. This is because we still have a large fraction of trials where animals look at the two sides of the screen sequentially. In the new Supp. Figure S6 we show what happens in subject do not look at the first offer and we do not find reactivation, as expected.

- Despite using EV for most analyses, the analyses in Fig 4 introduce a new term, SV, that depends on EV and offer variance. It is unclear why the authors switch to using SV for this analysis.

As explained in Results R2, it is possible that risk seeking attitude affects the decision-making behavior of non-human primates, as it has been reported before (Heilbronner and Hayden, 2013) ^[i]. Therefore, we included the variance term into the definition of subjective value. By testing different alternative logistic models of choice, we found that a model with SV_L and SV_R regressors is better than a model with EV_L and EV_R only (Supp. Fig. S4). Now, after the reviewer's proposal, we have further added a new regressor (reward magnitude) to the previous logistic regression model (Supp. Table ST3), leading to a more accurate model. In the new Fig. 4 and related analyses we have used the new definition of SV that includes the reward magnitude (Supp. Table ST3).

[i] Heilbronner, S. R., & Hayden, B. Y. (2013). Contextual factors explain risk-seeking preferences in rhesus monkeys. *Frontiers in Neuroscience*, 7 FEB. <https://doi.org/10.3389/fnins.2013.00007>

- Also, the SV term depends on fitting 2 parameters, w_1 and w_2 , to the data, which introduces additional uncertainty. Moreover, because SV is a fit term, for unbiased results, cross validation should be used: w_1 and w_2 should be fit on one set of trials and the betas (B1-3) should be estimated on a separate set of trials. Again, it may be simpler to stick with EV (if indeed EV is justified; see above).

This is considered in the model comparison by using cross-validation (last paragraph of Methods M2.3), showing that the model with SVs is better at predicting the choice than the model with EVs only (Supp. Fig. S4). Also, note that the parameters w_1 and w_2 and now also w_3 (since we included a magnitude regressor) are obtained from behavioral data (not neural data), and later the SV computed by logistic model of the choice is used to predict neural firing rate (all values of w_1 , w_2 , w_3 are detailed in Supp. Table ST3).

- For all behavioral logistic choice curves (e.g., Fig 1B), it is unclear what the points represent.
 - Are these individual trials? If so, for which session and which monkey? And if so, why are there so few? And if so, they we need some summary across behavioral sessions, such as a histogram of logistic fit parameters.

- If these are summary data, then it would be useful to show (some) individual sessions and a histogram across sessions to get a sense of the between-session variance.

All logistic choice curves include all trials (n=5971) from all (n=8) sessions in (n=2) subjects. We tried to carefully specify this information at the end of the captions of both figures, hopefully this was clear.

About the dots in logistic choice curves such as Fig.1B report, they show average number of trials with choice=R at binned values of the x axis. Since Fig. 1B and 2A, 2B include all trials in all sessions, it would be harder to visualize meaningful information without binning. We have edited the caption or the marker label from “data” to “binned avg. data”, so it is clearer.

We provide more details about session-by-session variability in the following.

First, we want to remark that results for Fig. 1B do match results in Fig. 2A “all trials” (bright red). These results were already shown separately for the two subjects (n=4 sessions each) in Supp. Fig. S1A. Results for each session in Fig. 2A and Fig. 2B are reported below in Comment Figures 5-6 and 7-8, respectively. The main extra insight learnt from the below figures is that modulation of behavioral curves during trial execution is more pronounced in subject 1. Yet, as reported in text (on the right of each row) of Comment Figures 5, 7, the probability of choosing R whenever R is best is about the same (approx. 50%) in both subjects and across sessions.

Comment Figure 5. Same as Fig. 2A, showing single sessions for subject1 (A) and subject2 (B).

Comment Figure 6. β_0 (top), β_1 (bottom) mean \pm s.d. (n=4) sessions per subject, for regressions in Fig. 2A.

Comment Figure 7. Same as Fig. 2B, showing single sessions for subject1 (A) and subject2 (B).

Comment Figure 8. β_0 (top), β_1 (bottom) mean \pm s.d. (n=4) sessions per subject, for regressions in Fig. 2B.

• I was unclear what to learn from Fig 1D that wasn't shown in Fig 1E (which is the more useful of the 2). Moreover, I wonder if 1D could actually lead to confusion. That is, since all trials contribute, we don't know if what appears to be fixation of both offers is occurring within a single trial (as I think the authors would like to claim), or whether different trials (e.g., chose L vs chose R trials) have systematically different eye positions (e.g., fixation of only 1 of the 2 offers). This is where 1E, or an alternative (see below), is useful.

We thank the reviewer for the suggestion. We have clarified in the caption of Fig. 1 that the icons refer to the most likely patterns. Note that we never claim that the fixation to both offers e.g. during delay 2 happens within the same trial. Note also that we have shown gaze conditioned on choice in Supp. Fig. S2D.

- I was interested in a slight variation of Fig 1E where the eye position is split by trials in which the subject ultimately chose L vs chose R, and ideally matched for dEV (or matched based on $p(\text{Chose R}) \pm \text{some epsilon}$). This would tell us about how eye position relates to the ultimate choice (i.e., the internal process), rather than how it relates to dEV (i.e., the external parameters). I'm not sure if my suggestion would be in addition to or in place of 1E.

As mentioned above, the average heatmap of eye position split by alternative choices is shown in Supp. Fig. S2D. Note that we also include the difference between groups of trials in Supp. Fig. S2E and S2F.

- Some colors are difficult to distinguish, including
 - Fig 2B, lines and points
 - Fig 2C, sigma L and sigma R.

We thank the reviewer for the suggestion. We have changed tonality of suggested colors to improve distinguishability.

- The Discussion highlights 3 major findings re: OFC activity, but I wasn't sure which results supported the 2nd finding: "(2) gazing to an empty region where no stimulus has been presented washes out that value encoding" (line 419). (Same topic comes up again on Line 449—see Major Points above.) Apologies if I simply missed this. Or consider making the link more clear?

We hope that we have responded to this above, describing results from new gaze-shift aligned analyses (Supp. Fig. S14). We are happy to address and respond to further points about this.

Reviewer #3 (Remarks to the Author):

Ferro and colleagues

This is an interesting manuscript that tests a clearly-specified and striking hypothesis. Ferro and colleagues consider how choice values are compared during decision making. First, they identify neurons in orbitofrontal cortex (OFC) in which activity covaries with choice value. They show that these neurons encode the value of each of two potential choices when they are shown to macaques but that they do so while the macaque is looking at them. Intriguingly, they re-encode those specific choice values when the animals are returning their gaze to the location in which the choice offer had been presented even if it is now no longer visible. The authors argue that the animals are reactivating one choice representation at a time in a manner that is guided by attention where the focus of attention continues to be specific and relatively exclusive even when the stimulus is no longer visible.

The results and the data are analysed carefully. A number of additional alternative analyses are presented in the supplementary materials. In general, the analyses support the authors' arguments. I was, however, unsure about one critical piece of evidence (although I think that are ways in which this might be addressed)

We thank the reviewer a lot for the insightful comments and proposals.

Main points

1. Lines 370-383 and figure 4a and 4c. Ferro and colleagues use a clever test to assess whether neural activity that occurs during periods when animals are looking at an empty screen, where an offer had been presented but is no longer visible, is related to the choice that animals will make at the end of the session. This is an important test because it is asking whether the neural activity encodes the choice the animal will make over and above the value of the potential choice; if this is the case then the neural activity is likely to be contributing to the decision process. The authors can therefore use this approach to ask whether activity that occurs when animals look at one empty location or another is related to the choice the animals will later make.

The activity in panel 4a suggests that the OFC activity during "empty" intervals does predict the choice the animal will later make (above chance level of the black line labelled ChR in 4a in the delay interval). However, it is only in figure 4c's bottom panel that the activity is analysed as a function of where the animal is gazing. The grey line reflects encoding of whether a choice will be made to the right (ChR) when the animal is gazing to the right. However, this grey line seems no higher than the black line which corresponds to trials when the animals were looking in the opposite direction. If I understand correctly this is what a strong interpretation of the authors' hypothesis would predict.

We thank the reviewer for this important comment. The fact that there is encoding of the choice variable regardless of whether animals LookR or LookL indeed shows that fluctuations of activity predict choice – otherwise, fluctuations of activity during delay 2 beyond what can be predicted by SVs would have not have shown any correlation with choice. However, note that we do not expect in this analysis to see any difference in the fraction of encoding cell on choice, so this result does not contradict our interpretation. This is because one expects that fluctuation of activity during delay 2 correlate with choice, regardless of gaze location.

Nevertheless, a stronger test of our interpretation that activity during delay 2 when *LookL* correspond to a re-evaluation of offer L, and thus to deliberation, requires additional analysis, so we do see the reviewer's point.

(Copy of reply to Reviewer 1, Comment 4).

To address this important question, we have performed two additional analyses (i, ii).

i) If reactivation implies reevaluation in *LookL* trials, then when the animal chooses L, we should see more fluctuation in activity of neurons with positive tuning to SV_L , and less activity in neurons with negative tuning. In other words, a positive fluctuation of activity of a neuron should increase the probability of the L choice if the tuning of the neuron is such that an increase of in SV_L leads to an increase of rate (positive modulation). We define fluctuations are the residuals $z_i = \eta_i - (\beta'_0 + \beta'_1 SV_{L,i} + \beta'_2 SV_{R,i})$ at the i^{th} trial, and study the relationship between β'_1 and choice. Note that SV s are now defined as $SV = \beta_1 EV + \beta_2 m + \beta_3 \sigma^2$; following suggestion by Reviewer 2, we have now included the magnitude among relevant regressors, corresponding to the best behavioral model to predict choice (Supp. Fig. S4).

Our prediction is confirmed by the results (Fig. 4F). We predict that neurons with positive tuning and positive activity fluctuations better predict L choices, and that neurons with negative tuning better predict opposite, R choices, when there are positive activity fluctuations. To simplify the visualization, we combined data based on the choice side (L, R) and the sign of β'_1 by defining 'preferred choice' (positive β'_1 and choice=L or negative β'_1 and choice=R) and 'non-preferred choice' units (positive β'_1 and choice=R or negative β'_1 and choice=L), analogous to what is done in the choice-probability literature. By comparing residual magnitudes in the two data pools, we find significantly larger residuals in preferred choice pools, confirming that the sign of fluctuations of residual activity during reactivation correlate with choice, and in the predicted direction. Since for every neuron we have both preferred and non-preferred choices, we used paired Wilcoxon tests, yielding significant difference ($p=0.00021$) for *LookL*.

In Fig. 4F (right) we show as a control, that repeating the same analysis in *LookR* trials leads to a non-significant relationship between noise fluctuations when aligning SV_L tuning with choice, in line with the lack of reactivation during *LookR*.

ii) Following the same idea as before, we can ask the same question but correlating the regressors of the linear model $\eta = \beta_0 + \beta_1 SV_L + \beta_2 SV_R + \beta_3 ch$ for each unit, where choice is encoded as $ch = +1$ for R, and -1 for L. Then, following the same reasoning as above, we expect that in *LookL* trials in delay 2, a neuron with positive β_1 would be associated with a negative β_3 (that is, more positive firing modulation to SV_L correlates to more negative modulation with choice). To avoid spurious correlations between regressors trained on the same dataset (Arandia et al., Neuron 2016)^[b], we estimated $\beta_1^{(1)}, \beta_3^{(1)}$ by using the first half of the trials, and $\beta_1^{(2)}, \beta_3^{(2)}$, by using the second half of the data, and then computed the correlation between $[\beta_1^{(1)} \beta_3^{(2)}]$ and $[\beta_3^{(2)} \beta_1^{(1)}]$ (combing β_1, β_3 in disjoint

sets). The results are shown in Fig. 4E, confirming the prediction of a significant negative correlation in *LookL* trials, which is absent in *LookR* trials, as expected.

We have described these new important results in Fig. 4E-F, in the last paragraph of Results *R4* and described the Methods in *M4.5*, *M4.6*.

We have added suggested references:

- [5] Conen, K. E., & Padoa-Schioppa, C. (2015). Neuronal variability in orbitofrontal cortex during economic decisions. *Journal of neurophysiology*, *114*(3), 1367-1381.
- [6] McGinty, V. B., & Lupkin, S. M. (2021). Behavioral read out of population value signals in primate orbitofrontal cortex. *BioRxiv*, 2021-03.

In the last paragraph of Results *R4*.

And added new references:

- [a] Britten et al., 1996 Britten, K. H., Newsome, W. T., Shadlen, M. N., Celebrini, S., & Movshon, J. A. (1996). A relationship between behavioral choice and the visual responses of neurons in macaque MT. *Visual Neuroscience*, *13*, 87–100.
- [b] Arandia-Romero, I., Tanabe, S., Drugowitsch, J., Kohn, A., & Moreno-Bote, R. (2016). Multiplicative and additive modulation of neuronal tuning with population activity affects encoded information. *Neuron*, *89*(6), 1305-1316.

In the last paragraph of Results *R4* and in Methods *M4.6*.

I think there might several ways of dealing with this. One might simply be to tone down the claims made and to be clearer in the text if there is a limitation in the strength of the evidence. Perhaps delay-related activity does predict choice but not in a manner that is related to the direction of gaze. Perhaps the authors could reconsider the manner in which they separate out trials into the black and grey traces in the bottom panel of 4c. If I understand correctly they are separated on the basis of whether the animal looks more one way than another but perhaps a more exacting criterion could be used such as focussing on trials when animals spend more than 50% as much time looking one way than the other. Or, alternatively, perhaps the authors could focus on the final part of the last delay and condition the analysis on that period.

We hope to have addressed this comment by major rewriting of Results and Discussion and through the previous analyses (i, ii), which lend further support to the idea that reactivation of activity during *delay 2* reflects an evaluation process.

Minor point

“Further, to test whether evaluation is sequential by default, we let subjects to move their eyes at will, and expect tracking the internal deliberation process by using the look-at-nothing effect ...” sounds a bit awkward. Maybe it should be something like: “Further, to test whether evaluation is

sequential by default, we let subjects move their eyes at will, and expect to be able to track the internal deliberation process by using the look-at-nothing effect ...”

We thank the reviewer for suggestion, which we have followed in the next version of the manuscript (third paragraph of Introduction).

REVIEWER COMMENTS

Reviewer #1 (Remarks to the Author):

The revisions and additional analyses satisfy many of the concerns with the original manuscript. However, there are still remaining concerns regarding points 3 and 4 in my original review, details of which are below.

To summarize:

- The transition-locked analysis shown in Fig. S14 partially supports the claim of gaze-triggered value coding, but also suggests a different potential mechanism that could explain some of the findings. My opinion is that the claim of gaze-triggered reactivation of value signals is believable with modest confidence.

- The choice encoding analysis seems to be only half complete (using LookL trials, but not LookR trials), and also has some minor weaknesses that should be addressed. My opinion is that the claim linking reactivated value signals to choice is believable with only low confidence, but this could be improved with additional analyses.

Taken together, my opinion is that the findings do suggest potentially interesting and significant results. However, my confidence in these results is modest at best, given the small effect sizes, the modest number of neurons available (248), and the remaining analysis/interpretational issues discussed below.

Details:

First, Supplemental Figure 14 shows the fraction of cells encoding values when the data are time-locked to the moment that the gaze shifts from one side of the screen to the other. Panel E (left) of this figure shows that during delay2, when the gaze transitions to the location of the target shown first in the trial, there is a small but significant fraction of cells that encodes the value of the first target. This is a fairly convincing demonstration of the main claim of the study – i.e. that looking at nothing reactivates a representation of a previously looked-at target. However, the other results from this figure do not strongly support this main claim: During delay1, shiftL events do not lead to an increase in EV_L encoding (panel D, left). And during delay2 there may be a small increase in EV_R after shiftR events (Panel E, right), but the magnitude and duration of the significant encoding is quite small, so it's unclear how reliable this result is. (The response to reviews points out that during delay1, shiftR events coincide with a decrease in EV_L, and likewise that during delay2, shiftL events coincide with a decrease in EV_R encoding. However, it's not clear that this is

reflecting a viewing-triggered process, because at least some of that pre-shift encoding likely occurs during the offer1 and offer2 period, when the two targets are visible.)

In sum, the results from the gaze-shift-triggered encoding analysis are somewhat ambiguous: There is solid evidence that during delay2, shifting gaze towards the empty first target location triggers an increase in the encoding of the first target value (S14, panel E left, dark orange), but there is not solid evidence that this gaze-triggered encoding occurs during delay1, or during delay2 for the second target.

My thanks to the reviewers for providing a complementary analysis, showing the encoding on trials where the gaze *does not* shift during the delay, because these are particularly revealing: During delay1, there is a convincing fraction of cells that encodes EV_L when the gaze stays on the left (S14, panel I left, dark orange), and a very similar convincing effect is seen during delay2 for EV_R when the gaze stays right (S14, panel J right, light purple). These complementary results suggest a clear association between gaze and value encoding, but not necessarily one in which gaze triggers the encoding of value.

Considering both of these supplemental analyses together, it seems possible that there are two kinds of gaze-related encoding effects that are driving the results: One effect that is triggered by shifts of gaze and that pertains to previously viewed offer location (positive effect in S14 E left) but not the offer that was most recently shown on the screen (negative effects in S14 D left and S14E right). This first effect is consistent with a “re-activated” value representation. The second effect pertains to a just-offered item location (positive effects in S14 I left and S14J right) when gaze does not shift away. This second effect might best be characterized as a “maintained” rather than “reactivated” value representation, with no clear indication of whether gaze causes a maintained value representation or whether it reflects it.

To sum up my thoughts on this particular claim: The paper’s main claim is that there is a gaze-triggered re-activation of value signals in OFC. Closer examination – in particular Figure S14 – partially supports this interpretation, but is also consistent with a different effect in which maintained gaze and maintained value signals coincide, with an unclear causal relationship. I think that the supplemental figure should be made a main figure, and the complexity of the data be discussed.

Second, the added analyses showing a relationship between value encoding and choice is intriguing, but could be improved in several ways.

- The analyses in Fig. 4D and E seem to be exclusive to LookL trials and the encoding of EV_L. Shouldn’t it be possible to also compute a complementary effect on LookR trials with EV_R

encoding? I.e. to compute the B_2 vs. B_3 correlations, and to sort trials according to the preferred/non-preferred encoding of SV_R? Why was this analysis not performed? It seems that the effects related to LookR/EV_R are just as important for this point, and if they aren't consistent with those from the LookL/EV_L analysis, then it complicates the interpretation of this result.

- Fig. 4E shows a correlation between regression estimates for value and choice, indicating the cells that encode value also encode choice, with the signs of encoding indicating that the choice and value signals are congruent – i.e. cells that encoding SV_L with a positive sign should encode a rightward choice with a negative sign. To make this analysis airtight, it would be best if the choice effect (B_3) were estimated over the residuals of a model fit to SV_L and SV_R. In other words, first fit a model that allows for SV_L and SV_R to explain all of the value-related variance it can, and then show that choice explains nontrivial residual variance and, critically, that the regression estimate for choice has the expected correlation with B_1.

- The correlation statistic in Fig. 4E appears to be a Pearson's correlation (line 921), which gives too much leverage to outliers. The result should be confirmed with a Spearman's or other outlier-resistant correlation statistic.

- The analysis in Fig. 4F is an interesting approach to showing correlation between choices and value encoding. I would like to recommend a few adjustments to help make this analysis better aligned with previous work:

- Analyses of this kind are most informative when considering trials in which choices are not certain (e.g. consider the "low coherence" trials in classic dot motion discrimination studies). I would like to see this result confirmed when using only trials taken from the range of EV differences that produce variable choices (e.g. EV differences of +/-1).

- By pooling the trials from all cells into a single test, the statistical power for detecting weak effects is improved. However, it would be informative if the effects for single cells were calculated, perhaps as a supplement.

- The overall magnitude of these effects is unclear, relative to other studies. Is it possible to express the difference in distributions in Fig. 4F in terms of an ROC or other standard measure of classification accuracy?

Reviewer #3 (Remarks to the Author):

The authors have addressed all my comments.

Reviewer #2 (Remarks to the Author):

I sincerely appreciate the authors' efforts to satisfy the reviewers' concerns and recognize that significant work has been done. As a result, the manuscript is clearer and the bases for the paper's claims are more thoroughly shown in supplemental figures. Unfortunately, a few fundamental issues still have not been resolved and represent weakness in the claims being made. To adequately support the paper's core claims, I believe changes to the manuscript's fundamental analysis are necessary, which would require a significant revision. I have attempted to outline these recommended changes below.

1. The paper's core claim concerns the interaction between where an animal is looking and the representation of value- and choice-related signals in OFC. Because gaze is a series of relatively long periods of approximately static fixation interspersed by rapid changes in eye position (i.e., saccades), this interaction predicts that OFC activity should be related to the object of gaze during periods of fixation as bookended by saccades, and likewise be *less* related before and after the fixation-acquiring and fixation-leaving saccades, respectively. The prior work on gaze-OFC interactions—as cited in the present manuscript^{1,2}—adheres to this logic by analyzing OFC activity aligned to changes in fixation. That is, how does OFC activity change when gaze is directed to—and away—from the stimulus location of interest?

Indeed, in response to Reviewer 1, the authors perform and present the appropriate “gaze-aligned” analysis for a small portion of their overall claims in the revised supplement (Fig S14). However, the entirety of the data presentation in the main text, as well as the vast majority of claims and statistical hypothesis testing, rely on a very different analysis in which OFC activity is related to *average* gaze position. This alternative approach deviates from the established precedent in OFC^{1,2} (not to mention in visual systems neuroscience) and, in my view, suggests a distorted logic and introduces several interpretive challenges that would be obviated by a gaze-aligned analysis (see below points). Moreover, when the recommended gaze-aligned analysis was done (Fig S14), the support for the paper's main claims was much less robust (see (2)).

- a. The logic of the present paper—as in past work on visuospatial attention—is that gaze is a marker for the locus of attention. For example, say gaze were directed at position “-1” at time point A and at position “+1” at time point B. We might say that attention was directed at “-1” *or* “+1” depending on the time point. But if we were to average gaze across the two time points $([-1, +1] = 0)$, this does *not* imply that attention was, on average, directed at position 0. Indeed, at no point was animal attending to middle of screen! And yet, the vast majority of analyses in the present paper rely on averaging gaze position over some period and then relating the average gaze to OFC activity. (It's possible the authors believe that

the neural process under study reflects the average of otherwise bimodal attentional selection. But this would be a new hypothesis that would have to be convincingly proven before using it as an analysis tool.)

- b. A number of interpretative challenges arise using the average-gaze analysis. One discussed at length in the prior round of reviews concerns the size, overlap, and therefore number of non-independent time bins. The authors helpfully explained their rationale for using very narrow time bins (10 ms) for eye position, and small-stepped (+10ms) overlapping (95%) time bins for neural data, as needed to capture the fast dynamics of eye movements. While it's true that eye position can change very rapidly during saccades (on the order of 10 ms) these events are readily detectable and are relatively rare compared to the long periods of static fixation (200-300 ms). Therefore, the need to isolate static periods of eye position can be achieved by aligning neural data to large changes in visual fixation (i.e., saccades) and analyzing neural activity in a small handful of time bins on either side of the eye movement. This would not only achieve the authors' goal of isolating static periods of eye movements, it would also presumably strengthen the analysis as variation in eye position would be substantially reduced during the period of neural analysis (because the period was *defined by* fixed eye position).

- c. The gaze-aligned analysis would also address a challenge in interpretation arising from uncertainty in the timecourse of *when* a change in eye position would impact neural activity. The present analysis, if I understand correctly, pairs a 10ms window of eye position with a 200ms window of neural activity, which (indirectly) hypothesizes that the effect of gaze on OFC activity occurs as early as immediately coincident with the change in eye position and that the bulk of the effect occurs within 200ms. From prior literature^{1,2}, it appears the OFC value-related response begins ~150ms after a change in gaze and usually lasts 100's of ms more. By using a gaze-aligned analysis, the precise nature of this timing becomes less critical, as one could use relatively large, non-overlapping time bins to capture the effect (e.g., 0 – 200ms, 200 – 400ms, 400 – 600ms, etc. after fixation). Moreover, because only a handful of time bins would be necessary, the previously discussed statistical concerns around multiple comparisons are significantly reduced. (As an aside, in their rebuttal, the authors justify the use of highly overlapping, sliding time windows by citing one of their own papers and one by Conen and Padoa-Schioppa. The latter used sliding windows to effect smoothness for visualization, while their statistical analyses were limited to a handful of pre-defined, non-overlapping windows, one for each epoch of the trial.)

In summary, analyzing and presenting the neural data by aligning to changes in gaze would significantly strengthen the interpretability of the data and therefore the ability to validate the paper's central claims. Moreover, the gaze-aligned analysis would align the

paper with past studies of gaze-and-OFC and reduce, if not eliminate, many secondary problems discussed in the first round of reviews. Though I did not recommend the gaze-aligned analysis in my first review, Reviewer 1 did recommend this analysis explicitly and, after the authors clarified certain points in their rebuttal, I believe the gaze-aligned analysis is essential both because it directly addresses the core questions and because it addresses many of my concerns from the first round.

While the authors show a gaze-aligned analysis for a subset of the main claims in the supplement (S14), I believe the analysis is sufficiently important to be applied to all main claims and be prominently presented in the main text. Finally, additional issues exist with the gaze-aligned analysis in its current form that should also be addressed (see below).

2. In Figure S14, the authors analyze the OFC data aligned to changes in gaze, which is much appreciated, as discussed above. Even still, there are additional issues to address.
 - a. Perhaps the most significant challenge of the gaze-aligned results is that they are much less compelling than the non-aligned original analysis in the main text. The percentage of significant cells is only just above the population-level significance threshold during the predicted times and is often only just below the threshold during the remaining times. In contrast, in the non-aligned analysis (Fig 3), the results are well above the significance threshold and remain so for 100s of ms during the predicted epochs.

In addition, several specific predictions did not bear out in the gaze-aligned analysis:

- i. During delay1, when gaze changes R to L, there is no emergence of EVL encoding (D, left, dark orange). (The authors speculate this is because the subjects had not sampled offer1 during the offer1 period. If this were true, that should be demonstrated explicitly.)
 - ii. During delay2, encoding of EVL is not significant while gaze is directed to the left (except for during a vanishingly tiny period) and likewise cannot decay when gaze changes L to R (E, left, light orange).
 - iii. Of note, the latter observation, and combined with the lack of EVL encoding during uninterrupted viewing of the left side (J, left, dark orange), are inconsistent with the main thrust of the paper, and specifically the previously most compelling result (albeit using the non-aligned analysis) that EVL encoding emerges during delay2 conditioned on subject previously looking Right prior to delay2 and later looking Left during delay2 (main Fig 3G, left, dark orange).
- b. The fact that the gaze-aligned results are much less robust than the non-aligned analysis suggests one of two possibilities:

- i. Contrary to the paper's claim, OFC activity is not (robustly and consistently) dependent on the interaction between recent stimulus value and current gaze. And instead, OFC activity, during the delay period in particular, is better explained by another variable.

Or...

- ii. OFC does indeed represent the value of the stimulus located at (or recently at) the location of gaze, but that too many sources of variance obscure this finding in the current analysis. In this case, it's possible that a refined gaze-aligned analysis would reveal the hypothesized result and support the paper's main claims (which would be very interesting indeed!).

One suggestion is to perform the neural analysis in wide, non-overlapping time bins, as discussed above in point (1). This would obviate the need for the various bootstrapping and multiple comparison correction from the main text and may strengthen statistical power. It may also be less sensitive to the various dynamics between gaze and OFC response.

Another possibility is that the effect is more precisely sensitive to the overlap between gaze and (recent) offer location, and the analysis may be strengthened by accounting for whether (or how close) gaze was on the (recent) offer location. (A rough sense for this sensitivity could be gleaned by performing the gaze-aligned analysis during the offer period as a baseline for comparison; see below).

- c. One would like to see the gaze-aligned analysis for the offer periods, as well, when presumably the change in encoding would be highly sensitive to the change in gaze (toward or away). This would provide useful contextualization for the robustness of the delay-period analysis.
- d. The authors present a delay-aligned analysis (S14F-J), perhaps as a point of comparison for the gaze-aligned analysis. However, by averaging over multiple saccade-fixation sequences, this analysis still suffers from most of the concerns raised about the other non-aligned analyses in the main text. (The exception is that in S14J-F, the gaze is at least consistently on one side of the screen).

A preferred comparison is to align all analyses to the beginning of fixation, but then to compare the cases when the preceding fixation was on the same stimulus location or screen side (i.e., control condition) to when the preceding fixation was on the other location/side (i.e., test condition). This would control for the effect of changing fixation and isolate the OFC response due to a change

in value of the current/recent stimulus.

- e. It was not clear whether the gaze-aligned analyses excluded timepoints occurring either after gaze has changed (this would be most rigorous) or at least after gaze has shifted back across the midline (such as illustrated in the example traces in S14A at ~400ms). In addition, were timepoints after the end of the present delay period excluded? These exclusions would greatly increase interpretability, and also obviate the need to condition every time bin on eye position.
 - f. Finally, the gaze-aligned analyses were only done for the EV-encoding questions. The gaze-aligned analyses should be performed for all results, especially the critical result of choice-related activity (i.e., Fig 4 C bottom, E, and F).
3. The paper's core claim is that look-at-nothing gaze correlates with a "reactivation" of the value representation in OFC, *and* that this "reactivated" value representation is used to drive the eventual decision. The authors support this claim by showing that delay-period OFC activity correlates with stimulus value conditioned on gaze (e.g., Fig 4D) and *separately* that trial-to-trial variability in OFC activity is predictive of choice (i.e., Fig 4F). From the present analysis, it is not clear whether the choice-predictive variance is related to the same variance that correlates with value or to some orthogonal source of variance. This distinction is critical for the authors' claims. If the "reactivated" value representation is driving choice (as claimed), then it must be the value-related component of the OFC response that is predictive of choice.

This question is best illustrated at the population level, where one would test whether the dimension that best explains value is aligned/orthogonal to the dimension that best predicts choice.

For sequential recordings, as in the present dataset, Padoa-Schioppa offers a potential approach to this question: cull out the cells most sensitive to value and compare their choice predictivity to the remaining cells. Or more generally, test the overlap between the strength of choice predictivity and strength of value encoding.

4. The paper labels the value-encoding delay-period activity as "reactivation" with the interpretation that it is a retrospective recalling of the formerly presented offer. In the first round of reviews, both Reviewer 1 and myself asked the authors to address an alternative interpretation. Specifically, whether the signal could also be interpreted as a prospective signal, that is, expected value given the choice. In their rebuttal, the authors argue against the signal being motoric or categorical, and therefore seem to dismiss the prospective interpretation. However, the signal need not be categorical to be prospective in nature. Moreover the prospective interpretation is not merely theoretical. Rather, it aligns with decades of past work in OFC reporting signals that *continuously* represent value conditioned on choice, or what Padoa-Schioppa et al. called "chosen value"³. (They also reported a distinct *categorical* choice signal, termed "chosen offer".)

The distinction between the retrospective “reactivation” and the prospective “chosen value” is essential and any evidence to resolve these interpretations would be most welcome. Short of that evidence, the present paper must acknowledge the ambiguity and assimilate the authors’ interpretation with the past literature.

5. As discussed, a core claim is that the delay period OFC activity is predictive of choice. This is shown in two ways, each with some concerns.
 - a. Most of the choice-related analyses are based on a regression model in which choice is a predictor (Fig 4C,D,E). The problem with this approach is that regression methods struggle to precisely segregate variance between correlated predictors, such as value and choice. A preferred approach would be to assess how much additional variance is explained by choice after attempting to explain as much variance as possible using the other predictors. This can be done by comparing two nested models that differ only by the inclusion of the choice predictor (see “coefficient of partial determination”).
 - b. The most rigorous test of choice predictive activity is in Fig 4F, which is cited extensively in the authors’ rebuttal. As the authors state in both the paper and their rebuttal, the widely accepted method for this analysis is the so-called “choice probability” (CP) metric, as per Britten et al.⁴, which is based on a non-parametric comparison of two putative pools of neural evidence for either of the two choices and is reported in terms of the probability of making a given choice given the evidence. This makes CP a particularly useful metric both because it allows comparison to many past papers and offers an absolute value that is readily interpretable on its own. It seems that the authors are familiar with this literature, and so it wasn’t immediately clear why the CP analysis was not used and reported.

6. The authors introduce a model of subjective value (SV), The trouble is that SV is at once a computed value based on a model fit to choices *AND* SV is a predictor in a separate model of firing rate that *also* includes choice as a predictor. This could lead to spurious interactions between free parameters in the firing rate model. There are at least a couple ways to mitigate this confound:
 - a. Is it necessary to have a separate model of SV? Would it be possible to simply include the relevant terms on which SV is based (m , σ^2 , $EV=m*\sigma^2$) as stand-alone regressors in the firing rate model?
 - b. If SV is necessary as a computed value, cross-validation should be used to isolate the free parameters in the SV model from those in the firing rate model. That is, compute the SV term (which depends on free parameters $w_{\{1,2,3\}}$) using separate trials from those used in the model to predict firing rate (in which the independently computed SV could then serve as a predictor). The authors note that they use cross-validation, but this is at the earlier level of computing SV, not in applying SV to predict firing rate.

7. As discussed in the first round of reviews, the rationale for separate behavioral models for each temporal epoch still eludes me, particularly since the vast majority of predictors (e.g., EVL/R, mL/R, sigma2, sL/R) are identical across epochs (Fig 2C). The authors (thankfully) also provide the preferred approach in which a single model contains one copy of each of the static predictors and separate predictors for the one variable (f_R) that differs across epochs (Fig 2D). The problem with presenting both models is that they draw conflicting conclusions, and yet the authors point to both to support their claims. In particular, f_R is not significant in most epochs in the combined model (2D), and yet the authors point to each of the epoch-specific models to claim that f_R is significant in all the epochs (lines 195-198).

The problem with the single-epoch model is that 1) it is effectively a multiple comparisons problem (repeating the same test on the same, or highly overlapping, data), and 2) does not account for within-trial interactions (e.g., how f_R during epoch X may compete for variance with f_R during epoch Y), whereas the full model does account for these interactions (which indeed is likely why f_R delay 2 survives in the full model, but the other f_R for the other epochs do not).

The authors say the separate models are justified because starting with the full model would be “too much of a jump”. I suppose that this is a subjective assessment, but to at least this reviewer, the 11 bars in Fig 2D are simpler than interpreting and comparing the 8 bars/epoch * 6 epochs = 48 bars in 2C!

The authors also justify the separate model to mirror the visualization in Fig 2A. I agree with the visualization approach in Fig 2A. But the rationale for a particular visualization approach does not necessarily justify the same approach for the statistical analysis. On the contrary, it is typical to use one approach for visualization and another for rigorous analysis. In fact, Fig 2A and B are great examples of this, where trials are stratified into separate, discrete groups to allow for plotting multiple variables on a single axis, whereas the statistical analysis would include all variables in their continuous form (as in Fig 2C,D).

If authors are looking for more statistical power to test f_R , they could include an “overall f_R ” term, or “ f_R in offer₁₊₂+delay₁₊₂”. In the spirit of gradually building a model up, this would be more typical: a simple model that pools all epochs together, then a more complex model that has separate *terms* (not *models*) for each epoch.

As a separate point, it's unclear to me why the authors use the difference of offers for plotting (e.g., EVR-EVL) in Figs 2A,B — which implies that the difference is the critical variable for choice, and makes total sense! — but use the independent, absolute values of the offers (e.g., EVR, EVL, etc.) in the model. Why not include the difference terms in the model? If one thought that absolute magnitude mattered in addition to the difference in value, one could include a term EVR+EVL. Besides making things more consistent across analyses and making better intuitive sense, this model might actually

explain the data better and (possibly) help pull out the variables of interest.

8. The authors use the term “reevaluation” to refer to the choice-predictive activity during the delay period. However, it was not clear why this process constitutes “Reevaluation” and not merely “evaluation”. That is, what distinguishing the phases of “evaluation”, “reevaluation”, and in some cases “further reevaluation” in the present paper? The distinction between evaluation and reevaluation is important because there is a large literature on REevaluation in which it is shown that subjects have reached a decision, but then change that decision (usually in the face of new information). But identifying REevaluation requires some evidence that a decision was (tentatively) reached. It would be helpful if the authors could highlight this evidence.
9. I agree with Reviewer 1 that re-coding the offers as “Left” and “Right”, even when the offer was presented on the opposite side as its label, is confusing, and seemingly unnecessarily so. Why not just refer to the offers as they actually are—“1st” and “2nd”—and then combine across sides for most analyses? This would seem advantageous not only because it is a more accurate description of the actual task, but it also would avoid certain confusing exceptions to the current rule, such as when analyzing the effect of side (e.g., Fig 2C), when “Left” really does mean “Left”!
10. Fig 2C,D: colors are difficult to distinguish, particularly S_LR and f_R vs. EV_L and m_L (2C) and f_R offer 1/2 vs. EV_L and m_L (2D).
11. Line 111: the definition of variance includes the reward magnitude (“ $mp(1-p)$ ”? Typically, Bernoulli variance is “ $p(1-p)$ ”, no?

References

1. McGinty, V. B., Rangel, A. & Newsome, W. T. Orbitofrontal Cortex Value Signals Depend on Fixation Location during Free Viewing. *Neuron* **90**, 1299–1311 (2016).
2. McGinty, V. B. & Lupkin, S. M. Behavioral read-out from population value signals in primate orbitofrontal cortex. *Nature Neuroscience* **26**, 2203–2212 (2023).
3. Padoa-Schioppa, C. & Assad, J. A. Neurons in the orbitofrontal cortex encode economic value. *Nature* **441**, 223–6 (2006).
4. Britten, K. H., Newsome, W. T., Shadlen, M. N., Celebrini, S. & Movshon, J. A. A relationship between behavioral choice and the visual responses of neurons in macaque MT. *Vis Neurosci* **13**, 87–100 (1996).

RESPONSES TO REVIEWER COMMENTS

We have now reported all comments in this file followed by point-by-point responses, denoted with left vertical bars and indentation.

Reviewer #1 (Remarks to the Author):

The revisions and additional analyses satisfy many of the concerns with the original manuscript. However, there are still remaining concerns regarding points 3 and 4 in my original review, details of which are below.

To summarize:

- The transition-locked analysis shown in Fig. S14 partially supports the claim of gaze-triggered value coding, but also suggests a different potential mechanism that could explain some of the findings. My opinion is that the claim of gaze-triggered reactivation of value signals is believable with modest confidence.
- The choice encoding analysis seems to be only half complete (using LookL trials, but not LookR trials), and also has some minor weaknesses that should be addressed. My opinion is that the claim linking reactivated value signals to choice is believable with only low confidence, but this could be improved with additional analyses.

Taken together, my opinion is that the findings do suggest potentially interesting and significant results. However, my confidence in these results is modest at best, given the small effect sizes, the modest number of neurons available (248), and the remaining analysis/interpretational issues discussed below.

We thank the reviewer for a thorough review of our work. We hope to provide convincing additional support to our findings in the replies below.

Details:

First, Supplemental Figure 14 shows the fraction of cells encoding values when the data are time-locked to the moment that the gaze shifts from one side of the screen to the other. Panel E (left) of this figure shows that during delay2, when the gaze transitions to the location of the target shown first in the trial, there is a small but significant fraction of cells that encodes the value of the first target. This is a fairly convincing demonstration of the main claim of the study – i.e. that looking at nothing reactivates a representation of a previously looked-at target.

We thank the reviewer for agreeing on the interpretation of the gaze-locked reactivation. Note that we have moved the previous Supp. Fig. S14 from supplementary information to main text as new Fig. 4 and detailed the findings at the end of third section of Results.

However, the other results from this figure do not strongly support this main claim: During delay1, shiftL events do not lead to an increase in EV_L encoding (panel D, left).

We think that our results do not contradict with the interpretation we proposed. Rather, they are expected: shift L events during delay1 time imply that subjects were previously looking R (at least) at the end of offer1 period, and thus EV_L is not previously encoded (or weakly encoded, eventually coinciding with peripheral vision). Therefore, we expect to have little or no encoding of EV_L before and after the gaze shifts from R to L. Further, we do not expect to see encoding of EV_L in trials where subjects shift their gaze from L to R (shift R) during delay1 time, as observed.

We have now added Fig. 4 to the main text (previous Supp. Fig. S14) and detailed all results by clarifying this point at the end of third section of Results.

And during delay2 there may be a small increase in EV_R after shiftR events (Panel E, right), but the magnitude and duration of the significant encoding is quite small, so it's unclear how reliable this result is.

We apologize for the eventual misinterpretation. During delay2, after a shift R event (shift from left to right), we do not expect encoding of EV_R . In this case, the subject was previously looking at the left (at least) at the end of offer2 epoch time, hence the value of the R offer is not previously encoded since the gaze was directed to opposite screen side. In similar way as for previous reply, we thus expect that there was weak or no encoding of EV_R , regardless of where the animal looks afterwards.

We have also clarified this point at the end of third section of Results.

(The response to reviews points out that during delay1, shiftR events coincide with a decrease in EV_L , and likewise that during delay2, shiftL events coincide with a decrease in EV_R encoding. However, it's not clear that this is reflecting a viewing-triggered process, because at least some of that pre-shift encoding likely occurs during the offer1 and offer2 period, when the two targets are visible.)

As argued in the previous two comments, we find those decreases meaningful, precisely because after the shift it is expected that there will be a decrease in the encoding to n.s. level of the immediately previous offer if the subjects shift gaze away from the offer presentation side.

Note that in the previous Supp. Fig. S14F-I (current Fig. 4D-E) we find key aspects for the rest of the answer, as also the reviewer notes below: in trials where the subjects look L during offer1 epoch time, the encoding of EV_L persists whenever subjects keep looking on L

screen side. This is the exact reason why we decided to include “exclusively look on either side” panels (current Fig. 4D-E).

Also, we have now included analyses equivalent to current Fig. 4 but for gaze shifts that occur during offer1 / offer2 epochs in Comment Figure below, also added to the Supplementary Information as Supp. Fig. S15.

By using gaze-shifts time-locking during offer presentation, we confirm the previously shown result (Supp. Fig. S1C) that shifts are predominant towards left for first offer presentation time (Comment Figure 1B) and towards right for second offer presentation time (Comment Figure 1C). As expected, we find that gaze shifts to left screen side during the *offer1* epoch coincide with post-shift increase in encoding of EV_L (Comment Figure 1D, dark orange) while gaze shifts to right screen side at *offer2* coincide with post-shift increase in encoding of EV_R (Comment Figure 1E, light purple). Conversely, shifting gaze to the opposite direction with respect to first offer presentation side leads to decrease in encoding of EV_L (Comment Figure 1E, light orange; note that it is the most frequent case). Note that the latter effect is not possibly found for EV_R as it was never presented before offer2 time.

For completeness, we also include below (Comment Figure 1) results for trials where no shifts occurred during the offer epochs (pre-offer time could include shifts but they were not relevant for the respective presented offer encoding since respective offers had never been displayed before).

We can observe here pronounced encoding of ipsilateral offer whenever subjects pre-emptively looked at presentation screen side. As a remark, we find that the increases for EV_L (Comment Figure 1I, dark orange) and EV_R (Comment Figure 1J, light purple) temporally coincide with gaze relocations *within* screen sides (no screen midline crossing) towards target offer locations (Fig. 1C). Importantly, the decrease in first offer encoding before second offer is presented in look left trials (Comment Figure 1J, LookL, dark orange) can indicate that second offer acts as a distractor when gaze is not directed to its side, though we could not investigate this aspect further as it is rather infrequent.

We did not previously go in depth in illustrating this quite complex scenario, but we have now enriched last paragraph of third results section in current version. We hope that results and additional analyses help clarify our findings and interpretation.

Comment Figure 1: A, D-E, F, I-J. Same as Fig. 4 but for offer epoch times. B-C, G-H. Fractions of trials available used in the respective panels below.

In sum, the results from the gaze-shift-triggered encoding analysis are somewhat ambiguous: There is solid evidence that during delay2, shifting gaze towards the empty first target location triggers an increase in the encoding of the first target value (S14, panel E left, dark orange), but there is not solid evidence that this gaze-triggered encoding occurs during delay1, or during delay2 for the second target.

We hope to have provided convincing enough answers above, and we apologize if this was not sufficiently clear before.

My thanks to the reviewers for providing a complementary analysis, showing the encoding on trials where the gaze *does not* shift during the delay, because these are particularly revealing: During delay1, there is a convincing fraction of cells that encodes EV_L when the gaze stays on the

left (S14, panel I left, dark orange), and a very similar convincing effect is seen during delay2 for EV_R when the gaze stays right (S14, panel J right, light purple). These complementary results suggest a clear association between gaze and value encoding, but not necessarily one in which gaze triggers the encoding of value.

We thank the reviewer for pointing out the weakness of our previous explanations on all the above points.

Considering both of these supplemental analyses together, it seems possible that there are two kinds of gaze-related encoding effects that are driving the results: One effect that is triggered by shifts of gaze and that pertains to previously viewed offer location (positive effect in S14 E left) but not the offer that was most recently shown on the screen (negative effects in S14 D left and S14E right).

We want to remark again that in previous Supp. Fig. S14D (current Fig. 4B) for delay1 time, it is most likely that the first offer side has not been previously gazed directly at offer1 time in shift L trials, and thus EV_L has been only weakly (transiently) or not encoded at all before the shift. We want to recall that in Fig. 4B, just before the shift (hence at the end of offer1) the subjects were looking at the right, where no offer was presented (empty screen). Analogously, note that in previous Supp. Fig. S14E (current Fig. 4C) there is no (or weak, transient) encoding of EV_R in trials where the subjects were looking at the left during offer2 epoch time, and then shift R during delay2 time. All these results, along with the above ones, are consistent with our proposed interpretation of gaze-centered encoding of alternative offer values in OFC.

We have further clarified this by adding previous Supp. Fig. S14 in the main text as Fig. 4 and detailed all these points at the end of third section of Results.

This first effect is consistent with a “re-activated” value representation. The second effect pertains to a just-offered item location (positive effects in S14 I left and S14J right) when gaze does not shift away. This second effect might best be characterized as a “maintained” rather than “reactivated” value representation, with no clear indication of whether gaze causes a maintained value representation or whether it reflects it.

To sum up my thoughts on this particular claim: The paper’s main claim is that there is a gaze-triggered re-activation of value signals in OFC. Closer examination – in particular Figure S14 – partially supports this interpretation, but is also consistent with a different effect in which maintained gaze and maintained value signals coincide, with an unclear causal relationship. I think that the supplemental figure should be made a main figure, and the complexity of the data be discussed.

We thank the reviewer for the thoughts provided. We agree that while maintaining the eyes on the same side it is not possible to determine whether OFC activity reflects a memory

trace or a reactivation. We hope that our explanation above convincingly supports our gaze-centered view of value encoding in OFC.

We have now made previous Supp. Fig. S14 a main figure (current Fig. 4) and explained at the end of third Results section the results, elaborating on the complexity of configurations that unfold when considering all possible shift cases.

Second, the added analyses showing a relationship between value encoding and choice is intriguing, but could be improved in several ways.

- The analyses in Fig. 4D and E seem to be exclusive to LookL trials and the encoding of EV_L.

We believe that the reviewer referred here to analyses in previous Fig. 4E-F (current Fig. 5, panels E-F have been fully updated to new methods). For such analyses, we indeed only included SV_L encoding, but showed separately *LookL* and *LookR* trials, to show that only in *LookL* trials there was an effect of “looking to first offer site” on residuals. We have further commented on these aspects in below responses.

Shouldn't it be possible to also compute a complementary effect on LookR trials with EV_R encoding? I.e. to compute the B_2 vs. B_3 correlations, and to sort trials according to the preferred/non-preferred encoding of SV_R? Why was this analysis not performed? It seems that the effects related to LookR/EV_R are just as important for this point, and if they aren't consistent with those from the LookL/EV_L analysis, then it complicates the interpretation of this result.

The same figure has now been reported as Supp. Fig. S14B (with minor edits to the subsampling for comparisons, Methods 4.4, 4.5) by also including SV_R . On further consideration, we realized that cell-wise (hence time- and trial-averaged) comparison of residuals tended to be weak since the Least Squares estimation leads the residuals with zero mean along the trial dimension. We have moved the previous Fig. 4 to current Fig. 5. We have now extended the analysis to (time-averaged) cell-wise, trial-to-trial residuals comparison reported both in terms of Choice Probability (new Fig. 5E-F) and by performing cell-wise ROC analysis (Supp. Fig. S14A).

- Fig. 4E shows a correlation between regression estimates for value and choice, indicating the cells that encode value also encode choice, with the signs of encoding indicating that the choice and value signals are congruent – i.e. cells that encoding SV_L with a positive sign should encode a rightward choice with a negative sign. To make this analysis airtight, it would be best if the choice effect (B_3) were estimated over the residuals of a model fit to SV_L and SV_R . In other words, first fit a model that allows for SV_L and SV_R to explain all of the value-related variance it can, and then show that choice explains nontrivial residual variance and, critically, that the regression estimate for choice has the expected correlation with B_1.

We have now removed this analysis for the manuscript to avoid overloading, and because the choice probability analysis proposed further below seems to yield deeper and more robust results (see next).

- The correlation statistic in Fig. 4E appears to be a Pearson's correlation (line 921), which gives too much leverage to outliers. The result should be confirmed with a Spearman's or other outlier-resistant correlation statistic.

By scrutinizing the results further, we agree with the reviewer that previous correlational analyses might have been too weak to data outliers. By applying Spearman's correlations, we did not find significant differences between *LookL* and *LookR* conditions. The same was found when attempting removal of outliers (e.g., discarding betas >1.5 their cell-wise standard deviation) or when considering only data from cells with significant linear slopes. Thus, we have removed correlation analyses (previous Fig. 4E) and provided major improvements to cell-wise trial-to-trial residuals comparisons Choice Probability (CP) analyses (current Fig. 5E-F) and cell-wise ROC analyses (Supp. Fig. S14A). We find that residuals convey choice-probability signals aligned to SV encoding sign, and that most strongly do so during the *LookL* condition, that is during reactivation (Fig. 5).

Based on changes in methods and new results, we have fully updated Methods 4.5, added Methods 4.6, 4.7 and related results section.

- The analysis in Fig. 4F is an interesting approach to showing correlation between choices and value encoding. I would like to recommend a few adjustments to help make this analysis better aligned with previous work:

- Analyses of this kind are most informative when considering trials in which choices are not certain (e.g. consider the "low coherence" trials in classic dot motion discrimination studies). I would like to see this result confirmed when using only trials taken from the range of EV differences that produce variable choices (e.g. EV differences of +/-1).

Re-running spike rate model fits and repeating the analyses in previous Fig. 4F (current Supp. Fig. S14B, with minor edits to sub-sampling, Methods 4.4, 4.5) but for trials with EV difference in the range [-1, +1] we find approximately the same results (See Comment Figure 2 below).

We also re-run Choice Probability and cell-wise ROC analyses as in main analyses (Methods 4.6, 4.7) but only including trials for EV difference within [-1, +1] (not shown to prevent overloading). This led to replicate main results in new Fig. 5E-F and Supp. Fig. S14B with increased difference in the median CPs for the two looking conditions (CP medians are reported at the end of the Results section).

We have made a comment about this at the end of the Results section.

Comment Figure 2:

Same as Supp. Fig. S14B, but only including trials where EV difference was in $[-1, +1]$.

- By pooling the trials from all cells into a single test, the statistical power for detecting weak effects is improved. However, it would be informative if the effects for single cells were calculated, perhaps as a supplement.

We have now applied cell-wise ROC analyses of residuals for choice detection (Methods 4.7) and show cell-wise results in Supp. Fig. S14A (grey, solid lines for each cell; black dotted lines for median across cells; black solid lines for mean across cells). We have again pooled the data in single tests by implementing Choice Probability analyses (Fig. 5E-F), showing median CP (downward triangle markers in Fig. 5E-F) matching AUC for cell-wise ROC analyses (Fig. S14A).

- The overall magnitude of these effects is unclear, relative to other studies. Is it possible to express the difference in distributions in Fig. 4F in terms of an ROC or other standard measure of classification accuracy?

We thank the reviewer for this suggestion, and, as mentioned above, we have followed it throughout the revisions of the manuscript, providing detailed ROC analysis.

We have now included cell-wise CP (Fig. 5E-F) and ROC (Supp. Fig. S14A) analyses, thus providing numerical evidence for improved choice detection when comparing looking

conditions. We have extensively edited Methods 4.4, 4.5, included new Methods 4.6, 4.7, and written all related results in the last three paragraphs of Results.

We thank Reviewer #1 once again for fruitful discussion and insights for further results interpretation.

Reviewer #2 (Remarks to the Author):

Please see attached MS Word doc.

We have now reported all comments in this file and replied at each point, marking our replies with left vertical bars and indentation.

We highly appreciate the effort and willingness of the reviewer to help us to improve the paper. We have responded to all comments and used all relevant suggestions to improve data analysis and presentation. We hope that the reviewer will be agreeable about our subjective preferences about presentation styles and ordering of the results.

Ferro et al. - 2024 - Nat Comm - OFC and post-stimulus gaze-modulated value
2nd round of reviews

1/11/24

I sincerely appreciate the authors' efforts to satisfy the reviewers' concerns and recognize that significant work has been done. As a result, the manuscript is clearer and the bases for the paper's claims are more thoroughly shown in supplemental figures. Unfortunately, a few fundamental issues still have not been resolved and represent weakness in the claims being made. To adequately support the paper's core claims, I believe changes to the manuscript's fundamental analysis are necessary, which would require a significant revision. I have attempted to outline these recommended changes below.

We thank the reviewer for the additional comments. We find that they mostly concern the interpretation of our results, in addition to a few new relevant analyses and suggestions. We have addressed all of them in the responses below.

1. The paper's core claim concerns the interaction between where an animal is looking and the representation of value- and choice-related signals in OFC. Because gaze is a series of relatively long periods of approximately static fixation interspersed by rapid changes in eye position (i.e., saccades), this interaction predicts that OFC activity should be related to the object of gaze during periods of fixation as bookended by saccades, and likewise be *less* related before and after the fixation-acquiring and fixation-leaving saccades, respectively. The prior work on gaze-OFC interactions—as cited in the present manuscript^{1,2}—adheres to this logic by analyzing OFC activity aligned to changes in fixation. That is, how does OFC activity change when gaze is directed to—and away—from the stimulus location of interest?

Indeed, in response to Reviewer 1, the authors perform and present the appropriate “gaze-aligned” analysis for a small portion of their overall claims in the revised supplement (Fig S14). However, the entirety of the data presentation in the main text, as well as the vast majority of claims and statistical hypothesis testing, rely on a very different analysis in which OFC activity is related to *average* gaze position. This alternative approach deviates from the established precedent in OFC^{1,2} (not to mention in visual systems neuroscience)

and, in my view, suggests a distorted logic and introduces several interpretive challenges that would be obviated by a gaze-aligned analysis (see below points). Moreover, when the recommended gaze-aligned analysis was done (Fig S14), the support for the paper's main claims was much less robust (see (2)).

We appreciate the above comments and agreement about the utility of gaze-aligned analyses (in previous Supp. Fig. S14). We have indeed moved previous gaze-aligned analyses to main text (previous Supp. Fig. S14 is now main Fig. 4). We have written additional extensive text in the last four paragraphs of Results to describe the new results, which agree with the previously shown results using the average-gaze analyses and added a new Supp Fig. S15. We have decided to keep both analyses because, as argued below, they strengthen each other, and also because we find the average-gaze analyses stronger in some aspects, and sound, as we explain next.

As discussed in the previous responses to the reviewer, since the task consists of only two alternative visual items presented at opposite sides of the screen, we deem sufficient to investigate the correspondence between behavioral tendency to look to either side and the activation of neural encoding, which we find relevant for the categorization of choice. We had previously tried to address this aspect by re-running spike rate analyses removing data in time points when gaze covered the center of the screen (Supp. Fig. S13).

We want to remark that we consider the average-gazed analysis sound for various reasons: (1) The time bins used to estimate average gaze position are very short, that is, 10ms. (2) The short time binning makes average gaze position approximately bimodal, especially in most relevant task times (see Fig. 1C, top; Fig. 1D; we have now included the epoch-wise distribution of gaze in new Supp. Fig. S1D, and the epoch-wise distribution of f_R in new Supp. Fig. S1E). The bimodality is evident in the distributions of gaze in 10 ms bins (Fig. 1C), where gaze is most often distributed outside the middle of the screen, also showing that cases where subjects keep their gaze around screen midline are infrequent, mostly coinciding with transient gaze relocation between screen sides (which we initially did not consider, and later analyze in depth in new Fig. 4 and Supp. Fig. S15). (3) The bimodality of gaze makes classification of right and left looking conditions in the 10 ms bins simple and well-defined with the *first* huge advantage that this method also maximizes trial usage. (4) Since the fraction of time looking at the right, f_R , computed in 10 ms bins peaks at the extreme values 0 and 1, splitting trials as a function of short-timed average gaze position is a good proxy for looked side categorization with the *second* huge advantage that we did not (initially) need to detect gaze-shifts, which are rather noisy (occur at varying times, and include many different transient configurations).

To add on the second advantage, there might be many different gaze shift configurations that could result hard to interpret (e.g., transient shifts across screen sides, or moving gaze within each screen side, or even transient, subsequent shifts), and hard to tackle at the neural level. Based on our experience, gaze-averaging resulted advantageous for an overall characterization of the relationship between short-time looked side and neural encoding.

Despite this, shift-locked analyses at delay epochs introduced in the first round of revision (previous Supp. Fig. S14, now moved to main text Fig. 4), and new shift-locked analyses at offer epochs (new Supp. Fig. S15, described in detail in below point 2.c, Comment Figure 1) to provide further, meaningful insights into shift-aligned neural encoding. We have explained the new results in the last two paragraphs of third Results section.

- a. The logic of the present paper—as in past work on visuospatial attention—is that gaze is a marker for the locus of attention. For example, say gaze were directed at position “-1” at time point A and at position “+1” at time point B. We might say that attention was directed at “-1” or “+1” depending on the time point. But if we were to average gaze across the two time points ($[-1, +1] = 0$), this does *not* imply that attention was, on average, directed at position 0. Indeed, at no point was animal attending to middle of screen! And yet, the vast majority of analyses in the present paper rely on averaging gaze position over some period and then relating the average gaze to OFC activity. (It’s possible the authors believe that the neural process under study reflects the average of otherwise bimodal attentional selection. But this would be a new hypothesis that would have to be convincingly proven before using it as an analysis tool.)

This concern does not apply to our analysis. Note that instances where gaze is kept around screen midline are rather infrequent. In other words, the average eye analysis is dominated by instances where gaze is stable (at least over periods >10 ms) on either screen side, thus average eye during every 10 ms reflects actual gazed side. This can be further checked by observing the bimodality of eye position (Fig. S1D) and by the fact that the fraction of time looking at the right, f_R , computed in 10 ms bins peaks at the extreme values 0 and 1 (Fig. S1E). Therefore, the cases where the average gaze is close to zero are limited.

In spite of the above observations, we have now enriched the analysis with gaze-shifts analyses (Fig. 4 in delay epochs, and Supp. Fig. S15 in offer epochs) discussed in the following.

- b. A number of interpretative challenges arise using the average-gaze analysis. One discussed at length in the prior round of reviews concerns the size, overlap, and therefore number of non-independent time bins. The authors helpfully explained their rationale for using very narrow time bins (10 ms) for eye position, and small-stepped (+10ms) overlapping (95%) time bins for neural data, as needed to capture the fast dynamics of eye movements. While it’s true that eye position can change very rapidly during saccades (on the order of 10 ms) these events are readily detectable and are relatively rare compared to the long periods of static fixation (200-300 ms). Therefore, the need to isolate static periods of eye position can be achieved by aligning neural data to large changes in visual fixation (i.e., saccades) and analyzing neural activity in a small handful of time bins on either side of the

eye movement. This would not only achieve the authors' goal of isolating static periods of eye movements, it would also presumably strengthen the analysis as variation in eye position would be substantially reduced during the period of neural analysis (because the period was *defined by fixed eye position*).

We agree with the above observations and acknowledge the relevance of the proposed shift-aligned analysis, previously included in Supplementary Information (previous Supp. Fig. S14), now moved to main Fig. 4. We have now also provided additional details about shift-aligned encoding at offer times in Supp. Fig. S15 and related response (point 2.c) below.

- c. The gaze-aligned analysis would also address a challenge in interpretation arising from uncertainty in the timecourse of *when* a change in eye position would impact neural activity. The present analysis, if I understand correctly, pairs a 10ms window of eye position with a 200ms window of neural activity, which (indirectly) hypothesizes that the effect of gaze on OFC activity occurs as early as immediately coincident with the change in eye position and that the bulk of the effect occurs within 200ms. From prior literature^{1,2}, it appears the OFC value-related response begins ~150ms after a change in gaze and usually lasts 100's of ms more. By using a gaze-aligned analysis, the precise nature of this timing becomes less critical, as one could use relatively large, non-overlapping time bins to capture the effect (e.g., 0 – 200ms, 200 – 400ms, 400 – 600ms, etc. after fixation). Moreover, because only a handful of time bins would be necessary, the previously discussed statistical concerns around multiple comparisons are significantly reduced. (As an aside, in their rebuttal, the authors justify the use of highly overlapping, sliding time windows by citing one of their own papers and one by Conen and Padoa-Schioppa. The latter used sliding windows to effect smoothness for visualization, while their statistical analyses were limited to a handful of pre-defined, non-overlapping windows, one for each epoch of the trial.)

We have provided above the rationale of why we chose the average gaze location analysis to start with, but we have also moved the shift-locked analyses from Supplementary Information (previous Supp. Fig. S14) to current main text Fig. 4, complementing with new shift-locked analyses in new Supp. Fig. S15.

In addition, specifically about timings, we have also shown that main results presented are qualitatively preserved when using shorter (150 ms) or longer (250 ms) time windows for spike counts (Supp. Fig. S10). Note that 250 ms would also cover the suggested ~150 ms delay in response + following ~100 ms duration. We experienced that the use of wider, non-overlapping time windows would not benefit our intent to dynamically track gaze on left/right screen side. Instead, using wider, non-overlapping windows would make considerations about looking sides blurred by trial-to-trial or short-time transitions within time windows (even more frequent in larger windows).

Another reason why we tended to keep shorter time windows is that task epoch times are rather short (offer1/offer2 are 400 ms; delay1/delay2 are 600 ms), and

gaze data include many possible shifts, resulting in extra difficulty in discerning data, and thus to interpret results across epochs.

We have removed the reference to Conen & Padoa-Schioppa, *J. of Neurophys.* 2015, in Methods 4.1 and updated reference to McGinty & Lupkin, *Nat. Neuro.* 2023.

In summary, analyzing and presenting the neural data by aligning to changes in gaze would significantly strengthen the interpretability of the data and therefore the ability to validate the paper's central claims. Moreover, the gaze-aligned analysis would align the paper with past studies of gaze-and-OFC and reduce, if not eliminate, many secondary problems discussed in the first round of reviews. Though I did not recommend the gaze-aligned analysis in my first review, Reviewer 1 did recommend this analysis explicitly and, after the authors clarified certain points in their rebuttal, I believe the gaze-aligned analysis is essential both because it directly addresses the core questions and because it addresses many of my concerns from the first round. While the authors show a gaze-aligned analysis for a subset of the main claims in the supplement (S14), I believe the analysis is sufficiently important to be applied to all main claims and be prominently presented in the main text. Finally, additional issues exist with the gaze-aligned analysis in its current form that should also be addressed (see below).

We thank the reviewer for the suggestions. We have moved previous Supp. Fig. S14 to main Fig. 4, provided the new Supp. Fig. S15, and written an additional paragraph in results to explain how all these results support all our previous interpretation at the end of third section of Results.

2. In Figure S14, the authors analyze the OFC data aligned to changes in gaze, which is much appreciated, as discussed above. Even still, there are additional issues to address.
 - a. Perhaps the most significant challenge of the gaze-aligned results is that they are much less compelling than the non-aligned original analysis in the main text. The percentage of significant cells is only just above the population-level significance threshold during the predicted times and is often only just below the threshold during the remaining times. In contrast, in the non-aligned analysis (Fig 3), the results are well above the significance threshold and remain so for 100s of ms during the predicted epochs.

Note that the results of the shift-locked analysis are consistent with the previous average-gaze analyses, with new results providing additional support to our initial findings. Also, we want to remind that shift-locked analysis does not need to be less noisy than gaze-average analysis both because of the behavioral variability in gaze-shift response, and because the temporal delay of signals reaching OFC after a gaze shift could be highly variable. In addition, minor transient differences in gaze-shift patterns could be present in trial-based data as opposed to the overall behavioral pattern identified in gaze-average analyses. All these sources of variability might introduce additional noise in the

gaze-shift locked analyses, and, also due to the reduced trial availability in gaze-shift analyses, the statistical power of detecting significant effects might be impacted. This is the reason we would like to keep the two types of analyses.

In addition, several specific predictions did not bear out in the gaze-aligned analysis:

- i. During delay1, when gaze changes R to L, there is no emergence of EVL encoding (D, left, dark orange). (The authors speculate this is because the subjects had not sampled offer1 during the offer1 period. If this were true, that should be demonstrated explicitly.)

Indeed, we have explicitly demonstrated this in our previous manuscript: We have shown in Supp. Fig. S6C that if subjects do not look at the first offer side during the *offer1* epoch, then looking left during *delay2* time does not evoke encoding of EV_L . In addition, we want to highlight that this is very infrequent (fraction of trials where this happens is reported in Supp. Fig. S6E): as soon as the first offer is presented, subjects tend to sample it by looking at the first offer screen side (Supp. Table ST2).

We are aware that the results shown in Supp. Fig. S6C consider the scenario where “subjects did not *LookL* during the last 200ms of *offer1* nor during *delay1*”, which is less restrictive of “subjects did not ever *LookL* during all *offer1* and *delay1* time”. The decision to only focus on the last 200ms was motivated by major reasons: i) the observation that gaze is more stable (hence its behavioral role interpretable) towards the end of such epoch times (Fig. 1C); ii) there needs to be physical delay from target appearance to gaze reach; iii) the numerical need to collect enough data to perform a relevant spike encoding analysis (though trial availability was still pretty low, Supp. Fig. S6E).

Since the Supp. Fig. S6C configuration already shows lack of EV_L encoding, we initially did not show the more restrictive case (no *LookL* during full *offer1* and *delay1*) as it would provide even less trials available and be even less likely to yield significant EV_L encoding (*reactivation*) for *LookL* at delay2.

We have now included Comment Figure below equivalent to Supp. Fig. S6 but here we make sure that gaze is not directed to left screen side during the whole *offer1/delay1*. Note that in this figure (as in Supp. Fig. S6) we access to trials where subjects do not look left during first offer and first delay time, but we still consider whether subjects *LookL/LookR* in 10 ms bins throughout task execution (using 200 ms boxcar windows for spiking activity as for the main analyses in Fig.3-5).

As it is evident by the fractions of trials available, not looking left during *offer1/delay1* is reasonably very infrequent given that as soon as first offer is displayed subjects tend to gaze the offer (left) screen side. Despite this, we hereby show that whenever this happens there is no encoding (*reactivation*) of first offer even if subjects look to first offer

side (left) during *delay2* time and did not previously look left during *offer1/delay1*.

We apologize if this was not remarked enough in previous responses to reviewer's comments.

Comment Figure 3: A, B, C. Respectively the same as Supp. Fig. S6A, S6C, S6E but subjects do not *LookL* in the whole *offer1 / delay1* time (instead of not *LookL* in the last 200 ms).

- ii. During delay2, encoding of EVL is not significant while gaze is directed to the left (except for during a vanishingly tiny period) and likewise cannot decay when gaze changes L to R (E, left, light orange).

We agree with the above observation. The fact that there is very weak detection of EV_L encoding during delay2 when subjects *LookL* could be in part due to the low number of trials available in this condition (approx. 6.7% of the total trials, Supp. Table ST2), and in part due to the peripheral view of the right offer, displayed during offer2 time, which may act as a distractor.

In the manuscript (new Fig. S15) and further below (response to point 2.c; Comment Figure 1 and related response) we have now performed analyses equivalent to gaze-locked analyses in Fig. 4 but focusing on offer presentation times.

We have now reworded “decay” in third results section, fourth paragraph and kept this observation in account in the remainder of the main text.

- iii. Of note, the latter observation, and combined with the lack of EVL encoding during uninterrupted viewing of the left side (J, left, dark orange), are inconsistent with the main thrust of the paper, and specifically the previously most compelling result (albeit using the non-aligned analysis) that EVL encoding emerges during delay2 conditioned on subject previously looking Right prior to delay2 and later looking Left during delay2 (main Fig 3G, left, dark orange).

There is a very short period of significance (~ 60 ms) for encoding of EV_L during delay2 when subjects *LookL*, but we agree with the reviewer that a larger response might have been expected if the event *LookL* during the whole delay2 time was more frequent. The weakness of this particular result coincides with the fact that in this configuration (current Fig. 4E) the fraction of trials used (Supp. Fig. S15D, *LookL*) is even smaller than in other configurations in current Fig. 4, because we heavily constrain the gaze to stay within the side opposite to most recently presented offer for the whole delay duration. Therefore, we argue that this observation does not invalidate our main interpretation of the results. We have made a comment about this at the end of Figure 4 caption.

We remind that we have moved previous Supp. Fig. S14 to main text Fig. 4 and described results in detail at the end of third Results section. In addition, we have now included new gaze-shift analyses in offer epochs in new Supp. Fig. S15 and confirmed that offer encoding occurs in post-shift times also at offer epochs (detailed in response to 2.c, Comment Figure 1 below).

- b. The fact that the gaze-aligned results are much less robust than the non-aligned analysis suggests one of two possibilities:
- i. Contrary to the paper's claim, OFC activity is not (robustly and consistently) dependent on the interaction between recent stimulus value and current gaze. And instead, OFC activity, during the delay period in particular, is better explained by another variable.

Or...

- ii. OFC does indeed represent the value of the stimulus located at (or recently at) the location of gaze, but that too many sources of variance obscure this finding in the current analysis. In this case, it's possible that a refined gaze-aligned analysis would reveal the hypothesized result and support the paper's main claims (which would be very interesting indeed!).

One suggestion is to perform the neural analysis in wide, non-overlapping time bins, as discussed above in point (1). This would obviate the need for the various bootstrapping and multiple comparison correction from the main text and may strengthen statistical power. It may also be less sensitive to the various dynamics between gaze and OFC response.

The technicality of time windows being overlapping in neural analyses is dealt with our permutation analysis (further checked in Supp. Fig. S12), and to our understanding it does not pose any statistical limitation to the possibility of detecting encoding of value. Note that we had previously adjusted this specific aspect of the methods in a way that trial order permutations preserved the temporal autocorrelation of spike series analyzed (next to last paragraph in Methods 4.1).

As indicated in previous responses, we experienced that using wide, non-overlapping time bins would complicate the possibility of tracking fast, time-resolved changes in gaze. This would result in blurred split between looking conditions that would prevent us from discerning trial-to-trial differences, hence impact the detection of linear interactions between spike counts and offer value for which we find that gaze is of crucial relevance.

Also, we started trying to consider segments where gaze is stable, but we found this extra problematic as it led us to examine many possible configurations where we had multiple screen side inspection segments of variable length, which in turn led to drastic reduction in trials availability. This was particularly problematic to conceptualize as the effect of inspecting given target could not be uniquely related to the specific (ipsi- or contra-lateral) screen side. We proposed that gaze-averaged analysis in 10 ms bins, sided by gaze shift time-locked analyses and by analyses where gaze was not shifted allow for exhaustive characterization fulfilling the purpose of trials usage maximization for improved statistical power.

Finally, performing the analyses in wide, non-overlapping time bins would not fully obviate to the need for trial-shuffling bootstrapping as we would still have uneven pools of trials in different epochs because looking conditions would still follow the presented or most recently presented offer (Fig. 1C, Supp. Table ST2).

Above all, we deem relevant the ability to develop a method that is sensitive to short-timed, hence dynamic task contingencies and manages to track the rather fast interplay between gaze and OFC response to encode offer value.

Another possibility is that the effect is more precisely sensitive to the overlap between gaze and (recent) offer location, and the analysis may be strengthened by accounting for whether (or how close) gaze was on the (recent) offer location. (A rough sense for this sensitivity could be gleaned by performing the gaze-aligned analysis during the offer period as a baseline for comparison; see below).

- c. One would like to see the gaze-aligned analysis for the offer periods, as well, when presumably the change in encoding would be highly sensitive to the change in gaze (toward or away). This would provide useful contextualization for the robustness of the delay-period analysis.

We thank the reviewer for the suggestion. We have now provided Fig. S15 showing the saccade time-locked analysis during offer epochs.

Copy of response to reviewer1:

By using gaze-shifts time-locking during offer presentation, we confirm the previously shown result (Supp. Fig. S1C) that shifts are predominant towards left for first offer presentation time (Comment Figure 1B) and towards right for second offer presentation time (Comment Figure 1C). As expected, we find that gaze shifts to left screen side during the *offer1* epoch coincide with post-shift increase in encoding of EV_L (Comment Figure 1D, dark orange) while gaze shifts to right screen side at *offer2* coincide with post-shift increase in encoding of EV_R (Comment Figure 1E, light purple). Conversely, shifting gaze to the opposite direction with respect to first offer presentation side leads to decrease in encoding of EV_L (Comment Figure 1E, light orange; note that it is the most frequent case). Note that the latter effect is not possibly found for EV_R as it was never presented before *offer2* time.

For completeness, we also include below (Comment Figure 1) results for trials where no shifts occurred during the offer epochs (pre-offer time could include shifts but they were not relevant for the respective presented offer encoding since respective offers had never been displayed before).

We can observe here pronounced encoding of ipsilateral offer whenever subjects pre-emptively looked at presentation screen side. As a remark, we find that the increases for EV_L (Comment Figure 1I, dark orange) and EV_R

(Comment Figure 1J, light purple) temporally coincide with gaze relocations *within* screen sides (no screen midline crossing) towards target offer locations (Fig. 1C). Importantly, the decrease in first offer encoding before second offer is presented in look left trials (Comment Figure 1J, LookL, dark orange) can indicate that second offer acts as a distractor when gaze is not directed to its side, though we could not investigate this aspect further as it is rather infrequent.

We did not previously go in depth in illustrating this quite complex scenario, but we have now enriched last paragraph of third results section in current version. We hope that results and additional analyses help clarify our findings and interpretation.

Comment Figure 1: A, D-E, F, I-J. Same as Fig. 4 but for offer epoch times. B-C, G-H. Fractions of trials available used in the respective panels below.

- d. The authors present a delay-aligned analysis (S14F-J), perhaps as a point of comparison for the gaze-aligned analysis. However, by averaging over multiple saccade-fixation sequences, this analysis still suffers from most of the concerns raised about the other non-aligned analyses in the main text. (The exception is that in S14J-F, the gaze is at least consistently on one side of the screen).

A preferred comparison is to align all analyses to the beginning of fixation, but then to compare the cases when the preceding fixation was on the same stimulus location or screen side (i.e., control condition) to when the preceding fixation was on the other location/side (i.e., test condition). This would control for the effect of changing fixation and isolate the OFC response due to a change in value of the current/recent stimulus.

We thank the reviewer for the proposal. The proposal however introduces additional problems. First, only the first gaze shift has long enough post-shift periods available for neural analysis. The duration of offer epochs is 400 ms and delay epochs last 600 ms, but it may take as long as 200 ms for first gaze shift to occur (e.g., at offer presentation, Supp. Fig. S1C). The subsequent gaze shift(s) will occur closer to the end of the epochs analyzed, leaving very little time for neural responses (hence low activity, due to the sparsity of OFC spikes). Second, it is possible that midline-crossing saccades happen earlier, while within-sides saccades happen later, so comparing the two would also have the confound that they tend to happen at different times, and presumably the neural processing would be contaminated by stimuli delays and so on.

- e. It was not clear whether the gaze-aligned analyses excluded timepoints occurring either after gaze has changed (this would be most rigorous) or at least after gaze has shifted back across the midline (such as illustrated in the example traces in S14A at ~400ms). In addition, were timepoints after the end of the present delay period excluded? These exclusions would greatly increase interpretability, and also obviate the need to condition every time bin on eye position.

We thank the reviewer for this important remark, we have now further clarified data exclusion criteria and addressed it.

The data were analyzed in delay times (in current Fig. 4) or in offer times (in new Supp. Fig. S15), by also including 400ms pre-shift time in all cases. In both cases, the timepoints exceeding the end of the delay/offer epochs are *excluded*.

We decided not to remove data for gaze-shifts posterior to first shift. We have now checked this aspect carefully and reported results for data exclusion after midline-crossing gaze-shifts posterior to first shift in Comment Figure below.

We initially did not explore screen midline shifts later the first screen midline shift, thus in Fig. 4 and Supp. Fig. S15 we did not remove data points for subsequent midline crossing in post-shift time. We did not apply data exclusion for gaze shifts

posterior to first shift since midline shifts do not imply instantaneous content switch in OFC encoding due to potential delays. Further, the switch in neural encoding could occur at variable delays across trials. Therefore, excluding data could penalize the detection of genuine encoding. Also, it is particularly relevant to note that the first post-shift time may be short (saccades may start much later than the epoch starts; Supp. Fig. S1C), making the neural context switch due to subsequent gaze switches possibly occur even later than the analyzed epoch duration.

Nevertheless, we checked the suggested data exclusion criteria, and reported results in comment figure below. As we can see, the main result, i.e., reactivation of first offer due to gaze-shift to left screen side in delay2 (Comment Figure 4E, dark orange) and other minor results are all retrieved. These results are to be compared with Fig. 4B-C, with the important difference that this time the fraction of trials posterior to first shift is decreasingly lower than the fraction of trials used for results in Fig. 4C (fractions of trials available in Fig. 4C are reported in Supp. Fig. S15B).

Comment Figure 4: A. Same as Fig. 4A, showing sketched illustration of timepoints selections and time-locking criteria, but also showing that time points posterior to midline-crossing shifts following first shift are excluded in B-E (dotted segments). B-C. Same as Supp. Fig. S15A-B but removing timepoints as in A. D-E. Same as Fig. 4B-C, but removing timepoints as in A.

We have specified in last paragraph of Methods 4.3:

“[...] we considered that the first midline-crossing gaze shifts could relate to value encoding reactivation even if they were transient (thus followed by further screen midline gaze shifts) due to variable response delays. Note that we did not consider gaze-shifts posterior to first gaze-shift, hence we used all timepoints starting at first shift time and until the end of current epoch.”

In Fig. 4 caption:

“[...]data are time-locked to the start of the earliest delay epoch gaze shift (involving

a midline gaze crossing from right to left screen sides –shift L–, or vice versa –shift R) and truncated at the end of delay epoch, allowing return on previous screen side;”

And in third section of Results:

“Midline-crossing gaze shifts during delay epochs were time-locked to their initiation, and could be transient, i.e., could entail post-shift midline crossings. We checked that the exclusion of data in timepoints posterior to midline-crossing gaze-shifts happening later than first shift yielded to qualitative match in the results.”

Overall fractions of trials with first / second shifts:

Delay1 shiftL 18% of total trials; shiftL then shiftR 29% of shiftL, 5% of total;
shiftR 45% of total trials; shiftR then shiftL 26% of shiftR, 12% of total.

Delay2 shiftL 63% of total trials; shiftL then shiftR 36% of shiftL, 23% of total;
shiftR 45% of total trials; shiftR then shiftL 41% of shiftR, 12% of total.

- f. Finally, the gaze-aligned analyses were only done for the EV-encoding questions. The gaze-aligned analyses should be performed for all results, especially the critical result of choice-related activity (i.e., Fig 4 C bottom, E, and F).

Note that previous Fig. 4A-D is now Fig.5A-D.

As can be seen in current Fig. 5C (bottom) and Fig. 5D (bottom), at delay times the choice encoding is low (in delay 2) or not significant (in delay 1). We decided not to perform choice encoding detection aligned to gaze-shifts as we observe that this would lead to reduced trial availability, which would further impact these results, possibly leading to n.s. level of encoding (also for delay2 choice).

We have now improved the study of the relationship between choice encoding and *SV* encoding by showing that choices tend to be congruent with the tuning of cell responses to offer *SV*. Previous Figures 4E-F have been replaced by improved cell-wise Choice Probability and cell-wise ROC detection methods (Methods 4.5, 4.6, 4.7, current Fig. 5E-F). Note that the latest methods developed further splits trials into subsets (to make choice variables used for *SV* computation and choice variables used for spike encoding across independent trial sets), making the eventual gaze-aligned analyses even less feasible.

3. The paper’s core claim is that look-at-nothing gaze correlates with a “reactivation” of the value representation in OFC, *and* that this “reactivated” value representation is used to drive the eventual decision. The authors support this claim by showing that delay-period OFC activity correlates with stimulus value conditioned on gaze (e.g., Fig 4D) and *separately* that trial-to-trial variability in OFC activity is predictive of choice (i.e., Fig 4F). From the present analysis, it is not clear whether the choice-predictive variance is related to the same variance that correlates with value or to some orthogonal source of variance. This distinction is critical for the authors’ claims. If the “reactivated” value representation is driving choice (as claimed), then it must be the value-related component of the OFC response that is predictive of choice.

This question is best illustrated at the population level, where one would test whether the dimension that best explains value is aligned/orthogonal to the dimension that best predicts choice.

For sequential recordings, as in the present dataset, Padoa-Schioppa offers a potential approach to this question: cull out the cells most sensitive to value and compare their choice predictivity to the remaining cells. Or more generally, test the overlap between the strength of choice predictivity and strength of value encoding.

We thank the reviewer for the question. We have now provided strong evidence in favor of our results interpretation, even improving correlational analyses (previous Fig. 4E-F) relating residual fluctuation in spiking activity with choice preference (improved in current Fig. 5E-F). Our new results corroborate on previous evidence that fluctuations of activity related to the choice are aligned to neuron offer value tuning.

We think that seeking additional, less-powered population analyses (in current dataset) would scatter in further interpretations and obscure our current interpretation. Regarding to “test the overlap between the strength of choice predictivity and strength of value encoding” we had already addressed it in previous Fig. 4E-F, improved in new *cell-wise* Choice Probability and ROC analysis in current Fig. 5E-F and in Supp. Fig. S14.

4. The paper labels the value-encoding delay-period activity as “reactivation” with the interpretation that it is a retrospective recalling of the formerly presented offer. In the first round of reviews, both Reviewer 1 and myself asked the authors to address an alternative interpretation. Specifically, whether the signal could also be interpreted as a prospective signal, that is, expected value given the choice. In their rebuttal, the authors argue against the signal being motoric or categorical, and therefore seem to dismiss the prospective interpretation. However, the signal need not be categorical to be prospective in nature. Moreover the prospective interpretation is not merely theoretical. Rather, it aligns with decades of past work in OFC reporting signals that *continuously* represent value conditioned on choice, or what Padoa-Schioppa et al. called “chosen value”³. (They also reported a distinct *categorical* choice signal, termed “chosen offer”.) The distinction between the retrospective “reactivation” and the prospective “chosen value” is essential and any evidence to resolve these interpretations would be most welcome. Short of that evidence, the present paper must acknowledge the ambiguity and assimilate the authors’ interpretation with the past literature.

We thank the reviewer for the comment. Indeed, we are not implying that our results purely reflect a memory reactivation of a formerly presented offer, as we find that fluctuations of activity also correlate (prospectively) with the choice. We have added related remarks in the second paragraph of Discussion to explain this better.

5. As discussed, a core claim is that the delay period OFC activity is predictive of choice. This is shown in two ways, each with some concerns.
- Most of the choice-related analyses are based on a regression model in which choice is a predictor (Fig 4C,D,E). The problem with this approach is that regression methods struggle to precisely segregate variance between correlated predictors, such as value and choice. A preferred approach would be to assess how much additional variance is explained by choice after attempting to explain as much variance as possible using the other predictors. This can be done by comparing two nested models that differ only by the inclusion of the choice predictor (see “coefficient of partial determination”).

This concern is precisely why in previous Fig. 4F (Current Supp. Fig. S14B) choice was *not* used as a regressor. We have now been extra careful in our new, additional analyses (Fig. 5E-F, Supp. Fig. S14) to always use non-overlapping trial sets to fit spike rates models and analyze regression residuals (we know that it may not be a common practice in the field, but as indicated in Methods 4.4 and 4.5 we are well aware of this issue, and we have tried to handle it extra carefully).

- The most rigorous test of choice predictive activity is in Fig 4F, which is cited extensively in the authors’ rebuttal. As the authors state in both the paper and their rebuttal, the widely accepted method for this analysis is the so-called “choice probability” (CP) metric, as per Britten et al.⁴, which is based on a non-parametric comparison of two putative pools of neural evidence for either of the two choices and is reported in terms of the probability of making a given choice given the evidence. This makes CP a particularly useful metric both because it allows comparison to many past papers and offers an absolute value that is readily interpretable on its own. It seems that the authors are familiar with this literature, and so it wasn’t immediately clear why the CP analysis was not used and reported.

We thank the reviewer for the suggestion. We have now extended previous analyses by adding Choice Probability analysis (Methods 4.5; new Figure 5E-F), corroborating our previous results.

6. The authors introduce a model of subjective value (SV), The trouble is that SV is at once a computed value based on a model fit to choices *AND* SV is a predictor in a separate model of firing rate that *also* includes choice as a predictor. This could lead to spurious interactions between free parameters in the firing rate model. There are at least a couple ways to mitigate this confound:
- Is it necessary to have a separate model of SV? Would it be possible to simply include the relevant terms on which SV is based (m , σ^2 , $EV=m*\sigma^2$) as stand-alone regressors in the firing rate model?
 - If SV is necessary as a computed value, cross-validation should be used to isolate the free parameters in the SV model from those in the firing rate model. That is, compute the SV term (which depends on free parameters $w_{\{1,2,3\}}$) using separate trials from those used in the model to predict firing rate (in which the independently computed

SV could then serve as a predictor). The authors note that they use cross-validation, but this is at the earlier level of computing SV, not in applying SV to predict firing rate.

We thank the reviewer for the suggestion, we have followed it. We have decided to split trials into two disjoint subsets, one used to estimate SV weights, and the other to perform linear regression on the spike rates using the estimated SVs (Methods 4.4). This strategy was also used for new analyses of residuals (Methods 4.5, 4.6 and 4.7).

We have clarified in detail this change in Methods 4.4, updated results in in Fig. 5A-D, related Supp. Table ST3, ST7, in new analyses (Methods 4.5, 4.6, 4.7), Fig. 5E-F and Supp. Fig. S14.

Another way to deal with the mentioned concern is to simply use EV as a regressor for spike rates, as it does not involve any new fitting. Indeed, we had previously performed the same analysis as in Fig. 5A-D by only focusing on EVs ($\eta = \beta_0 + \beta_1 EV_L + \beta_2 EV_R + \beta_3 ch$; Comment Figure below) and found results in agreement with Fig. 5A-D. This shows that the choice of SV or EV as a regressor is not instrumental for our results, but we follow the more careful method inspired by the reviewer's suggestion, we are thankful for that.

Comment Figure 5: Same as Figure 5 but using EVs instead of SVs.

- As discussed in the first round of reviews, the rationale for separate behavioral models for each temporal epoch still eludes me, particularly since the vast majority of predictors (e.g., EVL/R, mL/R, sigma2, sL/R) are identical across epochs (Fig 2C). The authors (thankfully)

also provide the preferred approach in which a single model contains one copy of each of the static predictors and separate predictors for the one variable (f_R) that differs across epochs (Fig 2D). The problem with presenting both models is that they draw conflicting conclusions, and yet the authors point to both to support their claims. In particular, f_R is not significant in most epochs in the combined model (2D), and yet the authors point to each of the epoch-specific models to claim that f_R is significant in all the epochs (lines 195-198).

We thank the reviewer for this comment. We would prefer to stick with our initial results presentation. Our main insight is that gaze predicts choice during delay2 epoch time, the most interesting epoch time. Regardless of the specific kind of analyses, this is the aspect we want to highlight.

The problem with the single-epoch model is that 1) it is effectively a multiple comparisons problem (repeating the same test on the same, or highly overlapping, data), and 2) does not account for within-trial interactions (e.g., how f_R during epoch X may compete for variance with f_R during epoch Y), whereas the full model does account for these interactions (which indeed is likely why f_R delay 2 survives in the full model, but the other f_R for the other epochs do not).

We understand the reviewer's point. Despite this, we would rather keep the results presentation the way it is by now. As stated above, the central result is that f_R predicts choice in the delay2 epoch time is preserved across analyses, so we think that our presentation and interpretation of the results is fairly supported by results.

The authors say the separate models are justified because starting with the full model would be "too much of a jump". I suppose that this is a subjective assessment, but to at least this reviewer, the 11 bars in Fig 2D are simpler than interpreting and comparing the 8 bars/epoch * 6 epochs = 48 bars in 2C!

The authors also justify the separate model to mirror the visualization in Fig 2A. I agree with the visualization approach in Fig 2A. But the rationale for a particular visualization approach does not necessarily justify the same approach for the statistical analysis. On the contrary, it is typical to use one approach for visualization and another for rigorous analysis. In fact, Fig 2A and B are great examples of this, where trials are stratified into separate, discrete groups to allow for plotting multiple variables on a single axis, whereas the statistical analysis would include all variables in their continuous form (as in Fig 2C,D).

If authors are looking for more statistical power to test f_R , they could include an "overall f_R " term, or " f_R in offer1+2+delay1+2". In the spirit of gradually building a model up, this would be more typical: a simple model that pools all epochs together, then a more complex model that has separate *terms* (not *models*) for each epoch.

We appreciate reviewer's suggestions and willingness to improve our results presentation. However, having a more powerful test was not the main goal of

analyses in Fig. 2C-D, rather we tried to build a statistical analysis that would allow us to dissect and assess the effects of f_R across task epoch times.

As a separate point, it's unclear to me why the authors use the difference of offers for plotting (e.g., $EVR-EVL$) in Figs 2A,B — which implies that the difference is the critical variable for choice, and makes total sense! — but use the independent, absolute values of the offers (e.g., EVR , EVL , etc.) in the model. Why not include the difference terms in the model? If one thought that absolute magnitude mattered in addition to the difference in value, one could include a term $EVR+EVL$. Besides making things more consistent across analyses and making better intuitive sense, this model might actually explain the data better and (possibly) help pull out the variables of interest.

For linear models of spiking activity, we have indeed used separate models for EV_L and EV_R , since we were interested in discerning separate effects of looking at either offer side whenever they were (or were not) displayed on the screen.

The use of $EV_L - EV_R$ would have only made sense when considering encoding of value difference variable posterior to offer2 presentation, since the subject has no access to EV_R before its presentation. In addition, combining the two EV s in one single variable would have prevented us from studying the relationship between presentation side and value difference encoding as there we would find no one-to-one correspondence between value difference and gazed side.

The choice of this specific SV model is motivated in Supp. Fig. S4 where we show that we choose the model which better allows choice prediction given task variables, above the performances of using both EV_R , EV_L (or their linear combination).

8. The authors use the term “reevaluation” to refer to the choice-predictive activity during the delay period. However, it was not clear why this process constitutes “Reevaluation” and not merely “evaluation”. That is, what distinguishes the phases of “evaluation”, “reevaluation”, and in some cases “further reevaluation” in the present paper? The distinction between evaluation and reevaluation is important because there is a large literature on REevaluation in which it is shown that subjects have reached a decision, but then change that decision (usually in the face of new information). But identifying REevaluation requires some evidence that a decision was (tentatively) reached. It would be helpful if the authors could highlight this evidence.

We thank the reviewer for the remark. We apologize by using “further reevaluation”. We meant “further evaluation” and now it has been corrected.

We would not like to overload terms: when we introduce reevaluation we simply mean that there is reactivation of a previously encoded value, and that residuals

fluctuations tuned to such reactivation predicts the choice. We have now clarified this in the first sentence of the fourth section of Results. We do not want to imply that there was a choice commitment before.

9. I agree with Reviewer 1 that re-coding the offers as “Left” and “Right”, even when the offer was presented on the opposite side as its label, is confusing, and seemingly unnecessarily so. Why not just refer to the offers as they actually are—“1st” and “2nd”—and then combine across sides for most analyses? This would seem advantageous not only because it is a more accurate description of the actual task, but it also would avoid certain confusing exceptions to the current rule, such as when analyzing the effect of side (e.g., Fig 2C), when “Left” really does mean “Left”!

We previously commented to Reviewer 1 about this, and we argued that “left” and “right” were preferred due to their matching to “look left” and “look right” during task execution. We seemed to agree on this aspect, and we would rather stick with our initial nomenclature at this stage.

10. Fig 2C,D: colors are difficult to distinguish, particularly S_{LR} and f_R vs. EV_L and m_L (2C) and f_R offer 1/2 vs. EV_L and m_L (2D).

We thank the reviewer for noticing this. We have now made S_{LR} gray, and f_R black, please check new Fig. 2C-D.

11. Line 111: the definition of variance includes the reward magnitude (“ $mp(1-p)$ ”)? Typically, Bernoulli variance is “ $p(1-p)$ ”, no?

We agree with reviewer that if we consider the variance for single Bernoulli variable, we should consider $p(1 - p)$. Here we assume that we have m (normalized to $m = 1, 2, 3$ for small, medium, large reward) independent Bernoulli variables, which means that their variance would be $mp(1 - p)$. We have clarified this in the Methods (third paragraph of first methods section), where we report that there is no qualitative difference in the choice prediction models studied (see below comment figures).

Comment Figure 6: A, B. Respectively same as Fig. 2C, 2D, but for $\sigma^2 = p(1 - p)$.

Comment Figure 7: Same as Fig5 A-D but with SV based on $\sigma^2 = p(1 - p)$. No qualitative differences are observed.

References

1. McGinty, V. B., Rangel, A. & Newsome, W. T. Orbitofrontal Cortex Value Signals Depend on Fixation Location during Free Viewing. *Neuron* **90**, 1299–1311 (2016).
2. McGinty, V. B. & Lupkin, S. M. Behavioral read-out from population value signals in primate orbitofrontal cortex. *Nature Neuroscience* **26**, 2203–2212 (2023).
3. Padoa-Schioppa, C. & Assad, J. A. Neurons in the orbitofrontal cortex encode economic value. *Nature* **441**, 223–6 (2006).
4. Britten, K. H., Newsome, W. T., Shadlen, M. N., Celebrini, S. & Movshon, J. A. A relationship between behavioral choice and the visual responses of neurons in macaque MT. *Vis Neurosci* **13**, 87–100 (1996).

We thank Reviewer #2 once again for the comments and for providing feedback helping us to improve the manuscript.

Reviewer #3 (Remarks to the Author):

The authors have addressed all my comments.

We thank once again the Reviewer #3 for the comments provided.

REVIEWERS' COMMENTS

Reviewer #1 (Remarks to the Author):

My thanks to the authors for answering my previous questions. To recap, my previous review highlighted two issues. The first was that the data supporting the claim of gaze-triggered reactivation of value coding was, in my opinion, ambiguous. The second was that the analysis linking reactivated value signals to choice outcomes (reevaluation) was not complete and could be improved on technical grounds. After reading the revised results, my opinion on the reactivation claim has not changed. The reevaluation analyses are more complete and have improved technically, but they show the same ambiguity as the reactivation results.

With regards to the reactivation claim, in my previous review I noted the positive results showing significant encoding of EV_L following leftward gaze shifts during delay 2 (Current Fig. 4C, left), but negative results for EV_L following left gaze shifts during delay 1 (Fig 4B left) and weak/ambiguous results for EV_R following right shifts during delay 2 (Fig. 4C, right). These weak/negative results are explained in the response to reviews and in a newly added results paragraph (line 328 in the clean doc) as resulting from the gaze conditions occurring during the offer epochs. Namely, that LookL events during delay1 occur mainly in trials where the monkey didn't look at the left-located offer1 and therefore are trials where EV_L is expected to be weak in the first place. Likewise, LookRs during delay2 mostly happen when the monkey didn't look at (and the OFC didn't encode) the right-located offer2. In other words, the claim is that the gaze-triggered reactivation of value signals during the delays only occurs contingent on the targets being viewed and encoded during the offer epoch.

On the one hand this is a plausible explanation. On the other hand, the data do not directly show this, because the results in Fig. 4B and 4C collapse the data across the critical condition (whether the monkey looked at the immediately preceding offer during the offer epochs). It's unclear why the data were not shown split by this condition, given that doing so could shed needed light on this particular point. If there aren't enough trials with the appropriate gaze behavior in both the offer and delay epochs, this means that the data aren't sufficient to furnish the evidence necessary to settle this issue. Doing my best to judge the data as it's presented, I still see significant ambiguity for the reactivation claim.

Regarding the reevaluation results: My thanks to the authors for the additional analyses and for taking my suggestions. Taken together these new results suggest an interesting asymmetry, similar to that for the re-activation results above: In LookL trials, there is good evidence showing that residual activity predicts subsequent choices. This is given by the significantly shifted distributions (relative to 0.5) for LookL trials in Fig. 5E and 5F; by the deflections of the ROC curves in Supp Fig.

14 A (left column); and by the differences in the distributions in Supp Fig. 14B (left column, top). In contrast, in LookR trials these effects are all much weaker: the distributions in Fig. 5E and F are much closer to 0.5, and the ROC curves in Supp Fig. 14 are much closer to the unity line.

In my opinion, these results suggest that re-activation in LookL trials is predictive of choice, but that the same effect does not occur in LookR trials. It appears that the evidence for the choice-prediction-from-reactivation effect is also mixed.

In summary, my opinion is still that the evidence for the most novel claims of the paper (gaze-triggered reactivation and reevaluation) is mixed. For reactivation, the evidence is solid for one out of the three conditions tested (LookL in delay2) but weak or non-existent for the other two conditions (LookL in delay1 and LookR in delay2, possibly because there are not a sufficient number of trials where the gaze behavior would allow them to be tested. Likewise, for the reevaluation claim, the evidence is solid in one of the two tested conditions (LookL), but absent in the other (LookR).

It is of course the author's prerogative to interpret their results as they see fit. If this were a manuscript from my laboratory, I would be cautious, and would not overlook the nuance in these results. Specifically, I would give the positive and negative results equal weight in the results section, and in the discussion section I would dedicate significant space discussing the limitations of the findings (e.g. the trial number issue), potential alternative interpretations of the data, and how these ambiguities might be tested in future experiments. However, in the discussion there is no mention of the ambiguous results and no explicit discussion of limitations or alternative hypotheses. This is in my opinion a significant oversight, especially given that some of these issues were raised in the previous round of reviews.

Reviewer #3 (Remarks to the Author):

Reviewer 3: The editor has asked, in the absence of further comments from R2, that I look at the responses made by the authors to R2. I have looked at the re-revised manuscript and the rebuttal letter and have focussed especially on figure 4 which is a particular point of contention.

R2 has made some very well reasoned arguments about supplementing the authors' original analysis of neural activity which is mainly reported in figure 3. As far as I can see, however, the authors have now done this in figure 4 and the results shown in figure 4 appear to confirm the

authors' interpretations of their results. The effects are slightly weaker in figure 4 but they remain clear and significant and seem to be based on a reasonable sample size.

The key question here is whether looking at an empty space where a stimulus used to be leads to reactivation of the representation of the stimulus' value. R2, and relatedly R1, point out that it is important not to look at time bins of data where the eye gaze position is, on average, at a given position because such a time bin may contain a mixture of trials where the gaze is a variety of positions. The authors, however, first point out that the eye gaze positions are distributed very bimodally – there are really just two different places that the gaze tends to be. In addition, R1 and R2 have suggested assessing whether there is activity linked to the object previously at a gazed location in a manner that is more strictly time-locked to the initiation of gaze to that location. The authors have supplemented their manuscript by reporting such an analysis in figure 4. As far as I can see, the effects are present when they should be and absent when they should.

The reviewers' points have questioned the neural analysis rather than the behavioural analysis and the inferences drawn from the behavioural analyses remain warranted.

In summary, I think that this is an interesting and novel claim that is supported by the data presented.

RESPONSES TO REVIEWER COMMENTS

We have now reported all comments in this file followed by point-by-point responses, denoted with left vertical bars and indentation.

Reviewer #1 (Remarks to the Author):

My thanks to the authors for answering my previous questions. To recap, my previous review highlighted two issues. The first was that the data supporting the claim of gaze-triggered reactivation of value coding was, in my opinion, ambiguous. The second was that the analysis linking reactivated value signals to choice outcomes (reevaluation) was not complete and could be improved on technical grounds. After reading the revised results, my opinion on the reactivation claim has not changed. The reevaluation analyses are more complete and have improved technically, but they show the same ambiguity as the reactivation results.

With regards to the reactivation claim, in my previous review I noted the positive results showing significant encoding of EV_L following leftward gaze shifts during delay 2 (Current Fig. 4C, left), but negative results for EV_L following left gaze shifts during delay 1 (Fig 4B left) and weak/ambiguous results for EV_R following right shifts during delay 2 (Fig. 4C, right). These weak/negative results are explained in the response to reviews and in a newly added results paragraph (line 328 in the clean doc) as resulting from the gaze conditions occurring during the offer epochs. Namely, that LookL events during delay1 occur mainly in trials where the monkey didn't look at the left-located offer1 and therefore are trials where EV_L is expected to be weak in the first place. Likewise, LookRs during delay2 mostly happen when the monkey didn't look at (and the OFC didn't encode) the right-located offer2. In other words, the claim is that the gaze-triggered reactivation of value signals during the delays only occurs contingent on the targets being viewed and encoded during the offer epoch.

One the one hand this is a plausible explanation. On the other hand, the data do not directly show this, because the results in Fig. 4B and 4C collapse the data across the critical condition (whether the monkey looked at the immediately preceding offer during the offer epochs). It's unclear why the data were not shown split by this condition, given that doing so could shed needed light on this particular point. If there aren't enough trials with the appropriate gaze behavior in both the offer and delay epochs, this means that the data aren't sufficient to furnish the evidence necessary to settle this issue. Doing my best to judge the data as it's presented, I still see significant ambiguity for the reactivation claim.

Regarding the reevaluation results: My thanks to the authors for the additional analyses and for taking my suggestions. Taken together these new results suggest an interesting asymmetry, similar to that for the re-activation results above: In LookL trials, there is good evidence showing that residual activity predicts subsequent choices. This is given by the significantly shifted distributions (relative to 0.5) for LookL trials in Fig. 5E and 5F; by the deflections of the ROC curves in Supp Fig. 14 A (left column); and by the differences in the distributions in Supp Fig. 14B (left column, top). In

contrast, in LookR trials these effects are all much weaker: the distributions in Fig. 5E and F are much closer to 0.5, and the ROC curves in Supp Fig. 14 are much closer to the unity line.

In my opinion, these results suggest that re-activation in LookL trials is predictive of choice, but that the same effect does not occur in LookR trials. It appears that the evidence for the choice-prediction-from-reactivation effect is also mixed.

In summary, my opinion is still that the evidence for the most novel claims of the paper (gaze-triggered reactivation and reevaluation) is mixed. For reactivation, the evidence is solid for one out of the three conditions tested (LookL in delay2) but weak or non-existent for the other two conditions (LookL in delay1 and LookR in delay2, possibly because there are not a sufficient number of trials where the gaze behavior would allow them to be tested. Likewise, for the reevaluation claim, the evidence is solid in one of the two tested conditions (LookL), but absent in the other (LookR).

It is of course the author's prerogative to interpret their results as they see fit. If this were a manuscript from my laboratory, I would be cautious, and would not overlook the nuance in these results. Specifically, I would give the positive and negative results equal weight in the results section, and in the discussion section I would dedicate significant space discussing the limitations of the findings (e.g. the trial number issue), potential alternative interpretations of the data, and how these ambiguities might be tested in future experiments. However, in the discussion there is no mention of the ambiguous results and no explicit discussion of limitations or alternative hypotheses. This is in my opinion a significant oversight, especially given that some of these issues were raised in the previous round of reviews.

We thank the reviewer for agreeing on the interpretation of results about shift-locked analyses. We apologize if our attempt to simplify a quite complex picture resulted less clear in previous versions.

About the doubts on whether 'gaze-triggered reactivation of value signals only occurs contingent on the targets being viewed and encoded during the offer epoch', we have now confirmed that results in Fig. 4B-C would hold when performing the suggested analysis. We have performed shift-locked analyses by fixing pre-shift gaze (200 ms prior to shift initiation) on the side of the respectively presented offer (left during offer1 and right during offer 2), we have included results in Comment Figure below and in new Supp. Fig. S15E-H. We have selected 200 ms prior to shift initiation instead including the whole 400 ms offer presentation time since it would be too restrictive given the fast dynamics of gaze relocation and lead to poor trial availability. We set pre-shift time to 200 ms by observing that fixation to visually displayed offers most often started 200 ms after offer cue onset (Fig. 1D).

Comment Figure: Same as Fig. 4B-C, but conditioning pre-shift gaze (200 ms prior to shift initiation) on the side of offer presentation (left in offer1, right in offer2).

About the reactivation and reevaluation, we gladly appreciated the suggestions of ROC based analyses, which gave much more rigor to our previous methods based on correlations and comparisons of residual magnitudes. We agree with the reviewer that the most prominent effect is found when considering LookL at delay2 time. For the way we designed our analyses, LookL at delay2 refers to gazing back at first offer location soon after second offer presentation. We found this to be the most relevant scenario as we hypothesized that this entails switching neural content to first offer value in OFC, i.e., to a “reactivation” of its value representation posterior to visually sampling a second, distracting offer. As the reviewer pointed out, the evidence for this effect is solid by the results presented.

We hope to have improved the definition of reactivation and reevaluation, specifying that we mainly refer to first (left) offer value encoding at delay2 epoch. This is sided by the observation that residual fluctuations in spiking activity favor second choice when gazing at right screen side at delay 2 time, even though reactivation and/or reevaluation are not addressed in this case due to recency of second offer presentation at delay 2 epoch time.

We have better specified the definition of reactivation and reevaluation in the sixth paragraph of third Results section, and at the end of third from last paragraph of Results section, by always specifying “first offer” reactivation.

We thank the reviewer for many fruitful comments, helping to substantial improvent of the paper.

Reviewer #3 (Remarks to the Author):

Reviewer 3: The editor has asked, in the absence of further comments from R2, that I look at the responses made by the authors to R2. I have looked at the re-revised manuscript and the rebuttal letter and have focussed especially on figure 4 which is a particular point of contention.

R2 has made some very well reasoned arguments about supplementing the authors' original analysis of neural activity which is mainly reported in figure 3. As far as I can see, however, the authors have now done this in figure 4 and the results shown in figure 4 appear to confirm the authors' interpretations of their results. The effects are slightly weaker in figure 4 but they remain clear and significant and seem to be based on a reasonable sample size.

The key question here is whether looking at an empty space where a stimulus used to be leads to reactivation of the representation of the stimulus' value. R2, and relatedly R1, point out that it is important not to look at time bins of data where the eye gaze position is, on average, at a given position because such a time bin may contain a mixture of trials where the gaze is a variety of positions. The authors, however, first point out that the eye gaze positions are distributed very bimodally – there are really just two different places that the gaze tends to be. In addition, R1 and R2 have suggested assessing whether there is activity linked to the object previously at a gazed location in a manner that is more strictly time-locked to the initiation of gaze to that location. The authors have supplemented their manuscript by reporting such an analysis in figure 4. As far as I can see, the effects are present when they should be and absent when they should.

The reviewers' points have questioned the neural analysis rather than the behavioural analysis and the inferences drawn from the behavioural analyses remain warranted.

In summary, I think that this is an interesting and novel claim that is supported by the data presented.

We thank the reviewer for the positive response.